# MORE: A Multilingual Document Parsing Benchmark and Evaluation

**Long Xu** [* 1]  **Binghong Wu** [* 1]  **Tinghao Yu** [* 1]  **Hao Feng** [1]  **Zhenyu Huang** [1]  **Haoqing Jiang** [1]  **Yunhao Wang** [1]
**Shuo Huang** [1]  **Feng Zhang** [1]

## Abstract

Multilingual documents encapsulate rich regional cultures, scientific discoveries, and historical records. Parsing this content into structured, machine-readable formats is critical for unlocking global knowledge. However, existing benchmarks predominantly focus on high-resource languages like English and Chinese, creating an *evaluation blind spot* concerning model performance on other languages. While recent Vision-Language Models (VLMs) claim support for hundreds of languages, the lack of ground truth makes it impossible to empirically verify these capabilities. To bridge this gap, we introduce **MORE**, a large-scale benchmark designed for multilingual document parsing evaluation. MORE distinguishes itself through three key dimensions: (1) **Unprecedented Scale**: It covers **149 languages**, making it the most linguistically diverse benchmark to date; (2) **Structural Complexity**: Unlike previous works, it extends evaluation beyond plain text to include structural elements such as code blocks, tables, and catalogs; and (3) **Data Authenticity**: All samples are curated from real-world documents via a model-assisted, human-refined annotation pipeline. We evaluate state-of-the-art models using MORE, establishing new performance baselines for long-tail languages and validating the benchmark's effectiveness in diagnosing model capabilities in realistic, diverse scenarios. The MORE dataset will be available at `https://github.com/zimoqingfeng/MORE`.

## 1. Introduction

Documents are the ultimate vessel of human civilization. To unlock their full potential, Multilingual Document Parsing

---
*Equal contribution [1]Tencent, Shenzhen, China. Correspondence to: Binghong Wu <lemonbhwu@tencent.com>.

*Proceedings of the 43rd International Conference on Machine Learning*, Seoul, South Korea. PMLR 306, 2026. Copyright 2026 by the author(s).

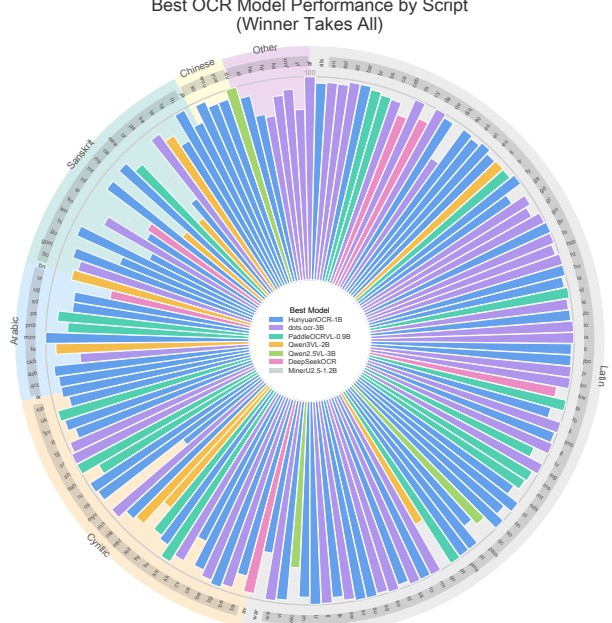

*Figure 1.* The *Winner-Takes-All* landscape of multilingual OCR. We evaluate state-of-the-art models on 149 languages spanning diverse script families (outer ring). The color of each bar denotes the model achieving the highest accuracy for that language, illustrating the competitive landscape across diverse linguistic regions.

stands as the critical link transforming them into machine-actionable knowledge for Large Language Models (LLMs) and Retrieval-Augmented Generation (RAG) systems (Touvron et al., 2023; Lewis et al., 2020). Yet, current advancements (Blecher et al., 2023; Kim et al., 2022; Lv et al., 2023; Wei et al., 2024b;a; Feng et al., 2025) remain largely confined to English and Chinese, neglecting the vast, culturally rich landscape of global languages (Ouyang et al., 2025). This narrow focus creates a massive *Knowledge Blind Spot* in modern AI, leaving a gap in global understanding that we cannot ignore. Thus, establishing robust multilingual parsing capabilities serves as the cornerstone of a global Artificial General Intelligence (AGI) ecosystem.

Driven by advancements in Vision-Language Models (VLMs), document parsing has entered a new era of mul-

*Table 1.* Comparison of statistics, language coverage, and supported tasks across different multilingual datasets.

| DATASET | LANGUAGE | IMAGE | TASK | | | | | | OPEN SOURCE |
|---|---|---|---|---|---|---|---|---|---|
| | | | TEXT | FORMULA | TABLE | CODE | CATALOG | READING ORDER | |
| OMNIDOCBENCH (OUYANG ET AL., 2025) | 2 | 1200 | ✓ | ✓ | ✓ | | | ✓ | ✓ |
| ICDAR RRC-MLT (NAYEF ET AL., 2017) | 6 | 1800 | ✓ | | | | | | ✓ |
| CC-OCR (YANG ET AL., 2025) | 10 | 1500 | ✓ | | | | | | ✓ |
| MISTRAL-OCR | 11 | - | ✓ | | | | | | |
| MDPBENCH (LI ET AL., 2026D) | 17 | 3400 | ✓ | ✓ | ✓ | | | ✓ | ✓ |
| XDOCPARSE (LI ET AL., 2025B) | 126 | - | ✓ | ✓ | ✓ | | | ✓ | |
| **OURS** | **149** | 1288 | ✓ | ✓ | ✓ | ✓ | ✓ | ✓ | ✓ |

tilingual capability. Leading models—including Surya[1], Qwen-VL (Bai et al., 2025b), dots.ocr (Li et al., 2025b), and HunyuanOCR (Team et al., 2025)—leverage massive training datasets to achieve impressive generalization, asserting support for hundreds of languages without the need for language-specific engineering. This rapid progress fosters a prevailing impression, namely that the barrier to universal multilingual understanding has been effectively dismantled.

While model capabilities have surged, the related scientific benchmarks have stagnated. Mainstream benchmarks, e.g., OmniDocBench (Ouyang et al., 2025), OLMBench (Poznanski et al., 2025), FoxBench (Liu et al., 2024), are still predominantly centered on English and Chinese, frequently discarding other languages as noise during evaluation. This creates a dangerous blind spot for long-tail languages: although VLMs may output text in languages like Thai or Amharic, there is a lack of gold-standard ground truth to verify their accuracy. In this uncertainty, neither academia nor industry can evaluate parsing results across the majority of languages based on empirical evidence. This prompts a critical inquiry: *Are we expanding language coverage blindly, with no way to measure actual precision?*

To bridge the widening gap between rapid model evolution and lagging evaluation standards, we introduce MORE (*Multilingual Document Parsing Benchmark*). As visualized in Figure 1, MORE enables the first quantitative comparison across a vast linguistic spectrum. Built upon the core pillars of data authenticity, linguistic diversity, and scenario complexity, it establishes a robust yardstick for assessing model performance in real-world, global contexts.

A detailed comparison between MORE and existing multilingual benchmarks is presented in Table 1. In terms of language coverage, we have significantly expanded the testing boundary to include 149 languages, establishing this work as **the most linguistically comprehensive benchmark** to date. Regarding data construction, every sample in MORE is collected directly from real-world sources and remains entirely raw, devoid of any synthesis or artificial processing. To guarantee annotation quality, we employ a model-

assisted, human-refined pipeline to minimize noise and ensure Ground Truth reliability. Furthermore, distinguishing our work from mainstream benchmarks, we introduce support for the evaluation of structured parsing elements including code blocks and catalogs, thereby addressing a critical gap in community assessment.

We list our contributions as follows:

- **Largest Language Scale Benchmark**: We release the first document parsing benchmark covering **149 languages**. This scale significantly surpasses existing datasets and aligns with the linguistic spectrum of current state-of-the-art models.

- **Expanded Structural Evaluation**: Beyond standard elements (text, tables, formulas), we extend our evaluation to include structures like code blocks and catalogs. Though naturally sparse in long-tail scenarios, they ensure a realistic assessment of document complexity.

- **Comprehensive Analysis**: We conduct an exhaustive evaluation of existing advanced models. The results establish baselines for **under-represented languages** and validate the benchmark's effectiveness in distinguishing model capabilities.

**Conflict of Interest Disclosure** All authors are employed by Tencent, the company that leads the development of HunyuanOCR, one of the models evaluated in this paper.

## 2. Related Work

### 2.1. Advancements in General VLMs and Specialized Document Parsers

Recent advancements in document parsing VLMs show a bifurcated trend: general models are prioritizing broad generalization, whereas specialized models are honing their deep, domain-specific capabilities.

General VLMs (Chen et al., 2024; Guo et al., 2025) leverage robust zero-shot capabilities to handle diverse tasks. For instance, the Qwen2.5-VL series (Bai et al., 2025b),

---

[1]https://github.com/datalab-to/surya

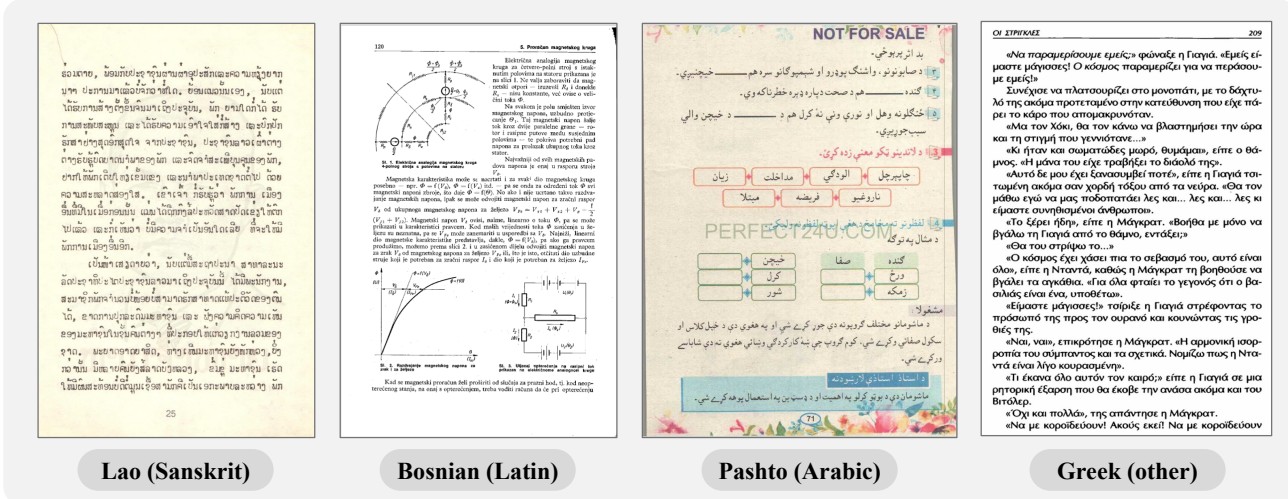

*Figure 2.* Diverse document samples from the MORE benchmark. The dataset covers a wide spectrum of languages and scripts, including Lao (Sanskrit script), Bosnian (Latin script), Pashto (Arabic script), and Greek. These samples highlight the benchmark's diversity in handling complex layouts and diverse content across 149 languages.

established a strong baseline with support for over 10 languages. Building on this, Qwen3-VL (Bai et al., 2025a) has expanded its linguistic repertoire to 32 common languages, delivering document parsing capabilities that rival large-scale closed-source models such as Gemini Pro (Comanici et al., 2025) and GPT (Achiam et al., 2023).

Conversely, Specialized VLMs prioritize parameter efficiency and extensive language coverage through domain-specific optimization. For instance, dots.ocr (Li et al., 2025b) demonstrates robust utility in multilingual scenarios. Advancing this efficiency, PaddleOCR-VL (Cui et al., 2025) supports 109 languages and achieves impressive performance on OmniDocBench (Ouyang et al., 2025) with only 0.9B parameters. In a similar vein, DeepSeekOCR (Wei et al., 2025) delivers consistent recognition across over 100 languages. Finally, pushing the boundary even further, Hun-yuanOCR (Team et al., 2025) extends support to over 130 languages, significantly enhancing performance on long-tail scripts. These developments highlight a shift towards models balancing high precision with linguistic inclusivity.

## 2.2. Evolution of Document Benchmarks

As model capabilities grow, evaluation benchmarks have transitioned from pure text recognition to complex structural parsing and semantic understanding.

Foundational benchmarks like MLT2017 (Nayef et al., 2017) primarily focused on multilingual text detection and recognition across 6 languages. The introduction of MTVQA (Tang et al., 2025) marked a shift towards semantic evaluation by incorporating multilingual visual question answering. While datasets like DL-CSVTR (Li et al., 2025a) highlight lay-

out complexities in natural scenes, more recent benchmarks address the structural challenges of real-world document parsing: OmniDocBench (Ouyang et al., 2025) emphasizes parsing precision but is limited to 2 languages (Chinese and English); CC-OCR (Yang et al., 2025) evaluates comprehensive multi-task capabilities across 10 languages; and Mistral-OCR[2] introduces novel dimensions for document understanding covering 11 languages. Notably, XDocParse (Li et al., 2025b) significantly expands this scope, rigorously assessing 126 languages in realistic scenarios.

## 2.3. Limitations of Existing Benchmarks

Despite these advancements, evaluation benchmarks lag significantly behind model capabilities. While models now support 130+ languages (Li et al., 2025b; Cui et al., 2025; Team et al., 2025), existing benchmarks often suffer from narrow linguistic scope and restricted scenario diversity (Ouyang et al., 2025; Li et al., 2026a;b;c;d). Consequently, current datasets are insufficient for the quantitative assessment of modern, hyper-multilingual VLMs. This disparity necessitates a comprehensive benchmark to evaluate models across a significantly broader spectrum of languages and scenarios.

## 3. Dataset

Figure 2 showcases representative samples that highlight the linguistic and structural diversity of MORE. To achieve this level of coverage, we implemented a robust construction pipeline comprising large-scale PDF acquisition, stratified sampling, and a rigorous annotation process.

---

[2]https://mistral.ai/news/mistral-ocr

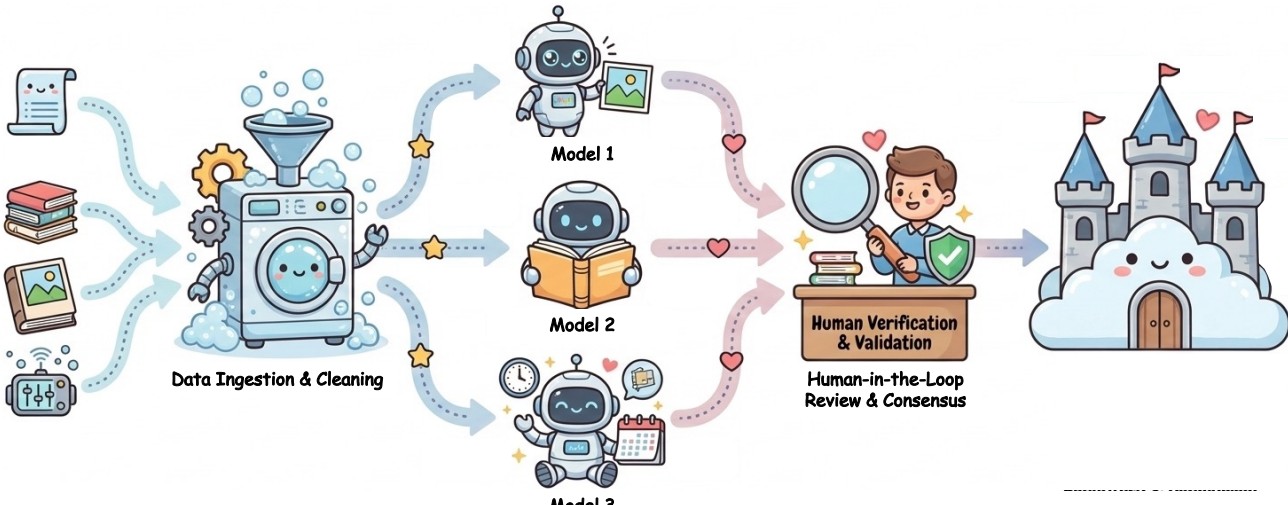

*Figure 3.* Illustration of the model-assisted, human-refined annotation pipeline. To guarantee data reliability, we employ a multi-stage approach: raw data is first processed by diverse models for pre-annotation, followed by rigorous human verification and validation to correct errors and resolve ambiguities.

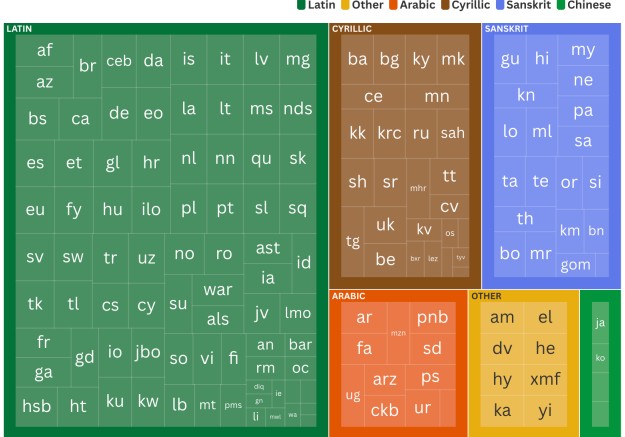

*Figure 4.* Treemap visualizing MORE's balanced language distribution across six script families via stratified sampling.

### 3.1. PDF Collection and Stratified Sampling

To ensure real-world diversity, our data collection process commenced with a robust web-crawling methodology inspired by CCpdf (Turski et al., 2023), harvesting an extensive pool of over 20 million PDFs from diverse web sources. To guarantee data quality, we first filtered out low-quality documents using heuristic spam detection (e.g., excessive repetitive links, broken encodings) and layout density checks, ensuring the retained pages contained rich structural elements rather than merely plain text. Subsequently, we categorized the remaining documents using language classification models (Joulin et al., 2016). In order to prioritize under-represented languages, we excluded Chinese, English, and unlabeled data, which narrowed the candidate pool to approximately 5.7 million documents.

*Table 2.* Language distribution by major language families.

| SCRIPT | COUNT | RATIO |
|---|---|---|
| LATIN | 80 | 53.69% |
| CYRILLIC | 26 | 17.45% |
| SANSKRIT | 20 | 13.42% |
| ARABIC | 11 | 7.38% |
| CHINESE | 4 | 2.68% |
| OTHER | 8 | 5.37% |

Finally, to achieve linguistic balance, we implemented a *stratified sampling strategy* by selecting up to 10 PDFs per language and extracting a single random page from each file. This pipeline yielded a curated dataset of 1,237 pages.

This strategy mitigates web corpus imbalance, ensuring unbiased evaluation. Crucially, it highlights a modality distinction: unlike Machine Translation parallel corpora such as Flores-101 (Goyal et al., 2022) that enforce dense 1D alignments, real documents exhibit inherent structural sparsity. Complex 2D elements naturally vary in frequency across long-tail scripts. Preserving this authentic distribution ensures MORE evaluates on realistic visual layouts.

As illustrated in Figure 4, our sampling strategy effectively mitigates the long-tail distribution often found in web-crawled data, ensuring sufficient representation for low-resource languages. In total, the MORE dataset covers 149 languages spanning six diverse script families.

Table 2 presents the distribution based on Wikipedia's writing system taxonomy[3]. While the Latin group predominates (53.69%), the dataset retains significant diversity. Cyrillic

---

[3]https://en.wikipedia.org/wiki/Writing_system#External_links

*Table 3.* Comparison of multilingual document benchmarks on statistics and annotation coverage.

| DATASET | IMAGE | LANGUAGE | #ANNOTATION | | | | | | OPEN SOURCE |
|---|---|---|---|---|---|---|---|---|---|
| | | | TEXT | FORMULA | TABLE | CODE | CATALOG | READING ORDER | |
| ICDAR RRC-MLT 2017 | 1800 | 6 | 1800 | - | - | - | - | - | ✓ |
| CC-OCR | 1500 | 10 | 1500 | - | - | - | - | - | ✓ |
| MISTRAL-OCR | - | 11 | - | - | - | - | - | - | |
| XDOCPARSE | - | 126 | - | - | - | - | - | - | |
| OURS | 1288 | **149** | 8221 | 82 | 94 | 73 | 104 | 1072 | ✓ |

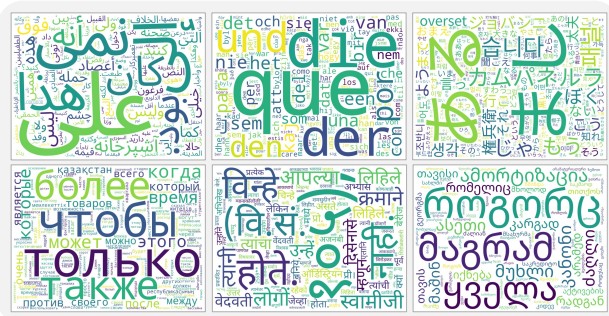

*Figure 5.* Word clouds illustrating the most frequent tokens in the MORE dataset, categorized by six major language families.

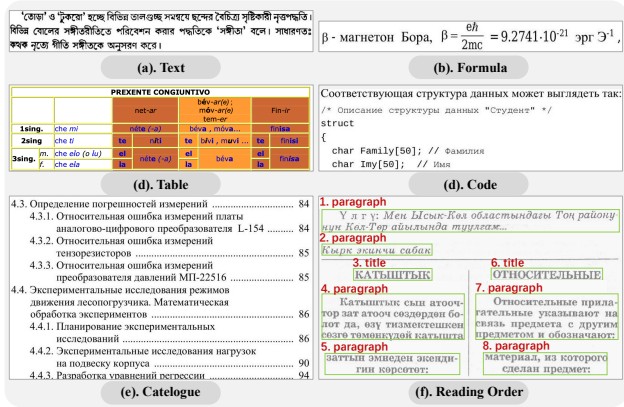

*Figure 6.* Six distinct tasks in MORE.

and Sanskrit together account for over 30%, followed by Arabic, Chinese, and the Other category (5.37%), which encompasses unique scripts like Greek, Hebrew, and Georgian. Collectively, these non-Latin scripts constitute nearly half of the dataset (46.31%), ensuring rigorous evaluation across typologically diverse writing systems.

### 3.2. Expert Annotation

As illustrated in Figure 3, we implement a *model-assisted, human-refined* pipeline to maximize efficiency and reliability. This workflow leverages an ensemble of models to generate anonymous candidates (Li et al., 2025b; Cui et al., 2025; Team et al., 2025; Bai et al., 2025b;a), which are strictly refined by human experts. The pipeline comprises two consecutive phases: *Page-wise Structure Annotation* (focusing on layout and order) and *Element-wise Content Annotation* (transcribing actual content).

**Page-wise Structure Annotation**: To establish a solid foundation for content extraction, this phase integrates layout annotation with reading order annotation. We initially employed D-FINE (Peng et al., 2024) (trained on 60,000 samples) to coarsely locate elements, followed by rigorous manual refinement to rectify localization errors. Building upon this verified layout, we explicitly annotate element dependencies via directional links, ensuring the resulting sequence strictly adheres to the natural reading flow.

Based on these verified layouts, we cropped the document

patches and filtered out irrelevant elements (e.g., figures and barcodes) to isolate target regions for content annotation. Figure 5 visualizes high-frequency tokens to provide a preliminary content overview.

**Element-wise Content Annotation**: For each element, we aggregated predictions from an InternVL-1B model (fine-tuned on a dataset of 250k samples spanning 149 languages) and open-source models (dots.ocr, PaddleOCR-VL, Qwen-VL series), consolidating outputs into an anonymized Markdown format. Human experts reviewed these candidates, **adopting exact matches or manually refining text** while discarding ambiguous cases to ensure reliability.

Specifically, *Text* and *Code* (Markdown) were derived from InternVL-1B and dots.ocr, resulting in 8,221 (from 10,403) and 73 (from 151) retained samples, respectively. *Formulas* were annotated in LaTeX using InternVL-1B and Qwen3-VL-2B (82 selected from 91). Finally, *Tables* (HTML with complex spans) and *Catalogs* were generated via HunyuanOCR and PaddleOCR-VL, yielding 94 and 104 finalized samples after filtering. We summarize the statistics in Table 3, with language-specific details in Appendix B.1.

## 4. Tasks and Evaluation Metrics

Building upon the OmniDocBench protocol, we extended the methodology to cover the six distinct tasks illustrated

*Table 4.* Overall performance evaluation on 149 languages. Main values represent task-wise averages, while subscript values denote page-wise averages. We provide detailed results in Appendix B.2.1.

| MODEL | SIZE | OVERALL | TEXT | FORMULA | TABLE | CODE | CATALOG | READ ORDER |
|---|---|---|---|---|---|---|---|---|
| HUNYUANOCR | 1B | $\mathbf{92.42}_{92.09}$ | $93.81_{91.72}$ | $\mathbf{93.28}_{93.28}$ | $\underline{78.56}_{78.87}$ | $\mathbf{97.07}_{97.08}$ | $\mathbf{95.36}_{95.14}$ | $\underline{96.45}_{96.45}$ |
| GEMINI3 | - | $\underline{91.61}_{91.55}$ | $\mathbf{95.39}_{92.55}$ | $90.27_{90.27}$ | $\mathbf{81.02}_{81.99}$ | $93.05_{93.13}$ | $94.31_{95.75}$ | $95.63_{95.63}$ |
| PADDLEOCR-VL | 0.9B | $87.96_{87.06}$ | $90.99_{88.10}$ | $\underline{91.11}_{90.59}$ | $61.11_{57.15}$ | $\underline{96.29}_{96.59}$ | $93.04_{94.74}$ | $95.19_{95.19}$ |
| dots.ocr | 3B | $84.31_{82.15}$ | $\underline{94.45}_{92.33}$ | $90.77_{91.50}$ | $39.81_{35.62}$ | $95.38_{95.38}$ | $88.26_{80.90}$ | $\mathbf{97.18}_{97.18}$ |
| QWEN2.5-VL | 3B | $83.93_{84.24}$ | $89.36_{86.98}$ | $84.48_{86.34}$ | $68.27_{68.22}$ | $86.69_{87.37}$ | $92.54_{94.31}$ | $82.23_{82.23}$ |
| QWEN3-VL | 2B | $83.56_{82.40}$ | $92.02_{87.89}$ | $65.45_{60.17}$ | $65.21_{66.66}$ | $92.38_{92.43}$ | $93.76_{94.69}$ | $92.53_{92.53}$ |
| DEEPSEEKOCR | 3B-A570M | $82.91_{82.59}$ | $85.27_{84.15}$ | $75.67_{75.71}$ | $61.63_{60.19}$ | $92.26_{92.53}$ | $88.26_{88.62}$ | $94.36_{94.36}$ |
| MINERU2.5 | 1.2B | $48.85_{51.84}$ | $27.12_{40.93}$ | $73.29_{74.99}$ | $33.83_{33.83}$ | $72.41_{76.35}$ | $21.61_{20.13}$ | $64.81_{64.81}$ |

in Figure 6, specifically incorporating code and catalog evaluation. Additionally, we employ a *decoupled approach* for overall scoring. Detailed metrics are defined as follows:

**Sequential Elements**: For linear sequences (Text, Code, Catalog, and Reading Order), we employ Normalized Edit Distance (Levenshtein, 1966). Task-specific normalizations (e.g., indentation for Code) are applied to the prediction ($P$) and ground truth ($G$) prior to calculation:

$$\text{NED} = 1 - \frac{1}{N} \sum_{i=1}^{N} \frac{\text{EditDist}(P_i, G_i)}{\max(|P_i|, |G_i|)} \quad (1)$$

**Formula**: We utilize the Character Detection Matching metric (Wang et al., 2025). By rendering LaTeX into images for spatial matching, the score is derived from the counts of matched ($TP$) and unmatched ($FP, FN$) bounding boxes:

$$\text{CDM} = \frac{1}{N} \sum_{i=1}^{N} \frac{2 \cdot TP_i}{2 \cdot TP_i + FP_i + FN_i} \quad (2)$$

**Table**: We employ TEDS (Zhong et al., 2020) to assess HTML integrity. The metric computes the edit distance between the predicted tree $T_{p,i}$ and ground truth $T_{g,i}$, normalized by the larger tree's node count:

$$\text{TEDS} = 1 - \frac{1}{N} \sum_{i=1}^{N} \frac{\text{TreeEditDist}(T_{p,i}, T_{g,i})}{\max(|T_{p,i}|, |T_{g,i}|)} \quad (3)$$

**Overall Score**: In contrast to OmniDocBench, we calculate the final score as *the arithmetic mean of the six distinct tasks*. This strategy intentionally decouples task dependencies, thereby eliminating error propagation across different evaluation stages. See Appendix A for detailed definitions.

## 5. Analysis

In this section, we evaluate representative models: General-purpose VLMs (e.g., Qwen2.5-VL, Qwen3-VL), which are designed for broad multimodal scenarios yet possess

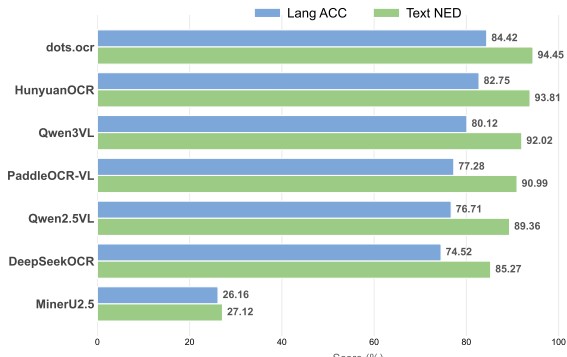

*Figure 7.* Language confusion analysis on text.

inherent document parsing capabilities; and Specialized VLMs (e.g., dots.ocr, PaddleOCR-VL, DeepSeek-OCR, HunyuanOCR, MinerU2.5), which have been specifically fine-tuned for document parsing tasks.

### 5.1. Overall Evaluation

**State-of-the-Art Performance**: As presented in Table 4, HunyuanOCR dominates with an overall score of **92.42**, surpassing the runner-up PaddleOCR-VL (87.96) by a significant margin. It ranks first in four out of six sub-metrics (Formula, Table, Code, and Catalog), establishing a new benchmark for specialized OCR models. While general VLMs like the Qwen series provide competitive baselines (between 83–84), they still lag behind the top-tier specialized model, particularly in structural parsing. Furthermore, we provide detailed metrics by language in Appendix B.2.1.

**The Text-Structure Dichotomy**: We observe a trade-off between sequential recognition and 2D spatial reasoning. dots.ocr excels as a *pure reader* (Text: 94.45, Order: 97.18) yet collapses on Tables (39.81), favoring 1D flow over complex layouts. In contrast, HunyuanOCR effectively balances semantic accuracy with structural integrity.

**The Bottleneck of Table Recognition**: Table parsing remains the primary bottleneck. Unlike Text and Code (>90), table recognition lags significantly. HunyuanOCR alone

*Table 5.* Overall analysis on six major scripts.

| MODEL | OVERALL | LAT. | CYR. | SAN. | ARA. | CHI. | OTH. |
|---|---|---|---|---|---|---|---|
| HUNYUANOCR | **90.17** | **95.45** | **91.45** | **71.94** | **84.72** | **94.68** | 81.95 |
| DOTS.OCR | 86.52 | 91.46 | 89.58 | 64.92 | 81.25 | 91.78 | **84.08** |
| PADDLEOCR-VL | 85.53 | 93.02 | 89.13 | 63.27 | 77.26 | 88.97 | 60.06 |
| QWEN3-VL-2B | 85.46 | 91.44 | 88.61 | 70.69 | 79.98 | 80.36 | 58.69 |
| DEEPSEEKOCR | 83.47 | 90.38 | 85.71 | 58.66 | 75.65 | 89.87 | 74.19 |
| QWEN2.5-VL-3B | 79.50 | 87.58 | 80.82 | 56.48 | 74.67 | 81.31 | 53.72 |
| MINERU2.5 | 40.47 | 52.97 | 27.21 | 24.15 | 24.94 | 44.65 | 16.91 |

*Note: Lat.: Latin, Cyr.: Cyrillic, San.: Sanskrit, Ara.: Arabic, Chi.: Chinese, Oth.: Other.*

*Table 6.* Text recognition across six major scripts, with detailed results in Appendix B.2.2.

| MODEL | AVG | LAT. | CYR. | SAN. | ARA. | CHI. | OTH. |
|---|---|---|---|---|---|---|---|
| HUNYUANOCR | **89.32** | 97.51 | **90.01** | **63.10** | 80.17 | **94.18** | 70.81 |
| DOTS.OCR | 89.06 | **97.83** | 89.65 | 61.78 | 76.23 | 92.71 | **73.57** |
| QWEN3-VL-2B | 85.84 | 95.91 | 89.00 | 55.48 | 78.94 | 89.70 | 44.29 |
| PADDLEOCR-VL | 85.48 | 95.69 | 88.94 | 56.50 | **80.45** | 88.22 | 35.57 |
| QWEN2.5-VL-3B | 84.27 | 94.74 | 85.05 | 52.68 | 75.92 | 85.13 | 53.99 |
| DEEPSEEKOCR | 80.12 | 90.19 | 82.56 | 50.52 | 59.05 | 87.07 | 59.80 |
| MINERU2.5 | 23.41 | 38.20 | 4.52 | 2.22 | 9.44 | 22.78 | 2.49 |

*Note: Lat.: Latin, Cyr.: Cyrillic, San.: Sanskrit, Ara.: Arabic, Chi.: Chinese, Oth.: Other.*

*Table 7.* Formula recognition on top 15 languages.

| LANG. | HUNYUAN OCR | DOTS.OCR | PADDLE OCR-VL | QWEN2.5 VL-3B | DEEPSEEK OCR | MINERU 2.5 | QWEN3 VL-2B |
|---|---|---|---|---|---|---|---|
| AR | 94.50 | 93.99 | **97.44** | 95.60 | 81.80 | 70.54 | 40.66 |
| BS | 90.87 | 77.63 | **92.80** | 84.43 | 81.60 | 59.77 | 87.93 |
| CA | 93.31 | 91.67 | **94.34** | 85.83 | 72.84 | 66.81 | 77.40 |
| CY | 72.70 | 57.10 | 24.20 | 48.50 | **100.00** | 0.00 | 0.00 |
| ES | **100.00** | 89.72 | 98.80 | 97.52 | 51.02 | 52.72 | 40.00 |
| EU | 96.20 | 96.20 | 96.20 | 90.20 | 96.20 | 0.00 | 96.20 |
| HY | 95.40 | 94.02 | **96.34** | 93.11 | 75.09 | 94.58 | 62.46 |
| KO | **96.13** | 93.11 | 85.48 | 87.06 | 90.02 | 84.40 | 15.57 |
| KY | 87.87 | 92.86 | 91.04 | 84.36 | 82.93 | 46.71 | 63.71 |
| NN | 94.13 | 84.50 | **95.00** | 89.20 | 88.97 | 56.20 | 91.77 |
| PNB | 69.80 | 83.00 | 80.90 | 81.80 | 0.00 | **100.00** | 81.80 |
| SA | 89.70 | 96.60 | 96.60 | 65.03 | 96.60 | 32.37 | 0.00 |
| SL | 93.69 | 93.21 | 85.09 | 67.85 | 74.66 | **94.94** | 90.54 |
| TT | 90.33 | 89.91 | 90.93 | 85.49 | 73.50 | **93.21** | 86.21 |
| VI | **100.00** | **100.00** | **100.00** | **100.00** | 0.00 | 99.50 | **100.00** |
| AVG | **90.98** | 88.90 | 88.34 | 83.73 | 71.02 | 63.45 | 62.28 |

*Note: ar: Arabic, bs: Bosnian, ca: Catalan, cy: Welsh, es: Spanish, eu: Basque, hy: Armenian, ko: Korean, ky: Kyrgyz, nn: Nynorsk, pnb: W. Punjabi, sa: Sanskrit, sl: Slovenian, tt: Tatar, vi: Vietnamese.*

maintains robustness (**78.56**), outperforming Qwen2.5-VL (68.27) and DeepSeekOCR (low 60s). This confirms that while semantic extraction is mature, parsing non-linear structures remains the critical frontier.

**Language Confusion Analysis**: To evaluate multilingual robustness, we measured language identification consistency on 8,221 samples across 149 languages using `fast-langdetect`[4]. Figure 7 reveals a strong positive correlation between recognition quality and identification accuracy, with text NED consistently outscoring language accuracy. dots.ocr achieves top performance (84.42% ACC, 94.45 NED), whereas MinerU2.5 collapses (26.16% ACC, 27.12 NED). This confirms that precise character recognition is a prerequisite for accurate language detection.

**Robustness Across Aggregation Methods**: Table 4 reports task-wise (main) and page-wise (subscript) averages. HunyuanOCR and Gemini 3 show minimal variance, adapting well to extreme element density or sparsity, whereas skewed models drop under natural distributions.

### 5.2. Analysis by Script

**Overall Dominance and Robustness**: As shown in Table 5, HunyuanOCR achieves state-of-the-art performance (Overall 90.17), dominating 5 out of 6 categories with near-saturated accuracy on high-resource scripts. However, dots.ocr (Overall 86.52) exhibits superior generalization on isolated languages, topping the "Other" category with 84.08. In contrast, Qwen3-VL-2B and PaddleOCR-VL drop below 60% in this metric, exposing significant limitations in generalizing to long-tail scripts.

**Complex Script Handling**: Performance on Sanskrit highlights the necessity of sufficient parameter scale. Qwen3-VL-2B ranks second (70.69), significantly outperforming PaddleOCR-VL (63.27). This suggests that larger model capacity is essential for decoding intricate script structures where smaller baselines struggle. Conversely, the performance collapse observed in lightweight models like MinerU2.5 (24.15) confirms that limited parameter size remains a primary bottleneck for complex text recognition.

---

[4]https://github.com/LlmKira/fast-langdetect

### 5.3. Element-wise Evaluation

**Evaluation of Text Recognition**: Table 6 presents performance across diverse scripts. HunyuanOCR (89.32) and dots.ocr (89.06) lead with comparable scores. While dots.ocr tops Latin (97.83) and Other scripts, HunyuanOCR excels in Chinese and Sanskrit. Notably, PaddleOCR-VL achieves the highest accuracy in Arabic (80.45), surpassing larger VLMs and highlighting domain-specific strengths. Overall, performance varies by complexity: Latin and Chinese reach near-saturation (>94%), whereas Sanskrit remains the most challenging (max 63.10). Furthermore, we also include granular results by language in Appendix B.2.2.

**Evaluation of Formula Recognition**: Table 7 details performance across top 15 languages. HunyuanOCR shows the best consistency, achieving the highest average 90.98. While ranking third on average (88.34), PaddleOCR-VL secures the most first places (7 categories, including Arabic and Bosnian), indicating specialized mastery. In contrast, DeepSeekOCR and MinerU2.5 show significant volatility. For instance, DeepSeekOCR achieves perfect accuracy on Welsh (100.00) yet fails on Vietnamese (0.00), where five other models reached saturation. Similarly, MinerU2.5 leads in Western Punjabi despite its lower overall rank.

**Evaluation of Table Recognition**: As presented in Table 8,

*Table 8.* Table recognition on top 15 languages, with detailed results in Appendix B.2.3.

| LANG. | HUNYUAN OCR | QWEN2.5 VL-3B | QWEN3 VL-2B | PADDLE OCR-VL | DEEPSEEK OCR | DOTS .OCR | MINERU 2.5 |
|---|---|---|---|---|---|---|---|
| WAR | **66.82** | 61.24 | 34.89 | 22.99 | 19.18 | 0.00 | 9.53 |
| LB | 80.37 | **83.41** | 62.74 | 78.40 | 76.84 | 44.29 | 57.00 |
| CV | 74.93 | 56.83 | 58.20 | **77.34** | 70.84 | 46.75 | 35.61 |
| AN | 97.31 | 86.63 | 86.07 | **98.15** | 91.91 | 94.92 | 97.03 |
| SU | **99.98** | 98.47 | 99.49 | 79.51 | 79.98 | 79.84 | 0.00 |
| EU | 91.34 | 57.84 | 73.30 | 72.64 | **93.95** | 43.98 | 83.87 |
| BS | 83.30 | 67.42 | 83.77 | 91.29 | 91.04 | **93.97** | 90.00 |
| CE | **89.98** | 77.52 | 60.59 | 62.50 | 64.21 | 63.60 | 0.00 |
| IT | 96.67 | **98.64** | 79.55 | 33.28 | 33.13 | 33.27 | 33.25 |
| MT | 87.36 | 77.10 | 72.65 | 93.15 | **96.04** | 64.92 | 23.88 |
| OC | 74.78 | 85.09 | **85.35** | 30.14 | 65.05 | 65.88 | 33.13 |
| PA | **41.68** | 36.28 | 36.34 | 0.00 | 0.00 | 0.00 | 2.88 |
| BR | 96.56 | 95.76 | **98.12** | 97.79 | 97.77 | 0.00 | 0.00 |
| CY | 99.72 | 96.90 | 74.55 | 99.72 | **100.00** | 0.00 | 0.00 |
| GA | 74.17 | 70.00 | 66.52 | 72.48 | 0.00 | 0.00 | 0.00 |
| AVG | **83.66** | 76.61 | 71.48 | 67.29 | 65.33 | 42.09 | 31.08 |

*Note: war: Waray, lb: Luxembourgish, cv: Chuvash, an: Aragonese, su: Sundanese, eu: Basque, bs: Bosnian, ce: Chechen, it: Italian, mt: Maltese, oc: Occitan, pa: Punjabi, br: Breton, cy: Welsh, ga: Irish.*

*Table 9.* Code recognition on all languages.

| LANG. | HUNYUAN OCR | PADDLE OCR-VL | DOTS .OCR | QWEN3 VL-2B | DEEPSEEK OCR | QWEN2.5 VL-3B | MINERU 2.5 |
|---|---|---|---|---|---|---|---|
| RU | 97.48 | **97.91** | 93.97 | 89.34 | 91.01 | 89.74 | 57.50 |
| PL | 97.51 | **98.86** | 96.71 | 94.02 | 93.76 | 79.29 | 77.50 |
| ES | **95.95** | 74.06 | 95.56 | 92.48 | 86.49 | 82.93 | 53.30 |
| PT | 98.80 | **99.50** | 91.70 | 91.43 | 95.62 | 88.05 | 92.78 |
| ID | 93.46 | 98.68 | 96.83 | 96.53 | **99.82** | 93.81 | 93.67 |
| DE | 93.55 | **99.19** | 98.30 | 93.98 | 75.20 | 72.30 | 97.92 |
| FR | 98.66 | **99.68** | 99.58 | 96.14 | 98.99 | 96.62 | 97.54 |
| IT | 99.48 | 99.69 | 99.27 | 95.31 | **99.79** | 95.67 | 48.22 |
| JA | 98.99 | **99.49** | 94.62 | 97.57 | 98.70 | 98.01 | 92.53 |
| AVG | **97.10** | 96.34 | 96.28 | 94.09 | 93.25 | 88.49 | 79.00 |

*Note: ru: Russian, pl: Polish, es: Spanish, pt: Portuguese, id: Indonesian, de: German, fr: French, it: Italian, ja: Japanese.*

HunyuanOCR establishes a clear lead with an average score of 83.66, demonstrating robust generalization across diverse scripts. Qwen2.5-VL-3B follows as the runner-up (76.61), outperforming other VLM baselines. While PaddleOCR-VL and DeepSeekOCR rank lower on average, they exhibit "spiky" performance profiles: Paddle excels in Chuvash (77.34) and DeepSeek achieves perfection in Welsh (100.00), yet both suffer from severe drops in other languages. In contrast, dots.ocr and MinerU2.5 struggle with structural generalization, evidenced by complete failures (0.00 scores) in languages like Breton and Irish. Linguistically, while Sundanese appears largely solved (>99%), Punjabi remains a universal bottleneck, with even the best model (HunyuanOCR) capping at 41.68, highlighting the challenge of parsing complex script tables. Additionally, Appendix B.2.3 details the breakdowns for every language.

**Evaluation of Code Recognition**: As shown in Table 9, performance is highly saturated, with the top three models exceeding 96% average accuracy. HunyuanOCR leads (97.10) by maintaining stability across all languages. In contrast, PaddleOCR-VL ranks second (96.34); despite dominating 6 categories (e.g., French, Japanese), it is penalized by a significant drop in Spanish (74.06). dots.ocr closely follows (96.28), demonstrating robust generaliza-

*Table 10.* Catalog recognition on all languages.

| LANG. | HUNYUAN OCR | QWEN3 VL-2B | DEEPSEEK OCR | QWEN2.5 VL-3B | PADDLE OCR-VL | DOTS .OCR | MINERU 2.5 |
|---|---|---|---|---|---|---|---|
| RU | **98.72** | 96.03 | 86.58 | 94.25 | 95.46 | 85.61 | 3.45 |
| FR | 83.92 | **89.53** | 81.54 | 87.33 | 88.40 | 69.63 | 45.69 |
| ES | **90.68** | 89.60 | 84.08 | 89.77 | 89.58 | 79.20 | 46.07 |
| DE | **99.00** | 97.85 | 96.23 | 98.08 | 96.52 | 79.87 | 54.71 |
| UK | **100.00** | 98.51 | 99.44 | 96.95 | 97.91 | 82.99 | 2.20 |
| JA | **76.76** | 59.05 | 73.43 | 53.89 | 48.00 | 50.50 | 1.07 |
| PT | **99.83** | 99.31 | 99.58 | 99.58 | 99.61 | 91.77 | 0.00 |
| TR | **99.95** | 96.83 | 94.76 | 98.26 | 98.40 | 99.59 | 54.77 |
| EL | **98.28** | 94.21 | 96.61 | 95.13 | 97.14 | 82.86 | 14.23 |
| PL | **100.00** | 97.13 | 99.37 | 98.01 | 98.84 | 99.37 | 0.00 |
| RO | **100.00** | 97.36 | 98.52 | 97.29 | 97.51 | 99.14 | 94.06 |
| ID | 99.67 | 99.75 | 99.02 | **99.75** | 99.75 | 85.85 | 0.00 |
| AVG | **95.57** | 92.93 | 92.43 | 92.36 | 92.26 | 83.87 | 26.35 |

*Note: ru: Russian, fr: French, es: Spanish, de: German, uk: Ukrainian, ja: Japanese, pt: Portuguese, tr: Turkish, el: Greek, pl: Polish, ro: Romanian, id: Indonesian.*

*Table 11.* Reading order evaluation grouped by script, with detailed results in Appendix B.2.4.

| MODEL | AVG | LAT. | CYR. | SAN. | ARA. | CHI. | OTH. |
|---|---|---|---|---|---|---|---|
| DOTS.OCR | **95.75** | **98.30** | **96.40** | 83.90 | 94.63 | **96.67** | **94.90** |
| HUNYUANOCR | 95.11 | 97.33 | 96.30 | 88.52 | 89.44 | **96.67** | 90.11 |
| PADDLEOCR-VL | 92.85 | 96.63 | 95.94 | 84.70 | 76.81 | **96.67** | 81.82 |
| DEEPSEEKOCR | 92.61 | 96.39 | 92.36 | 79.01 | **99.63** | 93.75 | 76.18 |
| QWEN3-VL-2B | 91.53 | 93.68 | 93.25 | **89.90** | 88.49 | 88.34 | 74.08 |
| QWEN2.5-VL-3B | 78.37 | 85.39 | 79.66 | 60.36 | 80.65 | 79.54 | 40.68 |
| MINERU2.5 | 58.79 | 70.13 | 52.32 | 44.90 | 39.12 | 60.36 | 21.82 |

*Note: Lat.: Latin, Cyr.: Cyrillic, San.: Sanskrit, Ara.: Arabic, Chi.: Chinese, Oth.: Other.*

tion by frequently securing the second-best scores (e.g., in Spanish and German) without suffering severe drops. Notably, DeepSeekOCR also demonstrates strong capabilities, securing top scores in Indonesian and Italian.

**Evaluation of Catalog Recognition**: In Table 10, HunyuanOCR dominates with an average score of 95.57, topping 10 of 12 categories with perfect accuracy in Ukrainian, Polish, and Romanian. Notably, the smaller Qwen3-VL-2B (92.93) secures second place, outperforming larger models like Qwen2.5-VL-3B. Japanese remains the primary bottleneck; while HunyuanOCR maintains 76.76, most other models drop below 60%. Conversely, languages like Indonesian and Portuguese appear saturated with near-perfect scores. dots.ocr is compromised by layout ambiguity, frequently misclassifying catalogs as tables (see Appendix C). In contrast, MinerU2.5 (avg. 26.35) struggles with complex layouts, scoring zero in three languages.

**Evaluation of Reading Order**: Table 11 presents the evaluation results for reading order prediction. dots.ocr and HunyuanOCR demonstrate the most robust and sophisticated spatial layout understanding, achieving top average scores of 95.75 and 95.11, respectively. While performance converges on standard layouts, evidenced by the top three models all reaching 96.67 on Chinese, significant gaps emerge in scripts with complex directionality. Notably, DeepSeekOCR achieves near-perfect performance (99.63) on Arabic, suggesting superior capability in han-

*Table 12.* Layout-dependent performance evaluation. Scores reflect end-to-end parsing capabilities, where layout and reading order alignment become the primary factors determining the final metric.

| MODEL | SIZE | OVERALL | TEXT | FORMULA | TABLE | CODE | CATALOG | READ ORDER |
|-------|------|---------|------|---------|-------|------|---------|------------|
| DOTS.OCR | 3B | **80.68** | **88.46** | 67.29 | 77.57 | 95.38 | 88.26 | **97.18** |
| GEMINI 3 | - | 79.14 | 87.44 | **69.16** | 72.64 | 93.05 | 94.31 | 95.63 |
| DEEPSEEKOCR | 3B-A570M | 76.96 | 86.64 | 58.21 | **78.32** | 92.26 | 88.26 | 94.36 |
| PADDLEOCR-VL | 0.9B | 76.40 | 78.86 | 60.75 | 73.25 | 96.29 | 93.04 | 95.19 |
| HUNYUANOCR | 1B | 76.08 | 84.84 | 59.09 | 72.71 | **97.07** | **95.36** | 96.45 |
| QWEN3-VL-2B | 2B | 72.68 | 80.34 | 58.39 | 67.11 | 92.38 | 93.76 | 92.53 |
| MINERU2.5 | 1.2B | 63.61 | 40.20 | 50.91 | 75.11 | 72.41 | 21.61 | 64.80 |
| QWEN2.5-VL-3B | 3B | 62.97 | 69.68 | 50.70 | 55.97 | 86.69 | 92.54 | 82.23 |

dling intricate script-specific spatial structures compared to PaddleOCR-VL (76.81), which struggles to resolve the correct reading path in visually complex scripts. Interestingly, the smaller Qwen3-VL-2B secures the highest score in Sanskrit (89.90), outperforming the overall leaders. Ultimately, dots.ocr secures the first place primarily due to its exceptional generalization in the Other category (94.90), whereas models like Qwen2.5-VL suffer severe degradation (40.68) outside of common distributions. Besides, comprehensive results for each language are listed in Appendix B.2.4.

### 5.4. Layout-Dependent Evaluation

To rigorously assess the models' end-to-end capabilities in real-world scenarios, we extend our analysis beyond the decoupled metrics. In previous sections, content recognition was evaluated on pre-cropped patches with human-verified layouts. While this effectively isolates character-level recognition capabilities, it abstracts away the inherent challenge of layout detection. To bridge this gap and address the modality constraints of real-world document parsing, we introduce a *layout-dependent evaluation* following the `quick_match` protocol from OmniDocBench (Ouyang et al., 2025). In this setting, layout misclassifications and reading order alignment become the primary factors determining the final content score.

As presented in Table 12, enforcing layout constraints shatters any illusion of a performance ceiling, resulting in a significant drop across all models. For instance, HunyuanOCR's overall score decreases from 92.42 (decoupled) to 76.08. Task-level metrics reveal a fragmented landscape: while dots.ocr excels in Text (88.46), DeepSeekOCR leads in Tables (78.32), and HunyuanOCR dominates Code (97.07). This confirms complex layout detection remains a primary bottleneck. Despite these drops, the relative hierarchy of model capabilities remains robust, validating our decoupled findings and proving the benchmark provides ample headroom for future advancements.

Notably, specialized models demonstrate exceptional resilience in this end-to-end setting. dots.ocr emerges as the leader with an overall score of 80.68, outperforming the much larger Gemini 3 (79.14). While Gemini 3 exhibits strong general recognition capabilities (leading in Formula with 69.16), its performance on highly structured elements like Tables (72.64) indicates that simply scaling up parameters is not a silver bullet for complex document layouts. This reinforces the necessity of specialized, structure-aware architectures for global document intelligence.

## 6. Conclusion

In this work, we present **MORE**, a comprehensive benchmark covering 149 languages and complex document elements to bridge the evaluation gap in multilingual parsing. Our extensive experiments reveal that while state-of-the-art VLMs have achieved impressive proficiency in text recognition for high-resource languages, significant bottlenecks remain in **structural understanding** (e.g., tables) and adaptability to **long-tail scripts**. By providing a rigorous testbed that exposes these specific limitations, MORE serves as a critical diagnostic tool, urging the community to move beyond simple character recognition toward developing truly robust, structure-aware, and universally inclusive document understanding systems. In future iterations, we plan to further expand the coverage of underrepresented structural elements across long-tail languages to provide an even more granular and balanced evaluation.

## Impact Statement

This work addresses the critical *knowledge blind spot* in modern AI by establishing a quantitative baseline for document parsing across 149 languages. We acknowledge that the sampling of complex structural elements remains sparse for rare languages. In future iterations, we are committed to expanding the coverage of these underrepresented elements and developing more objective, multi-dimensional evaluation protocols. Furthermore, as high-precision parsing still faces reliability bottlenecks in critical domains, practitioners must strictly implement safeguards against privacy breaches and hallucinations when deploying these technologies in real-world applications.

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

In this supplementary material, we present a comprehensive breakdown of our methodology and additional experimental results. Specifically, Section A elucidates the definitions and formulations of the metrics employed. Section B details the composition of our benchmark datasets, followed by extensive reports on monolingual performance and fine-grained analysis of individual elements. Lastly, Section C provides qualitative visualizations of model outputs to offer further insights.

## A. Formal Definitions of Metrics

In this section, we provide the formulations for the evaluation metrics introduced in Section 4. To ensure consistency across all definitions, let $N$ denote the total number of samples in the evaluation set, and the subscript $i$ denote the $i$-th sample.

### A.1. Sequential Elements (NED)

For linear text sequences, including *Text Recognition*, *Code Recognition*, *Catalog Recognition*, and *Reading Order Prediction*, we employ Normalized Edit Distance (NED) to measure accuracy.

Let $P_i$ and $G_i$ represent the predicted text sequence and the ground truth sequence for the $i$-th sample, respectively. The metric is defined as:

$$\text{NED} = 1 - \frac{1}{N} \sum_{i=1}^{N} \frac{\text{EditDist}(P_i, G_i)}{\max(|P_i|, |G_i|)} \tag{4}$$

where $\text{EditDist}(\cdot, \cdot)$ denotes the Levenshtein distance, and $\text{len}(\cdot)$ represents the string length. A higher NED score indicates better performance. Note that for Code and Catalog tasks, task-specific normalizations (e.g., indentation standardization) are applied to $P_i$ and $G_i$ prior to calculation.

### A.2. Structured Elements (TEDS)

For *Table Recognition*, we evaluate both structural and content accuracy using Tree Edit Distance-based Similarity (TEDS) based on HTML format.

Let $T_{p,i}$ and $T_{g,i}$ denote the HTML trees of the prediction and ground truth for the $i$-th sample, respectively. The TEDS score is calculated as:

$$\text{TEDS} = 1 - \frac{1}{N} \sum_{i=1}^{N} \frac{\text{TreeEditDist}(T_{p,i}, T_{g,i})}{\max(|T_{p,i}|, |T_{g,i}|)} \tag{5}$$

where $\text{TreeEditDist}(\cdot, \cdot)$ represents the minimum number of node edit operations (insertion, deletion, substitution) required to transform $T_{p,i}$ into $T_{g,i}$, and $|T|$ denotes the number of nodes in the tree.

### A.3. Mathematical Expressions (CDM)

For *Formula Recognition*, we utilize Character Detection Matching (CDM) metric (Wang et al., 2025) based on LaTeX.

Following (Wang et al., 2025), we render LaTeX pairs $(P_i, G_i)$ into images to perform spatial character matching, calculating the score as:

$$\text{CDM} = \frac{1}{N} \sum_{i=1}^{N} \frac{2 \cdot TP_i}{2 \cdot TP_i + FP_i + FN_i} \tag{6}$$

where $TP_i$ denotes the number of matched bounding box pairs, while $FP_i$ and $FN_i$ represent the count of unmatched bounding boxes in the prediction and ground truth, respectively.

## A.4. Overall Score Calculation

To provide a holistic assessment of model capability, the **Overall Score** is calculated as the arithmetic mean of these distinct task scores. This decoupled formulation prevents performance in one dominant modality from overshadowing others:

$$\text{Overall} = \frac{1}{6}\left(\text{TEDS} + \text{CDM} + \sum_{k \in \mathcal{S}} \text{NED}_k\right) \tag{7}$$

where TEDS and CDM correspond to Table and Formula recognition scores, and $\mathcal{S} = \{\text{Text}, \text{Code}, \text{Catalog}, \text{Order}\}$ denotes the set of tasks evaluated using NED.

## B. Benchmark Statistics and Detailed Results

In this section, we first provide a granular breakdown of the benchmark composition, followed by a comprehensive presentation of our fine-grained evaluation results.

### B.1. Benchmark Composition

Tables 13 and 14 detail the benchmark statistics, organized by language family to illustrate linguistic diversity. We further report sample counts for seven document elements, including layout, paragraph, formula, table, catalog, code, and order, offering a granular view of the dataset across all supported languages.

*Table 13.* Benchmark statistics (Part I: af - mg).

| ISO | LANG. | FAMILY | LAY. | PAR. | FOR. | TAB. | COD. | CAT. | ORD. | ISO | LANG. | FAMILY | LAY. | PAR. | FOR. | TAB. | COD. | CAT. | ORD. |
|---|---|---|---|---|---|---|---|---|---|---|---|---|---|---|---|---|---|---|---|
| AF | AFRIKAANS | LATIN | 10 | 86 | 0 | 0 | 0 | 0 | 7 | GL | GALICIAN | LATIN | 10 | 156 | 0 | 1 | 0 | 0 | 7 |
| ALS | TOSK ALB. | LATIN | 8 | 38 | 0 | 0 | 0 | 0 | 5 | GN | GUARANI | LATIN | 2 | 3 | 0 | 0 | 0 | 0 | 1 |
| AM | AMHARIC | OTHER | 10 | 0 | 0 | 0 | 0 | 0 | 1 | GOM | KONKANI | SANSKRIT | 6 | 10 | 0 | 0 | 0 | 0 | 5 |
| AN | ARAGONESE | LATIN | 5 | 23 | 0 | 5 | 0 | 0 | 3 | GU | GUJARATI | SANSKRIT | 10 | 8 | 0 | 0 | 0 | 0 | 3 |
| AR | ARABIC | ARABIC | 10 | 15 | 7 | 0 | 0 | 0 | 4 | GV | MANX | LATIN | 1 | 2 | 0 | 0 | 0 | 0 | 1 |
| ARZ | EGY. ARAB | ARABIC | 9 | 15 | 0 | 0 | 0 | 0 | 3 | HE | HEBREW | OTHER | 10 | 10 | 0 | 0 | 0 | 0 | 4 |
| AST | ASTURIAN | LATIN | 8 | 88 | 0 | 0 | 0 | 0 | 7 | HI | HINDI | SANSKRIT | 10 | 18 | 0 | 0 | 0 | 0 | 3 |
| AV | AVARIC | CYRILLIC | 2 | 14 | 0 | 0 | 0 | 0 | 1 | HR | CROATIAN | LATIN | 10 | 123 | 0 | 0 | 0 | 0 | 10 |
| AZ | AZERBAIJANI | LATIN | 10 | 93 | 0 | 0 | 0 | 0 | 7 | HSB | U. SORBIAN | LATIN | 9 | 37 | 0 | 0 | 0 | 0 | 5 |
| AZB | S. AZER. | ARABIC | 4 | 4 | 0 | 0 | 0 | 0 | 2 | HT | HAITIAN | LATIN | 9 | 31 | 0 | 0 | 0 | 0 | 4 |
| BA | BASHKIR | CYRILLIC | 10 | 105 | 0 | 0 | 0 | 0 | 9 | HU | HUNGARIAN | LATIN | 10 | 131 | 0 | 0 | 0 | 0 | 10 |
| BAR | BAVARIAN | LATIN | 4 | 2 | 0 | 0 | 0 | 0 | 1 | HY | ARMENIAN | OTHER | 10 | 31 | 12 | 0 | 0 | 0 | 6 |
| BE | BELARUSIAN | CYRILLIC | 9 | 34 | 0 | 0 | 0 | 0 | 6 | IA | INTERLINGUA | LATIN | 8 | 24 | 0 | 1 | 0 | 0 | 4 |
| BG | BULGARIAN | CYRILLIC | 10 | 52 | 0 | 0 | 0 | 0 | 7 | ID | INDONESIAN | LATIN | 8 | 77 | 0 | 0 | 6 | 1 | 7 |
| BN | BENGALI | SANSKRIT | 7 | 5 | 0 | 0 | 0 | 0 | 1 | IE | INTERLINGUE | LATIN | 2 | 6 | 0 | 0 | 0 | 0 | 1 |
| BO | TIBETAN | SANSKRIT | 9 | 0 | 0 | 1 | 0 | 0 | 0 | ILO | ILOCANO | LATIN | 10 | 56 | 0 | 2 | 0 | 0 | 6 |
| BR | BRETON | LATIN | 10 | 52 | 0 | 2 | 0 | 0 | 8 | IO | IDO | LATIN | 9 | 57 | 0 | 0 | 0 | 0 | 7 |
| BS | BOSNIAN | LATIN | 10 | 108 | 3 | 3 | 0 | 0 | 8 | IS | ICELANDIC | LATIN | 10 | 79 | 0 | 0 | 0 | 0 | 10 |
| BXR | BURYAT | CYRILLIC | 4 | 22 | 0 | 0 | 0 | 0 | 4 | IT | ITALIAN | LATIN | 10 | 76 | 0 | 3 | 2 | 0 | 7 |
| CA | CATALAN | LATIN | 10 | 61 | 12 | 0 | 0 | 0 | 7 | JA | JAPANESE | CHINESE | 10 | 40 | 0 | 0 | 2 | 3 | 5 |
| CE | CHECHEN | CYRILLIC | 10 | 131 | 0 | 3 | 0 | 0 | 8 | JBO | LOJBAN | LATIN | 9 | 36 | 0 | 2 | 0 | 0 | 4 |
| CEB | CEBUANO | LATIN | 10 | 31 | 0 | 1 | 0 | 0 | 6 | JV | JAVANESE | LATIN | 8 | 81 | 0 | 0 | 0 | 0 | 8 |
| CKB | C. KURD | ARABIC | 9 | 29 | 0 | 0 | 0 | 0 | 1 | KA | GEORGIAN | OTHER | 10 | 54 | 0 | 0 | 0 | 0 | 4 |
| CS | CZECH | LATIN | 9 | 96 | 0 | 0 | 0 | 0 | 9 | KK | KAZAKH | CYRILLIC | 10 | 159 | 0 | 0 | 0 | 0 | 7 |
| CV | CHUVASH | CYRILLIC | 6 | 93 | 0 | 6 | 0 | 0 | 5 | KM | KHMER | SANSKRIT | 8 | 5 | 0 | 2 | 0 | 0 | 3 |
| CY | WELSH | LATIN | 9 | 94 | 1 | 2 | 0 | 0 | 7 | KN | KANNADA | SANSKRIT | 10 | 20 | 0 | 0 | 0 | 0 | 2 |
| DA | DANISH | LATIN | 10 | 60 | 0 | 0 | 0 | 0 | 9 | KO | KOREAN | CHINESE | 7 | 53 | 11 | 0 | 0 | 0 | 7 |
| DE | GERMAN | LATIN | 10 | 75 | 0 | 0 | 4 | 9 | 10 | KRC | KARACHAY | CYRILLIC | 10 | 115 | 0 | 0 | 0 | 0 | 8 |
| DIQ | ZAZAKI | LATIN | 2 | 14 | 0 | 1 | 0 | 0 | 2 | KU | KURDISH | LATIN | 9 | 67 | 0 | 0 | 0 | 0 | 7 |
| DV | DHIVEHI | OTHER | 10 | 2 | 0 | 0 | 0 | 0 | 1 | KV | KOMI | CYRILLIC | 5 | 92 | 0 | 0 | 0 | 0 | 4 |
| EL | GREEK | OTHER | 10 | 62 | 0 | 0 | 0 | 2 | 8 | KW | CORNISH | LATIN | 9 | 44 | 0 | 0 | 0 | 0 | 6 |
| EO | ESPERANTO | LATIN | 10 | 70 | 0 | 0 | 0 | 0 | 8 | KY | KYRGYZ | CYRILLIC | 10 | 89 | 7 | 2 | 0 | 0 | 6 |
| ES | SPANISH | LATIN | 10 | 80 | 5 | 0 | 7 | 12 | 10 | LA | LATIN | LATIN | 10 | 60 | 0 | 0 | 0 | 0 | 8 |
| ET | ESTONIAN | LATIN | 10 | 100 | 0 | 0 | 0 | 8 | 8 | LB | LUXEMB. | LATIN | 7 | 123 | 0 | 8 | 0 | 0 | 3 |
| EU | BASQUE | LATIN | 10 | 75 | 1 | 4 | 0 | 0 | 7 | LEZ | LEZGIAN | CYRILLIC | 4 | 63 | 0 | 1 | 0 | 0 | 3 |
| FA | PERSIAN | ARABIC | 10 | 31 | 0 | 0 | 0 | 0 | 3 | LI | LIMBURGISH | LATIN | 2 | 29 | 0 | 0 | 0 | 0 | 1 |
| FI | FINNISH | LATIN | 7 | 77 | 0 | 0 | 0 | 0 | 7 | LMO | LOMBARD | LATIN | 8 | 50 | 0 | 0 | 0 | 0 | 5 |
| FR | FRENCH | LATIN | 9 | 68 | 0 | 0 | 3 | 14 | 7 | LO | LAO | SANSKRIT | 10 | 3 | 0 | 0 | 0 | 0 | 1 |
| FY | W. FRISIAN | LATIN | 10 | 31 | 0 | 0 | 0 | 0 | 4 | LT | LITHUANIAN | LATIN | 10 | 100 | 0 | 0 | 0 | 0 | 8 |
| GA | IRISH | LATIN | 9 | 43 | 0 | 2 | 0 | 0 | 6 | LV | LATVIAN | LATIN | 10 | 182 | 0 | 2 | 0 | 0 | 7 |
| GD | SC. GAELIC | LATIN | 9 | 163 | 0 | 0 | 0 | 0 | 8 | MG | MALAGASY | LATIN | 10 | 76 | 0 | 0 | 0 | 0 | 9 |

*Note: Lay.: Layout, Par.: Paragraph, For.: Formula, Tab.: Table, Cod.: Code, Cat.: Catalog, Ord.: Reading Order.*

*Table 14.* Benchmark statistics (Part II: mhr - yue).

| ISO | LANG. | FAMILY | LAY. | PAR. | FOR. | TAB. | COD. | CAT. | ORD. | ISO | LANG. | FAMILY | LAY. | PAR. | FOR. | TAB. | COD. | CAT. | ORD. |
|---|---|---|---|---|---|---|---|---|---|---|---|---|---|---|---|---|---|---|---|
| MHR | M. MARI | CYRILLIC | 9 | 133 | 0 | 1 | 0 | 0 | 5 | SD | SINDHI | ARABIC | 10 | 4 | 0 | 1 | 0 | 0 | 1 |
| MIN | MINANGK. | LATIN | 1 | 0 | 0 | 0 | 0 | 0 | 0 | SH | SERBO-CRO. | CYRILLIC | 10 | 92 | 0 | 0 | 0 | 0 | 9 |
| MK | MACEDONIAN | CYRILLIC | 10 | 61 | 0 | 0 | 0 | 0 | 7 | SI | SINHALA | SANSKRIT | 9 | 0 | 0 | 0 | 0 | 0 | 0 |
| ML | MALAYALAM | SANSKRIT | 10 | 6 | 0 | 0 | 0 | 0 | 3 | SK | SLOVAK | LATIN | 10 | 132 | 0 | 0 | 0 | 0 | 8 |
| MN | MONGOLIAN | CYRILLIC | 10 | 43 | 0 | 0 | 0 | 0 | 6 | SL | SLOVENIAN | LATIN | 10 | 90 | 7 | 0 | 0 | 0 | 9 |
| MR | MARATHI | SANSKRIT | 9 | 52 | 0 | 0 | 0 | 0 | 7 | SO | SOMALI | LATIN | 8 | 54 | 0 | 0 | 0 | 0 | 5 |
| MS | MALAY | LATIN | 10 | 41 | 0 | 0 | 0 | 0 | 5 | SQ | ALBANIAN | LATIN | 10 | 58 | 0 | 0 | 0 | 0 | 7 |
| MT | MALTESE | LATIN | 6 | 26 | 0 | 4 | 0 | 0 | 6 | SR | SERBIAN | CYRILLIC | 10 | 116 | 0 | 0 | 0 | 0 | 8 |
| MWL | MIRANDESE | LATIN | 2 | 0 | 0 | 0 | 0 | 0 | 0 | SU | SUNDANESE | LATIN | 9 | 70 | 0 | 5 | 0 | 0 | 6 |
| MY | BURMESE | SANSKRIT | 10 | 0 | 0 | 1 | 0 | 0 | 0 | SV | SWEDISH | LATIN | 10 | 54 | 0 | 0 | 0 | 0 | 9 |
| MYV | ERZYA | CYRILLIC | 2 | 1 | 0 | 0 | 0 | 0 | 0 | SW | SWAHILI | LATIN | 10 | 80 | 0 | 0 | 0 | 0 | 7 |
| MZN | MAZANDERANI | ARABIC | 10 | 136 | 0 | 0 | 0 | 0 | 10 | TA | TAMIL | SANSKRIT | 10 | 10 | 0 | 0 | 0 | 0 | 3 |
| NDS | LOW GER. | LATIN | 10 | 26 | 0 | 1 | 0 | 0 | 2 | TE | TELUGU | SANSKRIT | 10 | 14 | 0 | 1 | 0 | 0 | 4 |
| NE | NEPALI | SANSKRIT | 10 | 54 | 0 | 1 | 0 | 0 | 8 | TG | TAJIK | CYRILLIC | 10 | 74 | 0 | 0 | 0 | 0 | 10 |
| NEW | NEWAR | SANSKRIT | 2 | 24 | 0 | 0 | 0 | 0 | 2 | TH | THAI | SANSKRIT | 10 | 40 | 0 | 0 | 0 | 0 | 7 |
| NL | DUTCH | LATIN | 10 | 138 | 0 | 0 | 0 | 0 | 10 | TK | TURKMEN | LATIN | 10 | 32 | 0 | 1 | 0 | 0 | 5 |
| NN | NYNORSK | LATIN | 10 | 103 | 3 | 2 | 0 | 0 | 10 | TL | TAGALOG | LATIN | 10 | 75 | 0 | 1 | 0 | 0 | 9 |
| NO | NORWEGIAN | LATIN | 9 | 51 | 0 | 0 | 0 | 0 | 8 | TR | TURKISH | LATIN | 10 | 89 | 0 | 0 | 0 | 3 | 7 |
| OC | OCCITAN | LATIN | 4 | 38 | 0 | 3 | 0 | 0 | 2 | TT | TATAR | CYRILLIC | 9 | 132 | 7 | 0 | 0 | 0 | 6 |
| OR | ODIA | SANSKRIT | 9 | 1 | 0 | 0 | 0 | 0 | 1 | TYV | TUVAN | CYRILLIC | 2 | 13 | 0 | 0 | 0 | 0 | 1 |
| OS | OSSETIAN | CYRILLIC | 3 | 12 | 0 | 0 | 0 | 0 | 1 | UG | UYGHUR | ARABIC | 10 | 18 | 0 | 0 | 0 | 0 | 2 |
| PA | PUNJABI | SANSKRIT | 10 | 15 | 0 | 3 | 0 | 0 | 4 | UK | UKRAINIAN | CYRILLIC | 10 | 92 | 0 | 0 | 0 | 4 | 9 |
| PAM | KAPAM. | LATIN | 2 | 43 | 0 | 0 | 0 | 0 | 1 | UR | URDU | ARABIC | 9 | 4 | 0 | 0 | 0 | 0 | 0 |
| PL | POLISH | LATIN | 10 | 56 | 0 | 0 | 16 | 2 | 9 | UZ | UZBEK | LATIN | 10 | 81 | 0 | 1 | 0 | 0 | 7 |
| PMS | PIEDMONT. | LATIN | 6 | 18 | 0 | 0 | 0 | 0 | 3 | VEC | VENETIAN | LATIN | 2 | 18 | 0 | 2 | 0 | 0 | 2 |
| PNB | W. PUNJABI | ARABIC | 10 | 10 | 1 | 0 | 0 | 0 | 1 | VI | VIETNAMESE | LATIN | 8 | 65 | 2 | 0 | 0 | 0 | 7 |
| PS | PASHTO | ARABIC | 9 | 9 | 0 | 0 | 0 | 0 | 1 | WA | WALLOON | LATIN | 2 | 33 | 0 | 0 | 0 | 0 | 1 |
| PT | PORTUGUESE | LATIN | 10 | 80 | 0 | 0 | 7 | 3 | 9 | WAR | WARAY | LATIN | 9 | 95 | 0 | 10 | 0 | 0 | 8 |
| QU | QUECHUA | LATIN | 10 | 40 | 0 | 2 | 0 | 0 | 6 | WUU | WU CHIN. | CHINESE | 6 | 51 | 0 | 0 | 0 | 0 | 5 |
| RM | ROMANSH | LATIN | 5 | 15 | 0 | 0 | 0 | 0 | 2 | XAL | KALMYK | CYRILLIC | 1 | 25 | 0 | 0 | 0 | 0 | 1 |
| RO | ROMANIAN | LATIN | 9 | 103 | 0 | 0 | 0 | 2 | 8 | XMF | MINGRELIAN | OTHER | 10 | 110 | 0 | 0 | 0 | 0 | 10 |
| RU | RUSSIAN | CYRILLIC | 10 | 88 | 0 | 0 | 26 | 49 | 6 | YI | YIDDISH | OTHER | 10 | 58 | 0 | 0 | 0 | 0 | 2 |
| SA | SANSKRIT | SANSKRIT | 10 | 40 | 3 | 0 | 0 | 0 | 7 | YUE | CANTONESE | CHINESE | 7 | 12 | 0 | 0 | 0 | 0 | 3 |
| SAH | YAKUT | CYRILLIC | 10 | 149 | 0 | 0 | 0 | 0 | 8 | | | | | | | | | | |

*Note: Lay.: Layout, Par.: Paragraph, For.: Formula, Tab.: Table, Cod.: Code, Cat.: Catalog, Ord.: Reading Order.*

## B.2. Detailed Evaluation Results

In this section, we provide the comprehensive breakdown of the extensive evaluation results for all languages and models across the four critical metrics. The following tables present the granular statistics for Overall Score, Text Score, Table Score, and Reading Order, respectively.

### B.2.1. EVALUATION ON OVERALL SCORE

We report the extensive monolingual overall scores for the MORE benchmark in Tables 15 and 16, offering an in-depth and holistic view of diverse model performance.

*Table 15.* Overall evaluation (Part I: af - de).

| ISO | FAMILY | HY. | Qw3 | Qw2.5 | DOTS | PD. | DS | MIN. | ISO | FAMILY | HY. | Qw3 | Qw2.5 | DOTS | PD. | DS | MIN. |
|---|---|---|---|---|---|---|---|---|---|---|---|---|---|---|---|---|---|
| AF | LATIN | 99.38 | 98.34 | 82.22 | **99.90** | 97.89 | 93.84 | 47.32 | BN | SANSKRIT | 90.69 | **91.16** | 66.54 | 63.11 | 57.81 | 61.70 | 50.00 |
| ALS | LATIN | **96.78** | 96.28 | 91.38 | 93.83 | 96.58 | 89.73 | 43.84 | BO | SANSKRIT | 89.05 | 84.12 | 66.67 | **89.28** | 66.83 | 88.75 | 66.86 |
| AM | OTHER | **10.00** | 0.00 | 0.00 | 0.00 | 0.00 | 0.00 | 1.00 | BR | LATIN | 92.20 | 95.21 | 87.76 | 62.53 | **97.14** | 95.50 | 30.31 |
| AN | LATIN | 91.30 | 94.29 | 80.91 | **97.33** | 95.52 | 94.19 | 68.44 | BS | LATIN | 92.53 | 88.19 | 83.84 | 91.95 | **95.91** | 90.34 | 69.63 |
| AR | ARABIC | **94.56** | 73.65 | 87.80 | 90.01 | 79.82 | 81.33 | 36.01 | BXR | CYRILLIC | 97.25 | 96.27 | 91.15 | **97.34** | 96.47 | 94.47 | 25.62 |
| ARZ | ARABIC | **94.84** | 80.03 | 76.19 | 94.80 | 86.33 | 84.69 | 33.34 | CA | LATIN | 89.98 | 87.28 | 83.97 | **95.35** | 92.65 | 83.08 | 49.23 |
| AST | LATIN | 96.95 | 91.66 | 85.68 | **97.06** | 95.27 | 87.19 | 46.98 | CE | CYRILLIC | **94.28** | 81.29 | 85.42 | 82.39 | 83.85 | 81.37 | 18.80 |
| AV | CYRILLIC | 96.70 | 92.78 | 65.71 | 97.22 | 96.22 | **97.44** | 15.64 | CEB | LATIN | 78.85 | 64.50 | 69.19 | 70.18 | 65.69 | **89.28** | 19.20 |
| AZ | LATIN | 98.15 | 89.88 | 94.75 | **98.69** | 93.35 | 89.04 | 55.77 | CKB | ARABIC | 82.34 | 41.02 | 83.05 | **83.24** | 78.27 | 72.47 | 0.20 |
| AZB | ARABIC | **96.51** | 94.32 | 93.14 | 93.19 | 70.44 | 75.34 | 25.00 | CS | LATIN | 98.98 | 94.78 | 96.77 | **99.58** | 97.37 | 98.34 | 80.25 |
| BA | CYRILLIC | 89.81 | 85.92 | 82.13 | 89.12 | 89.07 | 85.76 | 33.00 | CV | CYRILLIC | **81.19** | 74.63 | 68.02 | 76.09 | 74.26 | 74.37 | 34.41 |
| BAR | LATIN | 98.56 | 98.06 | 98.56 | 98.56 | 97.01 | 98.13 | 50.00 | CY | LATIN | 89.79 | 68.15 | 84.84 | 63.72 | 80.65 | **91.77** | 18.07 |
| BE | CYRILLIC | 97.13 | 96.58 | 94.09 | **97.28** | 96.41 | 94.25 | 41.24 | DA | LATIN | 95.81 | 99.90 | 99.74 | **99.97** | 99.38 | 93.27 | 59.11 |
| BG | CYRILLIC | **99.05** | 95.66 | 89.25 | 94.25 | 96.91 | 92.87 | 16.80 | DE | LATIN | **98.10** | 97.80 | 92.51 | 94.20 | 97.24 | 90.74 | 81.33 |

*Note: HY.: HunyuanOCR, Qw3: Qwen3-VL-2B, Qw2.5: Qwen2.5-VL-3B, DOTS: dots.ocr, PD.: PaddleOCR-VL, DS: DeepSeekOCR, Min.: MinerU2.5.*

*Table 16.* Overall evaluation (Part II: diq - yue).

| ISO | FAMILY | HY. | Qw3 | Qw2.5 | DOTS | PD. | DS | MIN. | ISO | FAMILY | HY. | Qw3 | Qw2.5 | DOTS | PD. | DS | MIN. |
|---|---|---|---|---|---|---|---|---|---|---|---|---|---|---|---|---|---|
| DIQ | LATIN | 77.24 | 77.26 | 71.28 | **77.91** | 77.06 | 74.68 | 46.99 | MS | LATIN | 97.03 | 90.06 | 79.93 | 90.58 | **97.06** | 94.42 | 68.75 |
| DV | OTHER | 87.50 | 58.37 | **100.0** | 66.66 | 68.34 | 58.62 | 0.00 | MT | LATIN | 95.56 | 89.54 | 88.99 | 85.76 | **96.18** | 94.59 | 42.09 |
| EL | OTHER | 94.05 | 90.17 | 85.04 | 91.91 | 93.06 | 92.33 | 28.81 | MWL | LATIN | 2.00 | 0.00 | 0.00 | 0.00 | 0.00 | 0.00 | 0.00 |
| EO | LATIN | 98.62 | 98.19 | 92.09 | 98.45 | 97.62 | 96.78 | 76.79 | MY | SANSKRIT | 3.57 | 50.94 | 49.60 | 0.00 | 0.00 | 0.00 | 0.00 |
| ES | LATIN | 97.25 | 84.25 | 91.69 | 92.81 | 92.40 | 82.32 | 61.35 | MYV | CYRILLIC | 49.21 | 48.29 | 48.09 | 48.34 | 49.04 | 49.12 | 0.00 |
| ET | LATIN | 99.06 | 95.06 | 88.56 | 98.85 | 96.32 | 92.35 | 40.03 | MZN | ARABIC | 100.0 | 99.94 | 99.95 | 100.0 | 99.41 | 93.70 | 74.84 |
| EU | LATIN | 96.78 | 91.69 | 82.91 | 84.90 | 91.49 | 95.25 | 33.73 | NDS | LATIN | 93.16 | 88.48 | 79.95 | 90.35 | 89.96 | 86.22 | 38.47 |
| FA | ARABIC | 87.89 | 94.97 | 91.22 | 87.59 | 90.38 | 81.54 | 18.52 | NE | SANSKRIT | 95.68 | 90.39 | 85.07 | 60.95 | 94.58 | 48.82 | 24.07 |
| FI | LATIN | 91.59 | 97.27 | 81.40 | 96.70 | 96.48 | 69.52 | 35.48 | NEW | SANSKRIT | 88.76 | 84.79 | 75.07 | 86.94 | 89.77 | 66.92 | 50.00 |
| FR | LATIN | 95.62 | 95.72 | 95.16 | 92.29 | 97.01 | 93.31 | 67.02 | NL | LATIN | 99.97 | 99.90 | 84.75 | 99.95 | 95.70 | 91.19 | 69.47 |
| FY | LATIN | 99.44 | 93.06 | 98.23 | 97.65 | 93.69 | 94.86 | 53.15 | NN | LATIN | 95.50 | 89.02 | 79.83 | 85.05 | 82.85 | 79.74 | 42.22 |
| GA | LATIN | 90.18 | 82.58 | 83.22 | 65.46 | 87.03 | 63.19 | 48.17 | NO | LATIN | 99.92 | 99.71 | 93.30 | 99.92 | 99.28 | 99.16 | 81.09 |
| GD | LATIN | 95.78 | 88.78 | 85.27 | 97.52 | 95.26 | 89.56 | 50.23 | OC | LATIN | 86.41 | 81.08 | 92.16 | 87.11 | 67.77 | 78.95 | 46.38 |
| GL | LATIN | 94.64 | 94.58 | 92.97 | 95.05 | 94.63 | 89.11 | 53.52 | OR | SANSKRIT | 12.50 | 50.00 | 0.00 | 37.50 | 12.50 | 43.18 | 0.00 |
| GN | LATIN | 98.84 | 98.08 | 48.26 | 99.22 | 99.22 | 97.88 | 81.01 | OS | CYRILLIC | 97.95 | 95.96 | 95.29 | 96.53 | 96.03 | 97.25 | 51.34 |
| GOM | SANSKRIT | 93.62 | 91.28 | 90.34 | 92.86 | 92.20 | 86.31 | 50.10 | PA | SANSKRIT | 59.43 | 48.69 | 47.60 | 45.91 | 36.37 | 26.87 | 10.63 |
| GU | SANSKRIT | 70.96 | 69.35 | 49.03 | 52.62 | 61.88 | 39.28 | 36.02 | PAM | LATIN | 88.75 | 83.97 | 55.25 | 86.69 | 80.47 | 82.46 | 4.71 |
| GV | LATIN | 100.0 | 99.42 | 49.96 | 99.66 | 100.0 | 100.0 | 50.05 | PL | LATIN | 99.10 | 96.41 | 93.82 | 98.75 | 98.38 | 96.84 | 51.14 |
| HE | OTHER | 83.53 | 45.23 | 35.23 | 78.71 | 38.35 | 57.70 | 12.50 | PMS | LATIN | 99.17 | 89.80 | 80.14 | 92.62 | 94.00 | 95.47 | 54.80 |
| HI | SANSKRIT | 95.85 | 95.30 | 94.89 | 95.44 | 95.35 | 73.65 | 33.96 | PNB | ARABIC | 49.97 | 87.23 | 87.08 | 84.31 | 89.26 | 41.23 | 47.65 |
| HR | LATIN | 99.16 | 99.29 | 92.31 | 99.45 | 99.45 | 99.17 | 63.70 | PS | ARABIC | 85.44 | 93.02 | 40.43 | 92.12 | 94.52 | 76.19 | 5.42 |
| HSB | LATIN | 96.31 | 97.20 | 97.60 | 98.00 | 97.38 | 94.23 | 51.86 | PT | LATIN | 99.56 | 96.96 | 93.72 | 94.67 | 99.72 | 93.22 | 52.78 |
| HT | LATIN | 99.91 | 99.48 | 95.39 | 99.96 | 97.57 | 99.38 | 65.44 | QU | LATIN | 68.06 | 74.25 | 69.66 | 62.78 | 66.41 | 59.41 | 41.82 |
| HU | LATIN | 99.73 | 98.80 | 99.04 | 99.19 | 99.27 | 99.26 | 63.35 | RM | LATIN | 99.69 | 99.15 | 95.62 | 99.78 | 99.65 | 89.60 | 19.51 |
| HY | OTHER | 68.82 | 47.56 | 54.48 | 81.96 | 63.03 | 68.77 | 35.32 | RO | LATIN | 99.42 | 98.12 | 97.41 | 99.37 | 96.56 | 97.22 | 78.78 |
| IA | LATIN | 97.94 | 95.85 | 95.39 | 97.51 | 97.63 | 96.29 | 26.93 | RU | CYRILLIC | 99.02 | 96.03 | 95.80 | 94.84 | 98.31 | 86.97 | 34.44 |
| ID | LATIN | 98.18 | 98.51 | 94.58 | 95.57 | 99.40 | 98.10 | 55.10 | SA | SANSKRIT | 93.40 | 62.61 | 58.65 | 95.83 | 95.35 | 76.98 | 15.72 |
| IE | LATIN | 94.05 | 43.39 | 88.98 | 98.14 | 91.52 | 87.34 | 50.00 | SAH | CYRILLIC | 88.76 | 89.81 | 81.46 | 90.34 | 90.52 | 84.81 | 9.87 |
| ILO | LATIN | 98.71 | 83.63 | 94.98 | 64.03 | 97.01 | 96.61 | 59.91 | SD | ARABIC | 87.74 | 86.32 | 61.76 | 53.73 | 23.76 | 78.68 | 33.33 |
| IO | LATIN | 99.77 | 99.65 | 99.34 | 99.80 | 99.69 | 91.53 | 83.31 | SH | CYRILLIC | 99.12 | 97.55 | 98.59 | 98.09 | 99.75 | 98.15 | 77.09 |
| IS | LATIN | 99.20 | 99.03 | 93.41 | 99.87 | 97.72 | 98.44 | 78.33 | SK | LATIN | 99.06 | 88.68 | 91.72 | 99.66 | 93.36 | 96.57 | 48.80 |
| IT | LATIN | 98.70 | 90.45 | 98.39 | 83.10 | 83.02 | 80.40 | 52.14 | SL | LATIN | 97.82 | 96.65 | 87.91 | 97.66 | 94.97 | 91.01 | 78.98 |
| JA | CHINESE | 89.67 | 84.96 | 77.16 | 82.09 | 82.94 | 85.84 | 50.59 | SO | LATIN | 96.98 | 95.18 | 83.16 | 99.73 | 99.62 | 92.86 | 49.03 |
| JBO | LATIN | 98.82 | 93.81 | 81.65 | 65.73 | 90.51 | 90.66 | 49.09 | SQ | LATIN | 98.20 | 97.11 | 93.06 | 98.75 | 97.50 | 94.51 | 58.70 |
| JV | LATIN | 96.30 | 99.16 | 80.21 | 99.29 | 97.48 | 95.55 | 51.07 | SR | CYRILLIC | 98.75 | 99.66 | 98.34 | 99.93 | 99.65 | 90.92 | 68.73 |
| KA | OTHER | 84.43 | 65.75 | 28.24 | 91.44 | 45.94 | 77.19 | 0.37 | SU | LATIN | 99.67 | 99.45 | 98.78 | 92.88 | 92.84 | 91.88 | 39.43 |
| KK | CYRILLIC | 92.89 | 87.59 | 82.40 | 94.80 | 92.31 | 87.34 | 29.04 | SV | LATIN | 99.70 | 99.57 | 99.35 | 99.56 | 99.52 | 95.47 | 84.95 |
| KM | SANSKRIT | 58.57 | 43.15 | 36.22 | 28.22 | 40.91 | 37.86 | 16.55 | SW | LATIN | 98.75 | 97.86 | 90.71 | 98.77 | 98.29 | 97.00 | 74.31 |
| KN | SANSKRIT | 57.92 | 45.66 | 43.34 | 86.31 | 19.14 | 24.21 | 3.81 | TA | SANSKRIT | 88.00 | 91.14 | 57.40 | 89.58 | 88.56 | 87.24 | 16.66 |
| KO | CHINESE | 98.46 | 71.51 | 92.88 | 97.35 | 94.90 | 94.79 | 52.17 | TE | SANSKRIT | 84.98 | 74.37 | 43.93 | 53.71 | 65.11 | 55.86 | 17.27 |
| KRC | CYRILLIC | 98.38 | 94.38 | 85.89 | 98.23 | 98.75 | 87.11 | 12.81 | TG | CYRILLIC | 98.16 | 97.66 | 85.30 | 98.27 | 97.03 | 90.62 | 25.79 |
| KU | LATIN | 98.44 | 97.13 | 97.13 | 99.52 | 99.15 | 98.13 | 53.84 | TH | SANSKRIT | 99.19 | 93.30 | 73.63 | 98.77 | 99.00 | 97.45 | 28.57 |
| KV | CYRILLIC | 92.35 | 89.53 | 69.25 | 92.28 | 88.80 | 83.18 | 23.82 | TK | LATIN | 96.80 | 92.33 | 86.19 | 65.50 | 85.05 | 95.78 | 75.70 |
| KW | LATIN | 89.30 | 76.12 | 72.65 | 93.41 | 76.61 | 93.60 | 63.13 | TL | LATIN | 98.81 | 98.36 | 97.69 | 99.40 | 95.24 | 96.43 | 22.67 |
| KY | CYRILLIC | 88.26 | 68.94 | 68.25 | 82.77 | 90.38 | 83.59 | 39.20 | TR | LATIN | 99.74 | 98.33 | 97.18 | 99.55 | 99.16 | 96.60 | 59.28 |
| LA | LATIN | 99.34 | 99.14 | 99.03 | 99.25 | 99.52 | 94.55 | 74.11 | TT | CYRILLIC | 93.28 | 89.13 | 90.40 | 92.88 | 92.30 | 85.11 | 46.49 |
| LB | LATIN | 93.12 | 87.13 | 92.04 | 81.17 | 91.93 | 84.00 | 46.77 | TYV | CYRILLIC | 96.80 | 92.30 | 71.32 | 96.37 | 96.55 | 96.34 | 0.51 |
| LEZ | CYRILLIC | 70.22 | 77.08 | 51.69 | 63.48 | 71.85 | 54.72 | 11.23 | UG | ARABIC | 87.90 | 87.10 | 59.63 | 60.98 | 76.48 | 76.41 | 0.00 |
| LI | LATIN | 98.14 | 97.75 | 93.40 | 99.92 | 97.92 | 95.88 | 50.00 | UK | CYRILLIC | 95.33 | 95.18 | 89.56 | 93.89 | 98.74 | 95.85 | 25.64 |
| LMO | LATIN | 98.64 | 97.24 | 76.34 | 98.22 | 97.61 | 93.23 | 67.14 | UR | ARABIC | 64.76 | 42.15 | 41.16 | 53.80 | 61.19 | 70.54 | 0.00 |
| LO | SANSKRIT | 64.50 | 50.98 | 62.80 | 0.89 | 56.08 | 50.70 | 0.00 | UZ | LATIN | 96.21 | 78.23 | 90.57 | 65.20 | 93.18 | 92.01 | 32.30 |
| LT | LATIN | 99.43 | 98.33 | 97.88 | 99.74 | 97.42 | 99.45 | 69.72 | VEC | LATIN | 77.57 | 80.75 | 81.85 | 78.80 | 74.88 | 80.16 | 38.41 |
| LV | LATIN | 89.97 | 75.24 | 75.65 | 81.63 | 92.31 | 73.98 | 35.61 | VI | LATIN | 98.12 | 97.76 | 85.37 | 97.54 | 95.35 | 62.86 | 57.88 |
| MG | LATIN | 96.60 | 91.69 | 90.88 | 99.76 | 98.10 | 93.97 | 59.31 | WA | LATIN | 98.19 | 94.93 | 97.22 | 99.08 | 90.81 | 96.11 | 5.00 |
| MHR | CYRILLIC | 81.78 | 91.68 | 74.23 | 82.92 | 54.69 | 72.57 | 1.23 | WAR | LATIN | 76.11 | 62.36 | 66.37 | 56.45 | 58.97 | 55.03 | 27.03 |
| MIN | LATIN | 1.00 | 0.00 | 0.00 | 0.00 | 0.00 | 0.00 | 0.00 | WUU | CHINESE | 95.22 | 86.94 | 65.42 | 95.07 | 85.42 | 87.09 | 34.83 |
| MK | CYRILLIC | 93.37 | 90.61 | 88.61 | 90.62 | 90.12 | 87.79 | 33.03 | XAL | CYRILLIC | 91.69 | 82.78 | 78.14 | 82.74 | 83.34 | 79.94 | 0.92 |
| ML | SANSKRIT | 25.56 | 41.66 | 9.87 | 63.88 | 41.66 | 66.87 | 16.66 | XMF | OTHER | 76.50 | 26.34 | 17.00 | 94.04 | 43.47 | 92.72 | 22.34 |
| MN | CYRILLIC | 97.05 | 96.62 | 73.06 | 98.09 | 95.94 | 96.52 | 30.66 | YI | OTHER | 78.84 | 77.40 | 56.03 | 83.84 | 68.25 | 71.97 | 19.01 |
| MR | SANSKRIT | 94.64 | 84.17 | 62.38 | 91.77 | 89.12 | 82.20 | 21.98 | YUE | CHINESE | 95.37 | 78.03 | 89.76 | 92.61 | 92.60 | 91.75 | 41.01 |

*Note: HY.: HunyuanOCR, Qw3: Qwen3-VL-2B, Qw2.5: Qwen2.5-VL-3B, DOTS: dots.ocr, PD.: PaddleOCR-VL, DS: DeepSeekOCR, Min.: MinerU2.5.*

## B.2.2. EVALUATION ON TEXT SCORE

Complementing the overall evaluation, Tables 17 and 18 isolate the text-only performance metrics across the MORE.

*Table 17.* Text evaluation (Part I: af - war).

| ISO | FAMILY | HY. | Qw3 | Qw2.5 | DOTS | PD. | DS | MIN. | ISO | FAMILY | HY. | Qw3 | Qw2.5 | DOTS | PD. | DS | MIN. |
|---|---|---|---|---|---|---|---|---|---|---|---|---|---|---|---|---|---|
| AF | LATIN | 99.81 | 99.29 | 97.99 | 99.80 | 99.43 | 90.27 | 51.78 | KW | LATIN | 95.26 | 85.56 | 89.04 | 95.14 | 86.55 | 95.53 | 26.27 |
| ALS | LATIN | 93.57 | 92.55 | 91.08 | 92.65 | 93.15 | 84.46 | 27.69 | KY | CYRILLIC | 87.00 | 86.66 | 81.41 | 86.64 | 88.10 | 73.33 | 0.77 |
| AN | LATIN | 98.81 | 96.81 | 95.48 | 97.07 | 97.50 | 97.34 | 41.62 | LA | LATIN | 98.67 | 98.28 | 98.07 | 98.51 | 99.04 | 89.09 | 51.00 |
| AR | ARABIC | 89.18 | 80.30 | 76.12 | 76.05 | 79.53 | 62.19 | 0.00 | LB | LATIN | 98.98 | 98.64 | 92.71 | 99.22 | 97.38 | 75.17 | 16.64 |
| ARZ | ARABIC | 89.68 | 82.27 | 85.71 | 89.60 | 83.76 | 69.39 | 0.00 | LEZ | CYRILLIC | 94.13 | 90.72 | 85.05 | 92.66 | 92.76 | 83.18 | 0.36 |
| AST | LATIN | 93.90 | 88.76 | 90.18 | 94.11 | 93.13 | 83.60 | 22.52 | LI | LATIN | 96.28 | 95.50 | 97.90 | 99.84 | 95.84 | 91.77 | 0.00 |
| AV | CYRILLIC | 93.40 | 85.56 | 85.27 | 94.44 | 92.43 | 94.87 | 0.51 | LMO | LATIN | 97.28 | 94.48 | 92.67 | 97.97 | 95.22 | 86.47 | 40.44 |
| AZ | LATIN | 96.29 | 89.28 | 89.49 | 97.39 | 86.71 | 97.13 | 44.18 | LO | SANSKRIT | 29.00 | 1.96 | 25.59 | 1.78 | 12.15 | 1.39 | 0.00 |
| AZB | ARABIC | 93.02 | 88.64 | 86.28 | 86.38 | 90.87 | 50.68 | 0.00 | LT | LATIN | 98.86 | 97.91 | 97.00 | 99.48 | 94.85 | 98.91 | 39.43 |
| BA | CYRILLIC | 82.03 | 82.96 | 78.08 | 84.05 | 81.81 | 79.64 | 1.47 | LV | LATIN | 94.39 | 92.65 | 88.70 | 97.23 | 89.93 | 79.91 | 10.32 |
| BAR | LATIN | 97.11 | 96.12 | 97.11 | 97.11 | 94.02 | 96.26 | 0.00 | MG | LATIN | 99.55 | 98.98 | 98.85 | 99.52 | 99.37 | 94.29 | 51.96 |
| BE | CYRILLIC | 94.27 | 93.16 | 88.18 | 94.56 | 92.81 | 88.50 | 4.70 | MHR | CYRILLIC | 72.82 | 83.06 | 74.54 | 56.79 | 79.07 | 67.06 | 1.02 |
| BG | CYRILLIC | 98.09 | 97.67 | 95.16 | 98.02 | 93.82 | 85.73 | 1.21 | MK | CYRILLIC | 86.73 | 91.30 | 91.24 | 93.01 | 86.95 | 75.58 | 0.63 |
| BN | SANSKRIT | 81.37 | 26.20 | 26.52 | 26.22 | 15.63 | 23.40 | 0.00 | ML | SANSKRIT | 17.79 | 0.00 | 3.07 | 27.75 | 0.00 | 33.74 | 0.00 |
| BR | LATIN | 98.36 | 97.51 | 96.68 | 97.59 | 97.81 | 91.86 | 40.94 | MN | CYRILLIC | 94.10 | 93.25 | 85.55 | 96.19 | 93.95 | 93.04 | 2.40 |
| BS | LATIN | 99.50 | 98.92 | 98.14 | 99.75 | 99.55 | 94.17 | 49.57 | MR | SANSKRIT | 92.84 | 88.89 | 81.91 | 91.58 | 88.42 | 64.41 | 0.82 |
| BXR | CYRILLIC | 94.50 | 92.53 | 88.54 | 94.67 | 92.95 | 95.20 | 1.24 | MS | LATIN | 94.05 | 89.21 | 88.95 | 98.43 | 94.11 | 88.84 | 41.93 |
| CA | LATIN | 91.74 | 90.15 | 89.34 | 94.39 | 89.31 | 79.27 | 29.44 | MT | LATIN | 99.31 | 95.98 | 97.29 | 99.77 | 95.40 | 95.14 | 69.05 |
| CE | CYRILLIC | 92.87 | 92.70 | 88.15 | 94.11 | 92.45 | 79.90 | 0.77 | MYV | CYRILLIC | 49.21 | 48.29 | 48.09 | 48.34 | 49.04 | 49.12 | 0.00 |
| CEB | LATIN | 99.22 | 98.67 | 95.01 | 99.22 | 99.19 | 90.62 | 0.39 | MZN | ARABIC | 100.00 | 99.89 | 99.91 | 100.00 | 100.00 | 87.40 | 49.69 |
| CKB | ARABIC | 64.68 | 74.89 | 66.09 | 66.48 | 63.68 | 44.93 | 0.39 | NDS | LATIN | 97.23 | 95.93 | 96.44 | 96.95 | 96.95 | 85.46 | 4.04 |
| CS | LATIN | 97.96 | 94.49 | 96.01 | 99.16 | 94.73 | 96.67 | 60.51 | NE | SANSKRIT | 88.41 | 85.25 | 82.43 | 89.10 | 86.17 | 60.06 | 6.85 |
| CV | CYRILLIC | 80.07 | 79.41 | 75.97 | 81.53 | 73.21 | 60.42 | 0.64 | NEW | SANSKRIT | 77.52 | 85.21 | 78.27 | 83.25 | 79.54 | 40.10 | 0.00 |
| CY | LATIN | 98.64 | 98.04 | 96.80 | 97.77 | 98.68 | 85.65 | 9.56 | NL | LATIN | 99.93 | 99.79 | 99.01 | 99.90 | 99.47 | 87.38 | 38.95 |
| DA | LATIN | 99.94 | 99.80 | 99.48 | 99.94 | 98.77 | 86.54 | 62.66 | NN | LATIN | 97.91 | 98.18 | 97.10 | 99.32 | 97.21 | 87.31 | 32.67 |
| DE | LATIN | 99.84 | 99.38 | 99.67 | 99.90 | 99.93 | 91.54 | 72.68 | NO | LATIN | 99.84 | 99.42 | 99.09 | 99.84 | 98.57 | 98.31 | 74.67 |
| DIQ | LATIN | 94.73 | 94.79 | 91.16 | 97.04 | 94.19 | 87.05 | 40.96 | OC | LATIN | 97.77 | 97.89 | 96.39 | 98.79 | 98.16 | 85.12 | 22.67 |
| DV | OTHER | 75.00 | 16.73 | 100.00 | 33.33 | 36.67 | 17.23 | 0.00 | OR | SANSKRIT | 0.00 | 0.00 | 0.00 | 0.00 | 0.00 | 11.36 | 0.00 |
| EL | OTHER | 92.20 | 90.87 | 90.19 | 92.87 | 92.46 | 80.37 | 7.95 | OS | CYRILLIC | 95.90 | 91.92 | 90.58 | 93.05 | 92.06 | 94.50 | 2.67 |
| EO | LATIN | 97.25 | 96.37 | 96.69 | 96.90 | 95.23 | 93.55 | 53.58 | PA | SANSKRIT | 36.61 | 34.73 | 31.53 | 37.72 | 34.12 | 30.60 | 4.00 |
| ES | LATIN | 99.60 | 99.16 | 99.19 | 99.55 | 99.55 | 89.99 | 54.65 | PAM | LATIN | 97.50 | 87.93 | 81.92 | 93.39 | 89.51 | 93.49 | 9.43 |
| ET | LATIN | 98.11 | 95.59 | 95.08 | 97.71 | 96.51 | 90.06 | 30.07 | PL | LATIN | 98.90 | 97.68 | 97.99 | 98.92 | 98.30 | 94.23 | 60.41 |
| EU | LATIN | 99.56 | 97.25 | 97.88 | 99.43 | 99.71 | 90.87 | 36.77 | PMS | LATIN | 98.34 | 96.27 | 91.39 | 96.35 | 88.00 | 90.94 | 76.28 |
| FA | ARABIC | 81.35 | 89.95 | 84.30 | 78.87 | 80.75 | 66.78 | 0.00 | PNB | ARABIC | 80.11 | 79.88 | 79.45 | 69.93 | 86.89 | 23.69 | 42.94 |
| FI | LATIN | 91.83 | 94.54 | 77.09 | 95.74 | 93.63 | 44.02 | 15.30 | PS | ARABIC | 70.87 | 86.03 | 80.86 | 84.24 | 89.03 | 52.37 | 10.83 |
| FR | LATIN | 99.90 | 99.80 | 99.28 | 99.96 | 99.97 | 92.73 | 53.40 | PT | LATIN | 99.63 | 97.11 | 98.37 | 99.40 | 99.77 | 77.66 | 51.67 |
| FY | LATIN | 98.88 | 95.52 | 96.46 | 99.15 | 95.70 | 95.28 | 10.14 | QU | LATIN | 99.33 | 98.07 | 98.30 | 99.45 | 99.22 | 98.71 | 51.47 |
| GA | LATIN | 96.36 | 92.32 | 88.00 | 96.39 | 94.18 | 95.12 | 61.19 | RM | LATIN | 99.37 | 98.30 | 91.23 | 99.57 | 99.30 | 79.20 | 39.02 |
| GD | LATIN | 97.39 | 96.42 | 95.00 | 98.61 | 97.66 | 85.23 | 25.47 | RO | LATIN | 98.25 | 97.01 | 96.13 | 98.96 | 92.17 | 93.15 | 43.53 |
| GL | LATIN | 99.57 | 98.33 | 95.97 | 99.72 | 95.98 | 81.92 | 32.41 | RU | CYRILLIC | 99.89 | 99.96 | 99.21 | 99.79 | 99.86 | 83.63 | 0.01 |
| GN | LATIN | 97.67 | 96.15 | 96.52 | 98.45 | 98.45 | 95.75 | 62.02 | SA | SANSKRIT | 90.50 | 87.83 | 82.35 | 90.88 | 89.46 | 62.90 | 0.51 |
| GOM | SANSKRIT | 87.23 | 82.56 | 80.68 | 85.72 | 84.41 | 72.63 | 0.20 | SAH | CYRILLIC | 86.01 | 86.40 | 72.01 | 84.37 | 87.20 | 76.11 | 0.29 |
| GU | SANSKRIT | 41.92 | 38.70 | 31.39 | 38.56 | 23.75 | 45.24 | 5.36 | SD | ARABIC | 72.38 | 70.14 | 65.92 | 61.20 | 71.29 | 68.71 | 0.00 |
| GV | LATIN | 100.00 | 98.84 | 99.92 | 99.33 | 100.00 | 100.00 | 0.11 | SH | CYRILLIC | 99.54 | 99.02 | 98.48 | 98.79 | 99.50 | 96.29 | 54.17 |
| HE | OTHER | 67.06 | 15.47 | 45.47 | 69.92 | 1.71 | 40.41 | 0.00 | SK | LATIN | 98.13 | 97.36 | 96.13 | 99.33 | 96.66 | 93.14 | 35.09 |
| HI | SANSKRIT | 91.71 | 90.60 | 89.78 | 90.88 | 90.75 | 80.63 | 1.25 | SL | LATIN | 99.77 | 99.41 | 98.84 | 99.76 | 99.83 | 98.38 | 53.10 |
| HR | LATIN | 99.00 | 99.25 | 97.11 | 99.58 | 99.58 | 98.35 | 37.41 | SO | LATIN | 99.30 | 98.36 | 94.31 | 99.47 | 99.24 | 96.76 | 38.07 |
| HSB | LATIN | 92.63 | 94.40 | 95.21 | 96.00 | 94.75 | 90.46 | 43.71 | SQ | LATIN | 98.55 | 95.65 | 97.06 | 99.64 | 96.43 | 90.45 | 45.98 |
| HT | LATIN | 99.83 | 98.97 | 97.03 | 99.92 | 99.32 | 98.75 | 55.89 | SR | CYRILLIC | 99.77 | 99.33 | 98.95 | 99.86 | 99.67 | 94.34 | 37.46 |
| HU | LATIN | 99.46 | 99.03 | 98.08 | 99.82 | 98.74 | 98.52 | 36.70 | SU | LATIN | 99.04 | 98.85 | 97.86 | 98.80 | 99.01 | 95.65 | 51.62 |
| HY | OTHER | 44.39 | 24.67 | 27.01 | 60.19 | 21.09 | 56.23 | 1.38 | SV | LATIN | 99.41 | 99.14 | 98.70 | 99.11 | 99.03 | 90.94 | 81.02 |
| IA | LATIN | 93.86 | 88.33 | 91.96 | 92.57 | 93.02 | 89.00 | 45.08 | SW | LATIN | 97.50 | 98.36 | 95.71 | 97.53 | 96.58 | 93.99 | 52.20 |
| ID | LATIN | 99.60 | 97.77 | 99.07 | 99.60 | 99.18 | 93.57 | 55.31 | TA | SANSKRIT | 76.00 | 82.28 | 48.13 | 79.16 | 77.12 | 74.48 | 0.00 |
| IE | LATIN | 88.10 | 86.78 | 77.96 | 96.28 | 83.04 | 74.69 | 0.00 | TE | SANSKRIT | 61.96 | 53.76 | 56.78 | 86.14 | 88.55 | 67.58 | 0.20 |
| ILO | LATIN | 99.84 | 99.06 | 98.64 | 99.50 | 99.29 | 93.54 | 24.67 | TG | CYRILLIC | 96.32 | 95.33 | 92.81 | 96.53 | 94.06 | 88.97 | 0.26 |
| IO | LATIN | 99.54 | 99.29 | 98.68 | 99.59 | 99.37 | 83.05 | 66.63 | TH | SANSKRIT | 98.37 | 91.36 | 90.12 | 97.53 | 98.01 | 94.90 | 0.00 |
| IS | LATIN | 98.40 | 98.06 | 96.82 | 99.74 | 95.45 | 96.87 | 56.66 | TK | LATIN | 97.45 | 89.91 | 87.44 | 96.49 | 88.95 | 93.15 | 53.98 |
| IT | LATIN | 98.64 | 99.18 | 99.27 | 99.85 | 99.11 | 90.71 | 41.39 | TL | LATIN | 99.47 | 98.94 | 98.44 | 99.27 | 98.01 | 93.97 | 23.58 |
| JA | CHINESE | 96.28 | 96.54 | 93.39 | 96.55 | 97.59 | 91.24 | 27.82 | TR | LATIN | 99.26 | 98.16 | 97.66 | 99.07 | 99.09 | 95.05 | 37.37 |
| JBO | LATIN | 97.51 | 97.68 | 96.62 | 97.18 | 97.12 | 93.55 | 22.27 | TT | CYRILLIC | 89.50 | 90.57 | 85.72 | 90.94 | 88.92 | 84.05 | 0.48 |
| JV | LATIN | 98.84 | 98.32 | 97.57 | 98.58 | 94.96 | 91.10 | 39.64 | TYV | CYRILLIC | 93.59 | 84.60 | 84.31 | 92.73 | 93.10 | 92.69 | 1.01 |
| KA | OTHER | 68.86 | 42.20 | 21.36 | 82.89 | 11.51 | 72.25 | 0.74 | UG | ARABIC | 75.79 | 74.20 | 69.27 | 71.95 | 77.97 | 52.82 | 0.00 |
| KK | CYRILLIC | 89.90 | 89.48 | 81.93 | 91.06 | 85.47 | 86.98 | 0.97 | UK | CYRILLIC | 97.10 | 98.15 | 97.40 | 98.69 | 98.30 | 88.11 | 1.92 |
| KM | SANSKRIT | 52.23 | 56.63 | 50.34 | 51.34 | 54.17 | 46.92 | 10.93 | UR | ARABIC | 64.76 | 42.15 | 41.16 | 53.80 | 61.19 | 70.54 | 0.00 |
| KN | SANSKRIT | 49.18 | 37.15 | 36.69 | 72.61 | 38.29 | 48.43 | 7.61 | UZ | LATIN | 95.29 | 96.20 | 94.56 | 95.60 | 93.14 | 91.02 | 32.60 |
| KO | CHINESE | 99.25 | 98.96 | 98.73 | 98.94 | 99.22 | 94.36 | 23.06 | VEC | LATIN | 96.25 | 94.07 | 90.51 | 96.49 | 93.84 | 93.65 | 0.00 |
| KRC | CYRILLIC | 98.03 | 95.41 | 89.64 | 97.75 | 97.51 | 86.72 | 0.46 | VI | LATIN | 96.74 | 95.67 | 96.73 | 94.99 | 86.04 | 91.44 | 33.67 |
| KU | LATIN | 96.87 | 96.86 | 94.26 | 99.03 | 98.30 | 96.26 | 36.25 | WA | LATIN | 96.37 | 89.86 | 94.45 | 98.16 | 81.62 | 92.22 | 10.00 |
| KV | CYRILLIC | 84.71 | 79.07 | 76.49 | 84.57 | 78.59 | 66.36 | 0.20 | WAR | LATIN | 79.29 | 75.08 | 68.09 | 79.89 | 74.68 | 63.87 | 28.25 |

*Note: HY.: HunyuanOCR, Qw3: Qwen3-VL-2B, Qw2.5: Qwen2.5-VL-3B, DOTS: dots.ocr, PD.: PaddleOCR-VL, DS: DeepSeekOCR,*
*Min.: MinerU2.5.*

*Table 18.* Text evaluation (Part I: wuu - yue).

| ISO | FAMILY | HY. | Qw3 | Qw2.5 | DOTS | PD. | DS | MIN. | ISO | FAMILY | HY. | Qw3 | Qw2.5 | DOTS | PD. | DS | MIN. |
|---|---|---|---|---|---|---|---|---|---|---|---|---|---|---|---|---|---|
| WUU | CHINESE | 90.44 | 73.89 | 68.87 | 90.14 | 70.84 | 79.17 | 24.90 | YI | OTHER | 88.48 | 85.61 | 82.07 | 87.69 | 83.54 | 66.63 | 6.66 |
| XAL | CYRILLIC | 90.79 | 87.79 | 78.50 | 87.70 | 88.91 | 82.11 | 1.84 | YUE | CHINESE | 90.73 | 89.40 | 79.52 | 85.22 | 85.21 | 83.51 | 15.35 |
| XMF | OTHER | 59.68 | 34.49 | 11.80 | 88.08 | 2.00 | 85.45 | 0.67 | | | | | | | | | |

*Note: HY.: HunyuanOCR, Qw3: Qwen3-VL-2B, Qw2.5: Qwen2.5-VL-3B, DOTS: dots.ocr, PD.: PaddleOCR-VL, DS: DeepSeekOCR, Min.: MinerU2.5.*

## B.2.3. EVALUATION ON TABLE SCORE

Focusing on structural analysis, Tables 19 present the TEDS scores for table recognition, highlighting the models' proficiency in reconstructing tabular layouts across the MORE benchmark.

*Table 19.* Table evaluation (Part II: an - war).

| ISO | FAMILY | HY. | Qw3 | Qw2.5 | DOTS | PD. | DS | MIN. | ISO | FAMILY | HY. | Qw3 | Qw2.5 | DOTS | PD. | DS | MIN. |
|---|---|---|---|---|---|---|---|---|---|---|---|---|---|---|---|---|---|
| AN | LATIN | 97.31 | 86.07 | 86.63 | 94.92 | 98.15 | 91.91 | 97.03 | LEZ | CYRILLIC | 24.15 | 58.31 | 22.24 | 0.00 | 22.79 | 11.76 | 0.00 |
| BO | SANSKRIT | 89.05 | 84.12 | 66.67 | 89.28 | 66.83 | 88.75 | 66.86 | LV | LATIN | 77.04 | 47.03 | 53.62 | 47.66 | 95.09 | 47.28 | 47.13 |
| BR | LATIN | 96.56 | 98.12 | 95.76 | 0.00 | 97.79 | 97.77 | 0.00 | MHR | CYRILLIC | 98.18 | 99.31 | 94.15 | 99.31 | 0.00 | 89.32 | 0.00 |
| BS | LATIN | 83.30 | 83.77 | 67.42 | 93.97 | 91.29 | 91.04 | 90.00 | MT | LATIN | 87.36 | 72.65 | 77.10 | 64.92 | 93.15 | 96.04 | 23.88 |
| CE | CYRILLIC | 89.98 | 60.59 | 77.52 | 63.60 | 62.50 | 64.21 | 0.00 | MY | SANSKRIT | 3.57 | 50.94 | 49.60 | 0.00 | 0.00 | -0.37 | 0.00 |
| CEB | LATIN | 37.32 | 11.51 | 12.55 | 11.32 | 12.18 | 88.32 | 0.00 | NDS | LATIN | 82.26 | 69.51 | 43.41 | 74.11 | 72.92 | 73.21 | 61.38 |
| CV | CYRILLIC | 74.93 | 58.20 | 56.83 | 46.75 | 77.34 | 70.84 | 35.61 | NE | SANSKRIT | 98.63 | 95.93 | 97.15 | 0.00 | 97.56 | 0.00 | 0.00 |
| CY | LATIN | 99.72 | 74.55 | 96.90 | 0.00 | 99.72 | 100.00 | 0.00 | NN | LATIN | 92.47 | 69.45 | 44.54 | 60.54 | 43.53 | 46.24 | 0.00 |
| DIQ | LATIN | 36.98 | 36.98 | 36.98 | 36.68 | 36.98 | 36.98 | 0.00 | OC | LATIN | 74.78 | 85.35 | 85.09 | 65.88 | 30.14 | 65.05 | 33.13 |
| EU | LATIN | 91.34 | 73.30 | 57.84 | 43.98 | 72.64 | 93.95 | 83.87 | PA | SANSKRIT | 41.68 | 36.34 | 36.28 | 0.00 | 0.00 | 0.00 | 2.88 |
| GA | LATIN | 74.17 | 66.52 | 70.00 | 0.00 | 72.48 | 0.00 | 0.00 | QU | LATIN | 15.97 | 52.47 | 21.79 | 0.00 | 0.00 | 10.99 | 40.66 |
| GL | LATIN | 85.42 | 85.42 | 85.06 | 85.42 | 85.42 | 85.42 | 85.28 | SD | ARABIC | 90.83 | 88.83 | 19.35 | 0.00 | 0.00 | 67.33 | 0.00 |
| IA | LATIN | 99.97 | 99.22 | 97.77 | 99.97 | 99.88 | 99.86 | 0.00 | SU | LATIN | 99.98 | 99.49 | 98.47 | 79.84 | 79.51 | 79.98 | 0.00 |
| ILO | LATIN | 100.00 | 63.10 | 100.00 | 0.00 | 100.00 | 100.00 | 100.00 | TE | SANSKRIT | 92.99 | 69.35 | 0.00 | 0.00 | 6.78 | 0.00 | 26.61 |
| IT | LATIN | 96.67 | 79.55 | 98.64 | 33.27 | 33.28 | 33.13 | 33.25 | TK | LATIN | 92.94 | 87.08 | 71.13 | 0.00 | 66.21 | 94.18 | 93.12 |
| JBO | LATIN | 98.94 | 83.75 | 98.33 | 0.00 | 99.40 | 95.09 | 50.00 | TL | LATIN | 99.75 | 98.92 | 98.81 | 98.93 | 98.83 | 99.03 | 0.00 |
| KM | SANSKRIT | 40.15 | 6.16 | 8.33 | 0.00 | 1.89 | 0.00 | 5.39 | UZ | LATIN | 93.33 | 41.00 | 77.16 | 0.00 | 89.34 | 85.00 | 0.00 |
| KY | CYRILLIC | 87.44 | 43.17 | 37.79 | 60.84 | 91.63 | 87.74 | 35.73 | VEC | LATIN | 36.45 | 48.18 | 55.05 | 39.90 | 30.81 | 46.83 | 65.23 |
| LB | LATIN | 80.37 | 62.74 | 83.41 | 44.29 | 78.40 | 76.84 | 57.00 | WAR | LATIN | 66.82 | 34.89 | 61.24 | 0.00 | 22.99 | 19.18 | 9.53 |

*Note: HY.: HunyuanOCR, Qw3: Qwen3-VL-2B, Qw2.5: Qwen2.5-VL-3B, DOTS: dots.ocr, PD.: PaddleOCR-VL, DS: DeepSeekOCR, Min.: MinerU2.5.*

## B.2.4. EVALUATION ON READING ORDER

Tables 20 and 21 present a quantitative analysis of reading order performance, measured by the Normalized Edit Distance.

*Table 20.* Reading order evaluation (Part I: af - gn).

| ISO | FAMILY | HY. | Qw3 | Qw2.5 | DOTS | PD. | DS | MIN. | ISO | FAMILY | HY. | Qw3 | Qw2.5 | DOTS | PD. | DS | MIN. |
|---|---|---|---|---|---|---|---|---|---|---|---|---|---|---|---|---|---|
| AF | LATIN | 98.94 | 97.40 | 66.44 | 100.00 | 96.34 | 97.40 | 42.86 | CKB | ARABIC | 100.00 | 7.14 | 100.00 | 100.00 | 92.86 | 100.00 | 0.00 |
| ALS | LATIN | 100.00 | 100.00 | 91.67 | 95.00 | 100.00 | 95.00 | 60.00 | CS | LATIN | 100.00 | 95.06 | 97.53 | 100.00 | 100.00 | 100.00 | 100.00 |
| AM | OTHER | 100.00 | 100.00 | 0.00 | 100.00 | 100.00 | 0.00 | 0.00 | CV | CYRILLIC | 88.57 | 86.29 | 71.27 | 100.00 | 72.23 | 91.86 | 66.97 |
| AN | LATIN | 77.78 | 100.00 | 60.61 | 100.00 | 90.91 | 93.33 | 66.67 | CY | LATIN | 88.10 | 100.00 | 97.14 | 100.00 | 100.00 | 81.43 | 62.71 |
| AR | ARABIC | 100.00 | 100.00 | 91.67 | 100.00 | 62.50 | 100.00 | 37.50 | DA | LATIN | 91.67 | 100.00 | 100.00 | 100.00 | 100.00 | 100.00 | 55.56 |
| ARZ | ARABIC | 100.00 | 77.78 | 66.67 | 100.00 | 88.89 | 100.00 | 66.67 | DE | LATIN | 100.00 | 100.00 | 100.00 | 98.75 | 93.33 | 100.00 | 100.00 |
| AST | LATIN | 100.00 | 94.55 | 81.17 | 100.00 | 97.40 | 90.78 | 71.43 | DIQ | LATIN | 100.00 | 100.00 | 85.71 | 100.00 | 100.00 | 100.00 | 100.00 |
| AV | CYRILLIC | 100.00 | 100.00 | 46.15 | 100.00 | 100.00 | 100.00 | 30.77 | DV | OTHER | 100.00 | 100.00 | 100.00 | 100.00 | 100.00 | 100.00 | 0.00 |
| AZ | LATIN | 100.00 | 90.48 | 100.00 | 100.00 | 100.00 | 80.95 | 67.35 | EL | OTHER | 91.67 | 85.42 | 69.79 | 100.00 | 89.58 | 100.00 | 64.24 |
| AZB | ARABIC | 100.00 | 100.00 | 100.00 | 100.00 | 50.00 | 100.00 | 50.00 | EO | LATIN | 100.00 | 100.00 | 87.50 | 100.00 | 100.00 | 100.00 | 100.00 |
| BA | CYRILLIC | 97.58 | 88.89 | 86.19 | 94.20 | 96.33 | 91.88 | 64.53 | ES | LATIN | 100.00 | 100.00 | 89.05 | 100.00 | 100.00 | 100.00 | 100.00 |
| BAR | LATIN | 100.00 | 100.00 | 100.00 | 100.00 | 100.00 | 100.00 | 100.00 | ET | LATIN | 100.00 | 94.54 | 82.04 | 100.00 | 96.13 | 94.64 | 50.00 |
| BE | CYRILLIC | 100.00 | 100.00 | 100.00 | 100.00 | 100.00 | 100.00 | 77.78 | EU | LATIN | 100.00 | 100.00 | 85.71 | 100.00 | 97.40 | 100.00 | 14.29 |
| BG | CYRILLIC | 100.00 | 93.65 | 83.33 | 90.48 | 100.00 | 100.00 | 32.38 | FA | ARABIC | 94.44 | 100.00 | 98.15 | 96.30 | 100.00 | 96.30 | 37.04 |
| BN | SANSKRIT | 100.00 | 100.00 | 50.00 | 100.00 | 100.00 | 100.00 | 100.00 | FI | LATIN | 91.36 | 100.00 | 85.71 | 97.67 | 99.34 | 95.02 | 55.65 |
| BR | LATIN | 81.67 | 90.00 | 70.83 | 90.00 | 95.83 | 96.88 | 50.00 | FR | LATIN | 100.00 | 97.40 | 97.40 | 100.00 | 100.00 | 100.00 | 71.43 |
| BS | LATIN | 96.43 | 82.14 | 85.36 | 96.43 | 100.00 | 94.57 | 79.17 | FY | LATIN | 100.00 | 90.60 | 100.00 | 96.15 | 91.67 | 94.44 | 96.15 |
| BXR | CYRILLIC | 100.00 | 100.00 | 93.75 | 100.00 | 100.00 | 93.75 | 50.00 | GA | LATIN | 100.00 | 88.89 | 91.67 | 100.00 | 94.44 | 94.44 | 83.33 |
| CA | LATIN | 84.90 | 94.29 | 76.73 | 100.00 | 94.29 | 97.14 | 51.43 | GD | LATIN | 94.16 | 81.15 | 75.54 | 96.43 | 92.86 | 93.90 | 75.00 |
| CE | CYRILLIC | 100.00 | 90.58 | 90.58 | 89.45 | 96.59 | 100.00 | 55.64 | GL | LATIN | 98.94 | 100.00 | 97.88 | 100.00 | 98.94 | 100.00 | 42.86 |
| CEB | LATIN | 100.00 | 83.33 | 100.00 | 100.00 | 85.71 | 88.89 | 57.22 | GN | LATIN | 100.00 | 100.00 | 0.00 | 100.00 | 100.00 | 100.00 | 100.00 |

*Note: HY.: HunyuanOCR, Qw3: Qwen3-VL-2B, Qw2.5: Qwen2.5-VL-3B, DOTS: dots.ocr, PD.: PaddleOCR-VL, DS: DeepSeekOCR, Min.: MinerU2.5.*

*Table 21.* Reading order evaluation (Part II: gom - yue).

| ISO | FAMILY | HY. | Qw3 | Qw2.5 | DOTS | PD. | DS | MIN. | ISO | FAMILY | HY. | Qw3 | Qw2.5 | DOTS | PD. | DS | MIN. |
|---|---|---|---|---|---|---|---|---|---|---|---|---|---|---|---|---|---|
| GOM | SANSKRIT | 100.00 | 100.00 | 100.00 | 100.00 | 100.00 | 100.00 | 100.00 | NL | LATIN | 100.00 | 100.00 | 70.50 | 100.00 | 91.94 | 95.00 | 100.00 |
| GU | SANSKRIT | 100.00 | 100.00 | 66.67 | 66.67 | 100.00 | 33.33 | 66.67 | NN | LATIN | 97.50 | 96.67 | 88.46 | 95.83 | 95.67 | 96.43 | 80.00 |
| GV | LATIN | 100.00 | 100.00 | 0.00 | 100.00 | 100.00 | 100.00 | 100.00 | NO | LATIN | 100.00 | 100.00 | 87.50 | 100.00 | 100.00 | 100.00 | 87.50 |
| HE | OTHER | 100.00 | 75.00 | 25.00 | 87.50 | 75.00 | 75.00 | 25.00 | OC | LATIN | 86.67 | 60.00 | 95.00 | 96.67 | 75.00 | 86.67 | 83.33 |
| HI | SANSKRIT | 100.00 | 100.00 | 100.00 | 100.00 | 100.00 | 66.67 | 66.67 | OR | SANSKRIT | 25.00 | 100.00 | 0.00 | 75.00 | 25.00 | 75.00 | 0.00 |
| HR | LATIN | 99.33 | 99.33 | 87.50 | 99.33 | 99.33 | 100.00 | 90.00 | OS | CYRILLIC | 100.00 | 100.00 | 100.00 | 100.00 | 100.00 | 100.00 | 100.00 |
| HSB | LATIN | 100.00 | 100.00 | 100.00 | 100.00 | 100.00 | 98.00 | 60.00 | PA | SANSKRIT | 100.00 | 75.00 | 75.00 | 100.00 | 75.00 | 50.00 | 25.00 |
| HT | LATIN | 100.00 | 100.00 | 93.75 | 100.00 | 95.83 | 100.00 | 75.00 | PAM | LATIN | 80.00 | 80.00 | 28.57 | 80.00 | 71.43 | 71.43 | 0.00 |
| HU | LATIN | 100.00 | 98.57 | 100.00 | 98.57 | 100.00 | 100.00 | 90.00 | PL | LATIN | 100.00 | 96.83 | 66.67 | 100.00 | 97.53 | 100.00 | 66.67 |
| HY | OTHER | 66.67 | 55.56 | 43.33 | 91.67 | 71.67 | 75.00 | 10.00 | PMS | LATIN | 100.00 | 83.33 | 68.89 | 88.89 | 100.00 | 100.00 | 33.33 |
| IA | LATIN | 100.00 | 100.00 | 96.43 | 100.00 | 100.00 | 100.00 | 35.71 | PNB | ARABIC | 0.00 | 100.00 | 100.00 | 100.00 | 100.00 | 100.00 | 0.00 |
| ID | LATIN | 100.00 | 100.00 | 85.71 | 100.00 | 100.00 | 100.00 | 71.43 | PS | ARABIC | 100.00 | 100.00 | 0.00 | 100.00 | 100.00 | 100.00 | 0.00 |
| IE | LATIN | 100.00 | 0.00 | 100.00 | 100.00 | 100.00 | 100.00 | 100.00 | PT | LATIN | 100.00 | 100.00 | 88.89 | 95.83 | 100.00 | 100.00 | 66.67 |
| ILO | LATIN | 96.30 | 88.72 | 86.30 | 92.59 | 91.75 | 96.30 | 55.05 | QU | LATIN | 88.89 | 72.22 | 88.89 | 88.89 | 100.00 | 68.52 | 33.33 |
| IO | LATIN | 100.00 | 100.00 | 100.00 | 100.00 | 100.00 | 100.00 | 100.00 | RM | LATIN | 100.00 | 100.00 | 100.00 | 100.00 | 100.00 | 100.00 | 0.00 |
| IS | LATIN | 100.00 | 100.00 | 90.00 | 100.00 | 100.00 | 100.00 | 100.00 | RO | LATIN | 100.00 | 100.00 | 98.81 | 100.00 | 100.00 | 100.00 | 98.75 |
| IT | LATIN | 100.00 | 87.76 | 100.00 | 100.00 | 100.00 | 97.96 | 85.71 | RU | CYRILLIC | 100.00 | 99.19 | 100.00 | 100.00 | 100.00 | 86.67 | 76.81 |
| JA | CHINESE | 86.67 | 86.67 | 63.33 | 86.67 | 86.67 | 80.00 | 80.95 | SA | SANSKRIT | 100.00 | 100.00 | 28.57 | 100.00 | 100.00 | 71.43 | 14.29 |
| JBO | LATIN | 100.00 | 100.00 | 50.00 | 100.00 | 75.00 | 83.33 | 75.00 | SAH | CYRILLIC | 91.51 | 93.21 | 90.90 | 96.30 | 93.83 | 93.52 | 19.44 |
| JV | LATIN | 93.75 | 100.00 | 62.85 | 100.00 | 100.00 | 100.00 | 62.50 | SD | ARABIC | 100.00 | 100.00 | 100.00 | 100.00 | 0.00 | 100.00 | 100.00 |
| KA | OTHER | 100.00 | 89.29 | 35.12 | 100.00 | 80.36 | 82.14 | 0.00 | SH | CYRILLIC | 98.69 | 96.08 | 98.69 | 97.39 | 100.00 | 100.00 | 100.00 |
| KK | CYRILLIC | 95.88 | 85.71 | 82.86 | 98.53 | 99.16 | 87.69 | 57.11 | SK | LATIN | 100.00 | 80.00 | 87.31 | 100.00 | 100.00 | 100.00 | 62.50 |
| KM | SANSKRIT | 83.33 | 66.67 | 50.00 | 33.33 | 66.67 | 66.67 | 33.33 | SL | LATIN | 100.00 | 100.00 | 97.04 | 100.00 | 100.00 | 100.00 | 88.89 |
| KN | SANSKRIT | 66.67 | 54.17 | 50.00 | 0.00 | 0.00 | 0.00 | 0.00 | SO | LATIN | 94.67 | 92.00 | 72.00 | 100.00 | 100.00 | 88.95 | 60.00 |
| KO | CHINESE | 100.00 | 100.00 | 92.86 | 100.00 | 100.00 | 100.00 | 49.05 | SQ | LATIN | 97.86 | 98.57 | 89.05 | 97.86 | 98.57 | 98.57 | 71.43 |
| KRC | CYRILLIC | 98.72 | 93.36 | 82.14 | 98.72 | 100.00 | 87.50 | 25.16 | SU | LATIN | 100.00 | 100.00 | 100.00 | 100.00 | 100.00 | 100.00 | 66.67 |
| KU | LATIN | 100.00 | 97.40 | 100.00 | 100.00 | 100.00 | 100.00 | 71.43 | SV | LATIN | 100.00 | 100.00 | 100.00 | 100.00 | 100.00 | 100.00 | 88.89 |
| KV | CYRILLIC | 100.00 | 100.00 | 62.00 | 100.00 | 99.00 | 100.00 | 47.43 | SW | LATIN | 100.00 | 100.00 | 85.71 | 100.00 | 100.00 | 100.00 | 96.43 |
| KW | LATIN | 83.33 | 66.67 | 56.25 | 91.67 | 66.67 | 91.67 | 100.00 | TA | SANSKRIT | 100.00 | 100.00 | 66.67 | 100.00 | 100.00 | 100.00 | 33.33 |
| KY | CYRILLIC | 90.74 | 82.22 | 69.44 | 90.74 | 90.74 | 90.37 | 73.61 | TE | SANSKRIT | 100.00 | 100.00 | 75.00 | 75.00 | 100.00 | 100.00 | 25.00 |
| LA | LATIN | 100.00 | 100.00 | 100.00 | 100.00 | 100.00 | 100.00 | 97.22 | TG | CYRILLIC | 100.00 | 100.00 | 77.78 | 100.00 | 100.00 | 92.26 | 51.33 |
| LB | LATIN | 100.00 | 100.00 | 100.00 | 100.00 | 100.00 | 100.00 | 66.67 | TH | SANSKRIT | 100.00 | 95.24 | 57.14 | 100.00 | 100.00 | 100.00 | 57.14 |
| LEZ | CYRILLIC | 92.38 | 82.22 | 47.78 | 97.78 | 100.00 | 69.21 | 33.33 | TK | LATIN | 100.00 | 100.00 | 100.00 | 100.00 | 100.00 | 100.00 | 80.00 |
| LI | LATIN | 100.00 | 100.00 | 88.89 | 100.00 | 100.00 | 100.00 | 100.00 | TL | LATIN | 97.22 | 97.22 | 95.83 | 100.00 | 88.89 | 96.30 | 44.44 |
| LMO | LATIN | 100.00 | 100.00 | 60.00 | 98.46 | 100.00 | 100.00 | 93.85 | TR | LATIN | 100.00 | 100.00 | 95.63 | 100.00 | 100.00 | 100.00 | 85.71 |
| LO | SANSKRIT | 100.00 | 100.00 | 100.00 | 0.00 | 100.00 | 100.00 | 0.00 | TT | CYRILLIC | 100.00 | 90.61 | 100.00 | 97.78 | 97.06 | 97.78 | 45.79 |
| LT | LATIN | 100.00 | 98.75 | 98.75 | 100.00 | 100.00 | 100.00 | 100.00 | TYV | CYRILLIC | 100.00 | 100.00 | 58.33 | 100.00 | 100.00 | 100.00 | 0.00 |
| LV | LATIN | 98.47 | 86.03 | 84.62 | 100.00 | 91.91 | 94.74 | 49.37 | UG | ARABIC | 100.00 | 100.00 | 50.00 | 50.00 | 75.00 | 100.00 | 0.00 |
| MG | LATIN | 93.65 | 84.39 | 82.91 | 100.00 | 96.83 | 93.65 | 66.67 | UK | CYRILLIC | 88.89 | 88.89 | 74.33 | 100.00 | 100.00 | 100.00 | 72.80 |
| MHR | CYRILLIC | 74.33 | 92.67 | 54.00 | 92.67 | 85.00 | 61.33 | 2.67 | UZ | LATIN | 100.00 | 97.48 | 100.00 | 100.00 | 97.06 | 100.00 | 64.29 |
| MK | CYRILLIC | 100.00 | 89.92 | 85.99 | 88.24 | 93.28 | 100.00 | 65.43 | VEC | LATIN | 100.00 | 100.00 | 100.00 | 100.00 | 100.00 | 100.00 | 50.00 |
| ML | SANSKRIT | 33.33 | 83.33 | 16.67 | 100.00 | 83.33 | 100.00 | 33.33 | VI | LATIN | 97.62 | 97.62 | 59.37 | 97.62 | 100.00 | 97.14 | 40.48 |
| MN | CYRILLIC | 100.00 | 100.00 | 60.58 | 100.00 | 97.92 | 100.00 | 58.93 | WA | LATIN | 100.00 | 100.00 | 100.00 | 100.00 | 100.00 | 100.00 | 0.00 |
| MR | SANSKRIT | 96.43 | 79.46 | 42.86 | 91.96 | 89.82 | 100.00 | 43.15 | WAR | LATIN | 82.23 | 77.11 | 69.79 | 89.46 | 79.24 | 82.05 | 43.30 |
| MS | LATIN | 100.00 | 90.91 | 70.91 | 92.73 | 100.00 | 100.00 | 95.56 | WUU | CHINESE | 100.00 | 100.00 | 61.98 | 100.00 | 100.00 | 95.00 | 44.76 |
| MT | LATIN | 100.00 | 100.00 | 92.59 | 92.59 | 100.00 | 92.59 | 33.33 | XAL | CYRILLIC | 92.59 | 77.78 | 77.78 | 77.78 | 77.78 | 77.78 | 100.00 |
| MZN | ARABIC | 100.00 | 100.00 | 100.00 | 100.00 | 98.82 | 100.00 | 100.00 | XMF | OTHER | 93.33 | 18.20 | 22.20 | 100.00 | 84.94 | 100.00 | 44.00 |
| NDS | LATIN | 100.00 | 100.00 | 100.00 | 100.00 | 100.00 | 100.00 | 50.00 | YI | OTHER | 69.19 | 69.19 | 30.00 | 80.00 | 52.97 | 77.30 | 31.35 |
| NE | SANSKRIT | 100.00 | 90.00 | 75.62 | 93.75 | 100.00 | 86.40 | 65.37 | YUE | CHINESE | 100.00 | 66.67 | 100.00 | 100.00 | 100.00 | 100.00 | 66.67 |
| NEW | SANSKRIT | 100.00 | 84.38 | 71.88 | 90.62 | 100.00 | 93.75 | 100.00 | | | | | | | | | |

*Note: HY.: HunyuanOCR, Qw3: Qwen3-VL-2B, Qw2.5: Qwen2.5-VL-3B, DOTS: dots.ocr, PD.: PaddleOCR-VL, DS: DeepSeekOCR, Min.: MinerU2.5.*

## C. Qualitative Visualizations

We visualize model predictions on challenging elements, specifically formulas, tables, code, and catalogs.

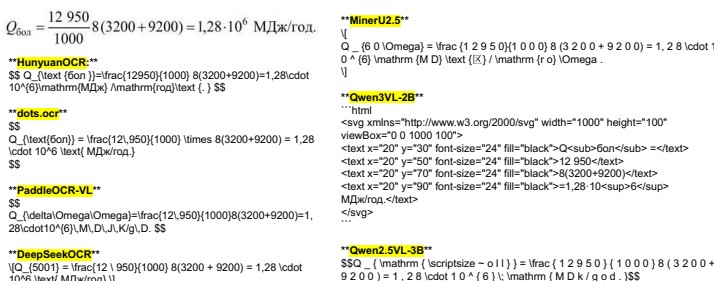

*Figure 8.* Model predictions on multilingual formula in MORE.

| | В %/о к предыд. квар-талу | | В %/о к соотв. кварт. прелыд. года | |
|---|---|---|---|---|
| | 3 кв. 1926-27 г. | 3 кв. 1925-26 г. | 3 кв. 1926-27 г. | 3 кв. 1925-26 г. |
| Сельхозналог (поступление в кассы НКФ) . . . . . . . | 8,9 | 49,2 | 31,2 | 443,6 |
| Остальные доходы гос.-и мест-бюджета . . . . . . . . | 105,5 | 116,1 | 126,6 | 139,0 |

**HunyuanOCR:**

<table><tr><td rowspan="3"></td><td colspan="2">В % к предыд. квар-талу</td><td colspan="2">В % к соотв. кварт. предыд. года</td></tr><tr><td>3 кв. 1926-27 г.</td><td>3 кв. 1925-26 г.</td><td>3 кв. 1926-27 г.</td><td>3 кв. 1925-26 г.</td></tr><tr><td>Сельхозналог (поступление в кассы НКФ) ......</td><td>8,9</td><td>49,2</td><td>31,2</td><td>443,6</td></tr><tr><td>Остальные доходы гос.-и мест-бюджета ......</td><td>105,5</td><td>146,1</td><td>126,6</td><td>139,0</td></tr></table>

**dots.ocr**

<table><thead><tr><th rowspan="2"></th><th colspan="2">В % к предыд. кварталу</th><th colspan="2">В % к соотв. кварт. предыд. года</th></tr><tr><th>3 кв. 1926-27 г.</th><th>3 кв. 1925-26 г.</th><th>3 кв. 1926-27 г.</th><th>3 кв. 1925-26 г.</th></tr></thead><tbody><tr><td>Сельхозналог (поступление в кассы НКФ)</td><td>8,9</td><td>49,2</td><td>31,2</td><td>443,6</td></tr><tr><td>Остальные доходы гос.-и мест-бюджета</td><td>105,5</td><td>116,1</td><td>126,6</td><td>139,0</td></tr></tbody></table>

**PaddleOCR-VL**

<table border=1 style='margin: auto; width: max-content;'><tr><td rowspan="2"></td><td colspan="2">В 0/0 к предыд. кварталу</td><td colspan="2">В 0/0 к соотв. кварт. препыт. года</td></tr><tr><td style='text-align: center;'>3 кв. 1926-27 г.</td><td style='text-align: center;'>3 кв. 1925-26 г.</td><td style='text-align: center;'>3 кв. 1926-27 г.</td><td style='text-align: center;'>3 кв. 1925-26 г.</td></tr><tr><td style='text-align: center;'>Сельхозвалог (поступление в кассы НКФ)</td><td style='text-align: center;'>8,9</td><td style='text-align: center;'>49,2</td><td style='text-align: center;'>31,2</td><td style='text-align: center;'>443,6</td></tr><tr><td style='text-align: center;'>Остальные походы гос.-и мест-бюджета</td><td style='text-align: center;'>105,5</td><td style='text-align: center;'>116,1</td><td style='text-align: center;'>126,6</td><td style='text-align: center;'>139,0</td></tr></table>

**DeepSeekOCR**

<table><tr><td colspan="2">В 0/0 к предыд. квар-
талу</td><td colspan="2">В 0/0 к соотв. кварт. 
предыд. года</td></tr><tr><td>3 кв.
1926-27 г.</td><td>3 кв.
1925-26 г.</td><td>3 кв.
1926-27 г.</td><td>3 кв.
1925-26 г.</td></tr><tr><td>Сельхозналог (поступление в
кассы НКФ)</td><td>8,9</td><td>49,2</td><td>31,2</td><td>443,6</td></tr><tr><td>Остальные доходы гос.-и мест-
бюджета</td><td>105,5</td><td>116,1</td><td>126,6</td><td>139,0</td></tr></table>

**MinerU2.5**

Cebixxobraor (noctynlenee В KacchНКФ)
OCTaJIbHbIe IOXoJbI Roc.-N MeCt-6IOJKeTa
<table><tr><td colspan="2">В 0/0 к пешил. Кразу
Талу</td><td colspan="2">В 0/0 к соотв. Кразу.
празу. Рда</td></tr><tr><td>3 КВ.
1926-27 г.</td><td>3 КВ.
1925-26 г.</td><td>3 КВ.
1926-27 г.</td><td>3 КВ.
1925-26 г.</td></tr><tr><td>8,9</td><td>49,2</td><td>31,2</td><td>443,6</td></tr><tr><td>105,5</td><td>116,1</td><td>126,6</td><td>139,0</td></tr></table>

**Qwen3VL-2B**
```html
<table border="1" class="dataframe">
<thead>
<tr style="text-align: right;">
<th></th>
<th colspan="2">В %/к предыд. кварталу</th>
<th colspan="2">В %/к соотн. кварт. прелыд. года</th>
</tr>
<tr>
<th></th>
<th>3 кв. 1926-27 г.</th>
<th>3 кв. 1925-26 г.</th>
<th>2 кв. 1926-27 г.</th>
<th>3 кв. 1925-26 г.</th>
</tr>
</thead>
<tbody>
<tr>
<td>Сельхозвалог (поступление в кассы НКФ)</td>
<td>8,9</td>
<td>49,2</td>
<td>31,2</td>
<td>443,6</td>
</tr>
<td>Остальные доходы гос.-и мест- бюджета</td>
<td>105,5</td>
<td>116,1</td>
<td>126,6</td>
<td>139,0</td>
</tr>
</tbody>
</table>
```

**Qwen2.5VL-3B**
```html
<table>
<tr>
<td></td>
<th rowspan="2">3 kv.</th>
<th rowspan="2">3 kv.</th>
<th rowspan="2">2 kv.</th>
<th rowspan="2">3 kv.</th>
</tr>
<tr>
<td>1926-27</td>
<td>1925-26</td>
<td>1926-27</td>
<td>1925-26</td>
</tr>
<tr>
<td>Сельхозналог (поступление в кассы НКФ)</td>
<td>8.9</td>
<td>49.2</td>
<td>31.2</td>
<td>443.6</td>
</tr>
<td>Остальные доходы гос.-и мест- бюджета</td>
<td>105.5</td>
<td>116.1</td>
<td>126.6</td>
<td>139.0</td>
</tr>
</table>
```

*Figure 9.* Model predictions on multilingual table in MORE.

```java
5  public class LogoExample {
6
7⊖     public static void main(String[] args) {
8          UserStorage task = new UserStorage();
9          Scanner sc = new Scanner(System.in);
10         System.out.print("Введите логин > ");
11         String login = sc.next();
12         System.out.print("Введите пароль > ");
13         String pwd = sc.next();
14         UserInfo user = task.getUser(new AuthorizeInfo(login, pwd));
15         if(user != null)
16         {
17             System.out.println(user.fio);
18             System.out.println(user.city);
19             System.out.println(user.mail);
20         }
21         else
22             System.out.println("нет такого пользователя!");
23
24     }
25  }
```

**HunyuanOCR:**
```java
public class LogoExample {

public static void main(String[] args) {

UserStorage task = new UserStorage();
Scanner sc = new Scanner(System.in);
System.out.print("Введите логин > ");
String login = sc.next();
System.out.print("Введите пароль > ");
String pwd = sc.next();
UserInfo user = task.getUser(new AuthorizeInfo(login, pwd));
if(user != null)
{
System.out.println(user.fio);
System.out.println(user.city);
System.out.println(user.mail);
}
else
System.out.println("нет такого пользователя!");

}
}
```

**dots.ocr**
```java
public class LogoExample {
public static void main(String[] args) {
UserStorage task = new UserStorage();
Scanner sc = new Scanner(System.in);
System.out.print("Введите логин > );

String login = sc.next();
System.out.print("Введите пароль > );

String pwd = sc.next();
UserInfo user = task.getUser(new AuthorizeInfo(login, pwd));
if (user != null)
{
System.out.println(user.fio);
System.out.println(user.city);
System.out.println(user.mail);
}
else
System.out.println("нет такого пользователя");
}
}
```

**PaddleOCR-VL**
public class LogoExample {

public static void main(String[] args) {

UserStorage task = new UserStorage();

Scanner sc = new Scanner(System.in);

System.out.print("Введите логин >");

String login = sc.next();

System.out.print("Введите пароль >");

String pwd = sc.next();

UserInfo user = task.getUser(new AuthorizeInfo(login, pwd));

if (user != null) {

System.out.println(user.fio);

System.out.println(user.city);

System.out.println(user.mail);

} else {

System.out.println("нет такого пользователя");

}

}

**DeepSeekOCR**
public class LogoExample {
public static void main(String[] args) {
UserStorage task = new UserStorage();
Scanner sc = new Scanner(System.in);
System.out.print("Введите логин > ");
String login = sc.next();
System.out.print("Введите пароль > ");
String pwd = sc.next();
UserInfo user = task.getUser(new AuthorizeInfo(login, pwd));
if(user != null)
{
System.out.println(user.fio);
System.out.println(user.city);
System.out.println(user.mail);
}
else
System.out.println("Нет такого пользователя!");
}

**MinerU2.5**
5 public class LogoExample {
6
7 public static void main(String[] args) {
8 UserStorage task = new UserStorage();
9 Scanner sc = new Scanner(System.in);
10 System.out.print("В 数и логин > ");
11 String login = sc.next();
12 System.out.print("В 数и паров > ");
13 String pwd = sc.next();
14 UserInfo user = task.getUser(newauthorizeInfo(login, pwd));
15 if (user != null) {
16 System.out.println(user.fio);
17 System.out.println(user.city);
18 System.out.println(user.mail);
19 }
20 else
21 System.out.println("Нет Такого польбователя!");
22 }
23

**Qwen3VL-2B**
```java
public class LogoExample {

public static void main(String[] args) {
UserStorage task = new UserStorage();
Scanner sc = new Scanner(System.in);
System.out.print("Введите логин > ");
String login = sc.next();
System.out.print("Введите пароль > ");
String pwd = sc.next();
UserInfo user = task.getUser(new AuthorizeInfo(login, pwd));
if (user != null) {
System.out.println(user.fio);
System.out.println(user.city);
System.out.println(user.mail);
} else {
System.out.println("нет такого пользователя!");
}
}
}
```

**Qwen2.5VL-3B**
```java
public class LogoExample {
public static void main(String[] args) {
UserStorage task = new UserStorage();
Scanner sc = new Scanner(System.in);
System.out.print("Введите логин > ");
String login = sc.next();
System.out.print("Введите пароль > ");
String pwd = sc.next();
UserInfo user = task.getUser(new AuthorizeInfo(login, pwd));
if (user != null) {
System.out.println(user.fio);
System.out.println(user.city);
System.out.println(user.mail);
} else
System.out.println("нет такого пользователя!");
}
}
```

*Figure 10.* Model predictions on multilingual code in MORE.

Ανδρόνικος Δ΄ Παλαιολόγος 1376-1379...............67-68
Ιωάννης Ε΄ Παλαιολόγος 1379-1391.............…..........68
Μανουήλ Β΄ Παλαιολόγος 1391-1399........................69
Ιωάννης Ζ΄ Παλαιολόγος 1399-1402..........................69
Μανουήλ Β΄ Παλαιολόγος 1402-1425..................69-70
Ιωάννης Η΄ Παλαιολόγος 1425-1448.................70-71
Κωνσταντίνος ΙΑ΄ Παλαιολόγος 1448-1453...........71-72
Βυζαντινοί αυτοκράτορες της Κωνσταντινούπολης
    1261-1453.................................................................73
Τα νομίσματα της ύστερης βυζαντινής περιόδου.....74-80
Η οικονομία της ύστερης βυζαντινής περιόδου..........81-89
Τα γεγονότα του εμφυλίου πολέμου
    Ανδρόνικου Β΄-Ανδρόνικου Γ΄ Παλαιολόγων
    (1321-1328)............................................…...90-122
Τα γεγονότα του εμφυλίου πολέμου
    Ιωάννη Ε΄ Παλαιολόγου-Ιωάννη Καντακουζηνού
    (1341-1347)......................................…...123-166
Η άλωση της Κωνσταντινούπολης 1453................…....167-172
Τα Πανεπιστήμια των Παλαιολόγων..................…....173-174
Βυζαντινοί λόγιοι στη Δύση...............................175-176
Υστεροβυζαντινά νομίσματα.............................177-180

**HunyuanOCR:**
Ανδρόνικος Δ΄ Παλαιολόγος 1376-1379.............67-68
Ιωάννης Ε΄ Παλαιολόγος 1379-1391.............68
Μανουήλ Β΄ Παλαιολόγος 1391-1399.............69
Ιωάννης Ζ΄ Παλαιολόγος 1399-1402.............69
Μανουήλ Β΄ Παλαιολόγος 1402-1425.............69-70
Ιωάννης Η΄ Παλαιολόγος 1425-1448.............70-71
Κονσταντίνος ΙΑ' Παλαιολόγος 1448-1453.............71-72
Βυζαντινοί αυτοκράτορες της Κονσταντινούπολης 1261-1453.............73
Τα νομίσματα της ύστερης βυζαντινής περιόδου.............74-80
Η οικονομία της ύστερης βυζαντινής περιόδου.............81-89
Τα γεγονότα του εμφυλίου πολέμου Ανδρόνικου Β'-Ανδρόνικου Γ' Παλαιολόγων (1321-1328).............90-122
Τα γεγονότα του εμφυλίου πολέμου Ιωάννα Ε' Παλαιολόγου-Ιωάννη Καντακουζηνού (1341-1347).............123-166
Η άλωση της Κονσταντινούπολης 1453.............167-172
Τα Πανεπιστήμια των Παλαιολόγων.............173-174
Βυζαντινοί λόγιοι στη Δύση.............175-176
Υστεροβυζαντινά νομίσματα.............177-180

**MinerU2.5**
Avopovko \(\Delta^{\prime}\) Palaiooouoc 1376-1379. .67-68
Iawvns E' Palaioooyoc 1379-1391. .68
MavounB' Palaiooouoc 1391-1399. .69
Iawvns Z' Palaioooyoc 1399-1402. .69
MavounB' Palaiooouoc 1402-1425. .69-70
Iawvns H' Palaioooyoc 1425-1448. .70-71
Kovotavtivoc IA' Palaioooouoc 1448-1453. .71-72
Buavtroi avokpatope ts Kowotavtvouno 1261-1453. .73
Ta voiuia tns osteepn Bucavtvnc πepiodov.. .74-80
H oukovua tns osteepn buavtvnc πepiodov.. .81-89
Ta yevovota tov eupuliov noeou Avopovikov B'-Avdpovikov F' Palaiooouov (1321-1328). .90-122
Ta yevovota tov eupuliov noeou Iwavn E' Palaiooouov-Iawn Kavtakounvov (1341-1347). .123-166
Haowon tnc Kowotavtvouno ncs 1453. .167-172
Ta laveaiothma tow Palaiooouov. .173-174
Buavtroi loyoi ot N DsO. .175-176
Yostepucaivva vouia uata. .177-180

**Qwen3VL-2B**
Ανδρόνικος Δ΄ Παλαιολόγος 1376-1379.................67-68
Ιωάννης Ε΄ Παλαιολόγος 1379-1391.................68
Μανουήλ Β΄ Παλαιολόγος 1391-1399.................69
Ιωάννης Ζ΄ Παλαιολόγος 1399-1402.................69
Μανουήλ Β΄ Παλαιολόγος 1402-1425.................69-70
Ιωάννης Η΄ Παλαιολόγος 1425-1448.................70-71
Κωνσταντίνος ΙΑ' Παλαιολόγος 1448-1453.................71-72
Βυζαντινοί αυτοκράτορες της Κωνσταντινούπολης 1261-1453.................73
Τα νομίσματα της ύστερης βυζαντινής περιόδου.................74-80
Η οικονομία της ύστερης βυζαντινής περιόδου.................81-89
Τα γεγονότα του εμφυλίου πολέμου Ανδρόνικος Β'-Ανδρόνικου Γ Παλαιολόγων (1321-1328).................90-122
Τα γεγονότα του εμφυλίου πολέμου Ιωάννη Ε' Παλαιολόγου-Ιωάννη Καντακουζηνού (1341-1347).................123-166
Η άλωση της Κωνσταντινούπολης 1453.................167-172
Τα Πανεπιστήμια των Παλαιολόγων.................173-174
Βυζαντινοί λόγιοι στη Δύση.................175-176
Υστεροβυζαντινά νομίσματα.................177-180

**Qwen2.5VL-3B**
Ανδρόνικος Δ΄ Παλαιολόγος 1376-1379......................67-68
Ιωάννης Ε΄ Παλαιολόγος 1379-1391...........................68
Μανουήλ Β΄ Παλαιολόγος 1391-1399.........................69
Ιωάννης Ζ΄ Παλαιολόγος 1399-1402.........................69
Μανουήλ Β΄ Παλαιολόγος 1402-1425..........................69-70
Ιωάννης Η΄ Παλαιολόγος 1425-1448.......................70-71
Κωνσταντίνος ΙΑ΄ Παλαιολόγος 1448-1453............71-72
Βυζαντινοί αυτοκράτορες της Κωνσταντινούπολης 1261-1453.................73
Τα νομίσματα της ύστερης βυζαντινής περιόδου........74-80
Η οικονομία της ύστερης βυζαντινής περιόδου........81-89
Τα γεγονότα του εμφυλίου πολέμου Ανδρόνικου Β'-Ανδρόνικου Γ΄ Παλαιολόγων (1321-1328).........................90-122
Τα γεγονότα του εμφυλίου πολέμου Ιωάννη Ε΄ Παλαιολόγου-Ιωάννη Καντακουζηνού (1341-1347).........................123-166
Η άλωση της Κωνσταντινούπολης 1453....................167-172
Τα Πανεπιστήμια των Παλαιολόγων......................173-174
Βυζαντινοί λόγιοι στη Δύση...........................175-176
Υστεροβυζαντινά νομίσματα................................177-180

**PaddleOCR-VL**
Ανδρόνικος Δ' Παλαιολόγος 1376-1379.....67-68
Ιωάννης Ε' Παλαιολόγος 1379-1391.....68
Μανουήλ Β' Παλαιολόγος 1391-1399.....69
Ιωάννης Ζ' Παλαιολόγος 1399-1402.....69
Μανουήλ Β' Παλαιολόγος 1402-1425.....69-70
Ιωάννης Η' Παλαιολόγος 1425-1448.....70-71
Κωνσταντίνος ΙΑ' Παλαιολόγος 1448-1453.....71-72
Βυζαντινοί αυτοκράτορες της Κωνσταντινούπολης 1261-1453.....73
Τα νομίσματα της ύστερης βυζαντινής περιόδου.....74-80
Η οικονομία της ύστερης βυζαντινής περιόδου.....81-89
Τα γεγονότα του εμφυλίου πολέμου Ανδρόνικου Β'-Ανδρόνικου Γ' Παλαιολόγων (1321-1328).....90-122
Τα γεγονότα του εμφυλίου πολέμου Ιωάννη Ε' Παλαιολόγου-Ιωάνη Καντακουζηνού (1341-1347).....123-166
Η άλωση της Κωνσταντινούπολης 1453.....167-172
Τα Πανεπιστήμια των Παλαιολόγων.....173-174
Βυζαντινοί λόγιοι στη Δύση.....175-176
Υστεροβυζαντινά νομίσματα.....177-180

**dots.ocr**
```
<table>
<tbody>
<tr>
<td>Ανδρόνικος Δ΄ Παλαιολόγος 1376-1379</td>
<td>67-68</td>
</tr>
<tr>
<td>Ιωάννης Ε΄ Παλαιολόγος 1379-1391</td>
<td>68</td>
</tr>
<tr>
<td>Μανουήλ Β΄ Παλαιολόγος 1391-1399</td>
<td>69</td>
</tr>
<tr>
<td>Ιωάννης Ζ΄ Παλαιολόγος 1399-1402</td>
<td>69</td>
</tr>
<tr>
<td>Μανουήλ Β΄ Παλαιολόγος 1402-1425</td>
<td>69-70</td>
</tr>
<tr>
<td>Ιωάννης Η΄ Παλαιολόγος 1425-1448</td>
<td>70-71</td>
</tr>
<tr>
<td>Κωνσταντίνος ΙΑ΄ Παλαιολόγος 1448-1453</td>
<td>71-72</td>
</tr>
<tr>
<td>Βυζαντινοί αυτοκράτορες της Κωνσταντινούπολης 1261-1453</td>
<td>73</td>
</tr>
<tr>
<td>Τα νομίσματα της ύστερης βυζαντινής περιόδου</td>
<td>74-80</td>
</tr>
<tr>
<td>Η οικονομία της ύστερης βυζαντινής περιόδου</td>
<td>81-89</td>
</tr>
<tr>
<td>Τα γεγονότα του εμφυλίου πολέμου
Ανδρόνικου Β΄-Ανδρόνικου Γ΄ Παλαιολόγων
(1321-1328)</td>
<td>90-122</td>
</tr>
<tr>
<td>Τα γεγονότα του εμφυλίου πολέμου
Ιωάννη Ε΄ Παλαιολόγου-Ιωάννη Καντακουζηνού
(1341-1347)</td>
<td>123-166</td>
</tr>
<tr>
<td>Η άλωση της Κωνσταντινούπολης 1453</td>
<td>167-172</td>
</tr>
<tr>
<td>Τα Πανεπιστήμια των Παλαιολόγων</td>
<td>173-174</td>
</tr>
<tr>
<td>Βυζαντινοί λόγιοι στη Δύση</td>
<td>175-176</td>
</tr>
<tr>
<td>Υστεροβυζαντινά νομίσματα</td>
<td>177-180</td>
</tr>
</tbody>
</table>
```

**DeepSeekOCR**
Ανδρόνικος Δ΄ Παλαιολόγος 1376-1379. . . . . . . . . . . . . . . . . . . . . . . . . . . . . . . . . . . 67-68
Ιωάννης Ε΄ Παλαιολόγος 1379-1391. . . . . . . . . . . . . . . . . . . . . . . . . . . . . . . . . . 68
Μανουήλ Β΄ Παλαιολόγος 1391-1399. . . . . . . . . . . . . . . . . . . . . . . . . . . . . . . . 69
Ιωάννης Ζ΄ Παλαιολόγος 1399-1402. . . . . . . . . . . . . . . . . . . . . . . . . . . . . . . 69
Μανουήλ Β΄ Παλαιολόγος 1402-1425. . . . . . . . . . . . . . . . . . . . . . . . . . . . . . 69-70
Ιωάννης Η΄ Παλαιολόγος 1425-1448. . . . . . . . . . . . . . . . . . . . . . . . . . . . . 70-71
Κωνσταντίνος ΙΑ΄ Παλαιολόγος 1448-1453. . . . . . . . . . . . . . . . . . . . . . . . . . . . . . . 71-72
Βυζαντινοί αυτοκράτορες της Κωνσταντινούπολης 1261-1453. . . . . . . . . . . . . . . . . . . . . . . 73
Τα νομίσματα της ύστερης βυζαντινής περιόδου. . . . . . . . . . . . . . . . . . . . . . . . . . . . . . . . . . . . . 74-80
Η οικονομία της ύστερης βυζαντινής περιόδου. . . . . . . . . . . . . . . . . . . . . . . . . . . . . . . . . . . . . . 81-89
Τα γεγονότα του εμφυλίου πολέμου Ανδρόνικου Β΄-Ανδρόνικου Γ΄ Παλαιολόγων (1321-1328). . . . . . . . . . . . . . . . . . . . . . . . . . . . . . . . . . . 90-122
Τα γεγονότα του εμφυλίου πολέμου Ιωάννη Ε΄ Παλαιολόγου-Ιωάννη Καντακουζηνού (1341-1347). . . . . . . . . . . . . . . . . . . . . . . . . . . . . . . . . . 123-166
Η άλωση της Κωνσταντινούπολης 1453. . . . . . . . . . . . . . . . . . . . . . . . . . . . . . . . . . . 167-172
Τα Πανεπιστήμια των Παλαιολόγων. . . . . . . . . . . . . . . . . . . . . . . . . . . . . . . . . . . 173-174
Βυζαντινοί λόγιοι στη Δύση. . . . . . . . . . . . . . . . . . . . . . . . . . . . . . . . . . 175-176
Υστεροβυζαντινά νομίσματα. . . . . . . . . . . . . . . . . . . . . . . . . . . . . . . . . . 177-180

*Figure 11.* Model predictions on multilingual catalog in MORE.

## D. Prompt Templates and Inference Details

As discussed in our methodology, we adopted a tailored evaluation strategy to ensure both fairness and optimal performance across different model architectures (the specific models are detailed in Appendix E).

**Specialized VLMs**: For specialized document parsing models (including HunyuanOCR, dots.ocr, PaddleOCR-VL, DeepSeekOCR, and MinerU2.5), we strictly utilized the default inference pipelines (e.g., vLLM-based engines) and the official prompts recommended in their respective open-source repositories. This approach ensures that their built-in formatting and structural reasoning capabilities are evaluated exactly as intended by the authors, without any prompt-induced degradation.

**General VLMs**: To evaluate General VLMs (Qwen3-VL, Qwen2.5-VL, Gemini) under identical conditions and ensure a fair comparison, we utilized the exact same set of task-specific prompts officially designed for HunyuanOCR. These instructions explicitly define the expected output formats to standardize the evaluation. Specifically, we used the following prompts (translated here from the original Chinese for formatting compliance): *"Extract the text from the image."* for paragraphs and catalogs, *"Parse the codeblock with markdown format."* for code, *"Recognize the formulas in the image and represent them in LaTeX format."* for formulas, *"Parse the table in the image into HTML."* for tables, and *"Extract all information from the main body of the document image in markdown format, ignoring headers and footers. Express tables in HTML and formulas in LaTeX, organizing the parsed content according to the reading order."* for end-to-end (md2md) parsing.

## E. Details of Evaluated Models

Building upon the inference strategies outlined above, Table 22 provides a comprehensive summary of the primary models evaluated in this benchmark, detailing their parameter sizes, architectural paradigms, and claimed language support.

As observed in our experiments, General VLMs typically employ a decoder-only LLM backbone coupled with a vision encoder, pre-trained on massive multimodal corpora. In contrast, specialized OCR models often utilize parameter-efficient architectures optimized specifically for high-resolution spatial reasoning and complex layout parsing.

*Table 22.* Summary of the evaluated models.

| TYPE | MODEL | PARAMETERS | ARCHITECTURE TYPE | CLAIMED LANGUAGES | OPEN SOURCE |
|---|---|---|---|---|---|
| | PADDLEOCR-VL | 0.9B | VIT + LLM | 109 | ✓ |
| | HUNYUANOCR | 1B | VIT + LLM | 130+ | ✓ |
| SPECIALIZED | MINERU2.5 | 1.2B | VIT + LLM | - | ✓ |
| | DEEPSEEKOCR | 3B (A570M) | VIT + MOE LLM | 100+ | ✓ |
| | DOTS.OCR | 3B | VIT + LLM | 126 | ✓ |
| | QWEN3-VL | 2B | VIT + LLM | 32+ | ✓ |
| GENERAL | QWEN2.5-VL | 3B | VIT + LLM | - | ✓ |
| | GEMINI 3 | - | - | - | ✗ |

