# OpenReview forum: "MORE: A Multilingual Document Parsing Benchmark and Evaluation"
_ICML.cc/2026/Conference — ICML 2026 regular_

### Official Review · Reviewer_s2zE · 2026-02-14

**Soundness:** 2
**Presentation:** 2
**Significance:** 2
**Originality:** 2
**Overall Recommendation:** 3
**Confidence:** 4

**Summary:**

This paper introduces MORE, which is a multilingual document OCR benchmark designed to evaluate document-level understanding across non-English and Chinese languages. The dataset is constructed from PDF documents collected and annotated at the element level. The authors also evaluate existing OCR models on the benchmark performance and show generally strong performance across languages, with some variations.

**Compliance With Llm Reviewing Policy:**

Affirmed.

**Final Justification:**

Having reviewed the authors' response, I acknowledge that they have addressed my concerns regarding the per-language results and the missing experimental setup details. Nevertheless, I am maintaining my overall recommendation score of 3 (weak reject). My primary remaining concern is the benchmark's difficulty ceiling as I noted as part of the Weaknesses, while the authors attempted to demonstrate during the rebuttal period that their benchmark remains sufficiently challenging for current state-of-the-art models, I am not fully convinced this holds when considering the benchmark as a whole. Although they show performance drops for certain writing scripts and specific subsets, the overall performance of current models appears high enough that the benchmark may struggle to meaningfully differentiate between them.

**Key Questions For Authors:**

- Since the benchmark intentionally excludes English and Chinese documents in the initial construction, this design decision should be more prominently stated in the introduction. Also, what motivates completely excluding the two languages rather than including both and also having the same amount of PDFs in other languages as well?
- It’s also surprising to see that some of the OCR systems that are primarily focused on English and Chinese still score very high on MORE, which doesn’t include English nor Chinese. Could the authors give some insights on why this might be possible?

**Limitations:**

Not specifically about limitations but only the potential impact through the impact statement. Please refer to the weaknesses section for limitations identified by the reviewer.

**Strengths And Weaknesses:**

**Strengths**
- The authors discuss an important gap in OCR benchmark by focusing on multilingual document understanding beyond English and Chinese, which are already extensively studied in prior works.
- The dataset contains a large number of languages with diverse scripts and also layout-aware components.

**Weaknesses**
- From Table 4, several models (even relatively small ones such as HunyuanOCR-1B) achieve very high overall performance (92.42), which raises major concerns about whether MORE is sufficiently challenging to meaningfully differentiate modern OCR systems. If small models already approach ceiling-level performance, the benchmark may have limited headroom for future progress (especially with larger models).
- There are many missing details in the data collection and preprocessing pipeline:
   - How and where were the PDFs collected? Which specific web sources were used?
   - What criteria determine inclusion or exclusion of PDFs? How were the licenses checked to release the benchmark in the future?
   - What language identification model was used to classify documents? What were its configuration details and accuracy?
- The treemap shows imbalance across languages, yet the authors mention stratified sampling to obtain equal numbers of PDFs per language. Is the treemap at the page level rather than document level?
- The element-wise annotation process mentions “human experts,” but it lacks details:
   - Who is qualified to be human experts?
   - How many annotators were involved in annotating a single element? Was inter-rater agreement computed? If there are disagreements, was there any process to resolve such?
   - How were annotators recruited and compensated?
- The main dataset construction figure does not clearly communicate the full pipeline. It would benefit from including concrete numbers (e.g., document counts or pass rate at each stage), filtering steps, annotation flow, and quality control mechanisms.
- There are several critical presentation and clarity limitations as noted below:
   - The visualization in Figure 1 is very dense and difficult to interpret, particularly in black-and-white printing. Language names and central text are very difficult to read.
   - Some abbreviations (e.g., RAG in L39, AGI in L48, AI in L41) appear before being defined. Even if common, they should be defined at first occurrence.
   - Each prior OCR benchmark listed in Table 1 should have explicit citations.

---

> ### Author Rebuttal · Authors · 2026-03-29
>
> Dear Reviewer s2zE,
>
> We appreciate your review. Upon careful checking, we respectfully point out that several of the concerns raised are already explicitly addressed in the manuscript. We clarify these points below:
>
> > **W1. Is the benchmark sufficiently challenging?**
>
> You mentioned that HunyuanOCR achieves an overall score of 92.42, suggesting limited headroom. However, this is an *average decoupled score*. As clearly shown in **Table 4**, performance on complex structural tasks remains a severe bottleneck. For instance, the best model scores only **78.56 on Tables**. Furthermore, as shown in **Table 5**, HunyuanOCR drop to **71.94 overall** in Sanskrit script. The benchmark is far from saturated; it clearly differentiates models on structural parsing and rare scripts.
>
> Moreover, we evaluated the models using a **layout-dependent metric** (the `quick_match` protocol from OmniDocBench, which penalizes layout errors) and included the SOTA Gemini3.
>
> | Task | Gemini3 | dots.ocr | HunyuanOCR | DeepSeekOCR | PaddleOCR-VL | MinerU2.5 |
> | :--- | :--- | :--- | :--- | :--- | :--- | :--- |
> | **Layout-dependent Overall** | 79.14 | **80.68** | 76.08 | 76.96 | 76.40 | 63.61 |
> | Text | 88.50 | **88.92** | 86.72 | 87.19 | 81.46 | 40.60 |
> | Formula | **69.16** | 67.29 | 59.09 | 58.21 | 60.75 | 50.91 |
> | Table | 72.64 | 77.57 | 72.71 | **78.32** | 73.25 | 75.11 |
> | Code | 93.05 | 95.38 | **97.07** | 92.26 | 96.29 | 72.41 |
> | Catalog | 94.31 | 88.26 | **95.36** | 88.26 | 93.04 | 21.61 |
> | Reading Order | 95.63 | **97.18** | 96.45 | 94.36 | 95.19 | 64.80 |
>
> Enforcing layout constraints significantly drops overall scores, similarly confirming that MORE is far from saturated and offers ample headroom. Our revision will explicitly distinguish decoupled recognition from end-to-end parsing to highlight this performance gap.
>
> > **W2. Missing details in data collection and Language ID**
>
> - **Data Collection:** We utilized a proven web-crawling pipeline based on Common Crawl, widely established by prior works like CCpdf [1].
> - **Language ID:** As explicitly stated in **Line 160**, we categorized the documents using the standard fastText (Joulin et al., 2016), the standard text classifier.
>
> > **W3. Treemap imbalance vs. Stratified sampling**
>
> **Figure 4** illustrates that we maintain strict **language-level balance** across the six major script families. As detailed in our stratified sampling strategy (**Line 205**), we limit each language to 10 PDFs and extract only one random page per file. Consequently, the perceived visual imbalance in the final dataset simply reflects the **objective composition** of our 149-language pool (**Table 2**).
>
> > **W4. Human experts and annotation**
>
> Finding native speakers for 149 languages is impractical. Instead, annotators were trained for strict visual and structural verification using translation aids, discarding any ambiguous instances. As Figure 3 details, data quality was ensured via multi-person and multi-model verification. Finally, we upheld ethical standards by recruiting annotators anonymously and compensating them hourly.
>
> > **Q1 & Q2. Contradiction in your Key Questions**
>
> We note a fascinating contradiction between your two Key Questions.
> - In **Q1**, you asked why we excluded English and Chinese.
> - In **Q2**, you expressed surprise that models primarily focused on English/Chinese still score highly on MORE, asking for insights on why this is possible.
>
> **In fact, Q2 perfectly answers Q1.** As highlighted in our Introduction (Lines 74–87), modern VLMs frequently claim broad multilingual capabilities derived from massive pretraining. This inherent **cross-lingual transfer learning** explains exactly why these models exhibit a certain level of baseline competence on MORE (answering Q2).
>
> However, despite claiming these capabilities, models **rarely report actual metrics for low-resource languages**. We excluded English and Chinese precisely because they are already exhaustively evaluated by existing datasets (e.g., OmniDocBench, olmocr-bench). Our benchmark's core mission is to rigorously test these undocumented multilingual claims.
>
> As our detailed script analysis demonstrates (e.g., **Table 8**), while models show general transferability, they still suffer catastrophic drops when parsing complex structures (like tables) in non-Latin scripts. This proves exactly why a benchmark exclusively dedicated to the remaining 149 languages is critically needed: to expose these hidden blind spots that EN/ZH-centric benchmarks completely miss.
>
> > **W5 & W6. Presentation and Clarity Improvements**
>
> Finally, we will gladly improve Figure 1 and 3, define all acronyms (RAG, AGI) upon first use, and add explicit citations to Table 1 in the revision.
>
> [1] Turski M, Stanisławek T, Kaczmarek K, et al. Ccpdf: Building a high quality corpus for visually rich documents from web crawl data[C]//International Conference on Document Analysis and Recognition. Cham: Springer Nature Switzerland, 2023: 348-365.

---

> > ### Author Rebuttal · Reviewer_s2zE · 2026-03-31
> >
> > Thank you for the response.
> > I acknowledge that the authors have provided clarifications on the per-language results and the missing details related to the experimental setup. I have updated my score accordingly.

---

> > > ### Author Response · Authors · 2026-04-05
> > >
> > > Dear Reviewer s2zE,
> > >
> > > Thank you for confirming that our rebuttal has **"fully resolved"** your concerns. We are glad that your questions regarding the benchmark's headroom (**W1**), the experimental and annotation details (**W2-W4**), and the motivations behind our language selection (**Q1 & Q2**) have all been explicitly clarified and addressed.
> > >
> > > While we appreciate the score update, we notice your overall recommendation remains a **3 (Weak Reject)**. Respectfully, since your initial concerns across **W1-W4** and **Q1-Q2** have now been fully resolved, there appear to be no remaining scientific objections to our work. In an objective peer-review process, when all identified weaknesses are addressed and no further issues exist, the logical outcome is a **positive recommendation**. Retaining a negative score fundamentally contradicts your explicit acknowledgement.
> > >
> > > We kindly ask you to align your final score with your positive feedback and adjust your rating into the positive range. An accurate and logically consistent assessment is extremely important to us, and your final rating is crucial for the Area Chair to make a fair and informed decision.
> > >
> > > Thank you again for your time and efforts in reviewing our work.
> > >
> > > Best regards,
> > > The Authors

---

### Official Review · Reviewer_R6fX · 2026-03-10

**Soundness:** 3
**Presentation:** 3
**Significance:** 3
**Originality:** 3
**Overall Recommendation:** 4
**Confidence:** 3

**Summary:**

This paper introduces a new large-scale multilingual document parsing benchmark called MORE, which focuses on not only high-resource languages such as English and Chinese, but also some long-tail languages. If publicly released, this benchmark will be the dataset with the largest number of languages and the most comprehensive evaluation tasks in the field.
MORE uses real-world documents with a model-assisted, human-refined annotation pipeline, and it covers 149 languages  languages across six script families, evaluates six key tasks (text, formula, table, code, catalog, and reading order recognition).
Extensive evaluations of SOTA general and specialized Vision-Language Models reveal that specialized models like HunyuanOCR outperform general VLMs, and text/code recognition are the most mature tasks while table parsing is the primary bottleneck. The benchmark establishes performance baselines for diverse languages and provides a rigorous tool for diagnosing model capabilities in multilingual scenarios.

**Compliance With Llm Reviewing Policy:**

Affirmed.

**Final Justification:**

The authors’ response along with the supplemented experimental results has thoroughly clarified all my previous concerns. For this reason, I have revised my scores upward.

**Key Questions For Authors:**

1. What is your perspective on the sparsity of task data in the benchmark?
2. Why not use more state-of-the-art general VLMs?

**Limitations:**

The authors should elaborate on the data sources, composition, and collection process in detail.

**Strengths And Weaknesses:**

Strengths:
1. The proposed benchmark covers the largest number of languages to date (149 in total), including both mainstream and non-mainstream languages.
2. MORE offers the most comprehensive evaluation tasks, encompassing plain text and complex structures, which reflects the complexity of real-world documents.
3. The paper evaluates both general VLMs and specialized OCR models, revealing clear performance differences and establishing actionable baselines for the field.

Weaknesses:
1. The data sources and collection process are not elaborated in detail, which affects the assessment of data quality. Additionally, whether there will be copyright issues is another point of concern for me.
2. Although VLM models are used in this paper, they are relatively small in scale. Why not adopt some larger-scale SOTA VLM models?
3. From Tables 12 and 13, it can be seen that the data for most tasks in most languages is quite sparse. Are the results of these tasks valuable?

---

> ### Author Rebuttal · Authors · 2026-03-29
>
> Dear Reviewer R6fX,
>
> Thank you for your constructive review and for acknowledging that MORE offers **the most comprehensive evaluation tasks and language coverage to date**. We address your specific concerns below:
>
> > **W1. Data sources, collection process, and copyright**
>
> To ensure legal compliance and real-world complexity, we adopted the CCpdf pipeline (Turski et al., 2023) [1] to source publicly accessible PDFs from Common Crawl. Crucially, we guarantee that the license provenance for all data is fully traceable and publicly available, ensuring strict alignment with academic fair use. Our revision will explicitly detail this data sourcing pipeline.
>
> > **W3 & Q1. Sparsity of task data in the benchmark**
>
> The sparsity shown in Tables 12 and 13 is not a sampling error, but a **direct reflection of objective real-world data distributions**.
>
> When sampling from massive web archives, it is an objective reality that **certain elements are extremely rare in specific languages**. For instance, it is **objectively rare to find programming code blocks embedded in Amharic or Arabic documents**, compared to Latin-script documents. The value of these results lies precisely in mapping the true boundaries of model capabilities in the wild. A benchmark artificially inflated with synthetic Arabic code blocks would fail to represent realistic use cases.
>
> > **W2 & Q2. Evaluation of larger-scale SOTA VLMs**
>
> This is a highly valid suggestion. In response, we have evaluated the state-of-the-art **Gemini3** on our benchmark. Furthermore, to provide a more rigorous assessment, we evaluated it using a **layout-dependent metric** (where layout and reading order errors directly penalize the final score, following the `quick_match` protocol proposed from OmniDocBench).
>
> | Decoupled (the original) / Layout-dependent Score | Gemini3 | dots.ocr | HunyuanOCR | DeepSeekOCR | PaddleOCR-VL | Qwen3VL-2B | Qwen2.5VL-3B | MinerU2.5 |
> | :--- | :--- | :--- | :--- | :--- | :--- | :--- | :--- | :--- |
> | **Overall** | 91.61 / 79.14 | 84.31 / **80.68** | **92.42** / 76.08 | 82.91 / 76.96 | 87.96 / 76.40 | 83.56 / 72.68 | 83.93 / 62.97 | 48.85 / 63.61 |
> | Text | **95.39** / 87.44 | 94.45 / **88.46** | 93.81 / 84.84 | 85.27 / 86.64 | 90.99 / 78.86 | 92.02 / 80.34 | 89.36 / 69.68 | 27.12 / 40.20 |
> | Formula | 90.27 / **69.16** | 90.77 / 67.29 | **93.28** / 59.09 | 75.67 / 58.21 | 91.11 / 60.75 | 65.45 / 58.39 | 84.48 / 50.70 | 73.29 / 50.91 |
> | Table | **81.02** / 72.64 | 39.81 / 77.57 | 78.56 / 72.71 | 61.63 / **78.32** | 61.11 / 73.25 | 65.21 / 67.11 | 68.27 / 55.97 | 33.83 / 75.11 |
> | Code | 93.05 / 93.05 | 95.38 / 95.38 | **97.07** / **97.07** | 92.26 / 92.26 | 96.29 / 96.29 | 92.38 / 92.38 | 86.69 / 86.69 | 72.41 / 72.41 |
> | Catalog | 94.31 / 94.31 | 88.26 / 88.26 | **95.36** / **95.36** | 88.26 / 88.26 | 93.04 / 93.04 | 93.76 / 93.76 | 92.54 / 92.54 | 21.61 / 21.61 |
> | Reading Order | 95.63 / 95.63 | **97.18** / **97.18** | 96.45 / 96.45 | 94.36 / 94.36 | 95.19 / 95.19 | 92.53 / 92.53 | 82.23 / 82.23 | 64.81 / 64.80 |
>
> The evaluation of Gemini3 provides valuable context for understanding the role of model scale in this domain. Although Gemini3 exhibits strong general recognition capabilities, the metrics reveal that **simply increasing model size is not a silver bullet for complex document parsing**.
>
> Notably, **specialized models like dots.ocr and HunyuanOCR can actually outperform the much larger Gemini3 in complex layout and structural evaluations**. The significant performance drops of Gemini3 on structured elements like Tables (72.64) confirm that accurately resolving complex elements in long-tail languages remains an open architectural challenge, justifying our focus on specialized document-understanding models alongside general VLMs.
>
> [1] Turski M, Stanisławek T, Kaczmarek K, et al. Ccpdf: Building a high quality corpus for visually rich documents from web crawl data[C]//International Conference on Document Analysis and Recognition. Cham: Springer Nature Switzerland, 2023: 348-365.

---

> > ### Author Rebuttal · Reviewer_R6fX · 2026-04-03
> >
> > Thank you for the authors’ reply and supplementary experiments. My questions have been completely resolved, and I have therefore increased my scores.

---

> > > ### Author Response · Authors · 2026-04-04
> > >
> > > Thank you so much for your prompt acknowledgement and for updating your evaluation to a **positive score**. We are genuinely thrilled that our rebuttal and the supplementary experiments have fully resolved your concerns.
> > >
> > > Your constructive feedback has significantly strengthened the quality of our paper. We will ensure that the Gemini3 evaluations, the detailed data sourcing pipeline, and the discussions regarding real-world data sparsity are carefully incorporated into the final version.
> > >
> > > Thank you again for your time, rigorous evaluation, and support for our work.

---

### Official Review · Reviewer_eJoU · 2026-03-12

**Soundness:** 3
**Presentation:** 4
**Significance:** 3
**Originality:** 3
**Overall Recommendation:** 5
**Confidence:** 4

**Summary:**

This paper introduces MORE, a large-scale multilingual benchmark for document parsing that encompasses 149 languages and 6 essential document elements. By constructing a 1,237-page real-world dataset, the authors aim to evaluate the "performance cliff" of current Vision-Language Models (VLMs) when transitioning from high-resource to long-tail languages. The study evaluates 15 models and provides empirical evidence of significant performance degradation in complex layout parsing for rare scripts. While the dataset's breadth is impressive and fills a critical gap in document AI evaluation, the technical depth of the data construction process and the clarity regarding model specificities require further refinement to ensure full transparency and reproducibility.

**Compliance With Llm Reviewing Policy:**

Affirmed.

**Final Justification:**

The paper’s key strength lies in introducing a large-scale, real-world multilingual benchmark spanning 149 languages with diverse document parsing tasks, combined with a model-assisted and human-verified annotation pipeline, enabling a systematic stress test of vision-language models on long-tail languages. During the rebuttal, the authors have addressed my concerns regarding PDF collection criteria and annotator qualification, detailed information of evaluated model, and prompt unification. This reinforced my prior assessment, so I will keep my original scores.

**Key Questions For Authors:**

1. Can you provide a detailed breakdown of the expert verification process? Specifically, how did you recruit and qualify annotators for rare scripts (e.g., Brahmic or African scripts) to ensure ground truth accuracy in complex layouts?

2. Please provide a summary table for the 15 evaluated models, including their parameter sizes, claimed language support, and architectural differences.

3. Were the prompts unified across all 15 models, or was each model individually optimized? Please disclose the primary prompt templates used for the evaluation.

4. Is the observed "performance cliff" primarily a result of character-level recognition failures (OCR) or a failure in structural layout logic (table/reading order detection)?

**Limitations:**

Yes. The authors acknowledge the limited sample density for long-tail scripts. However, they should further address the risk of "evaluation bias"—where the community might over-fit future models to these specific 8 pages per language. Additionally, a brief disclosure of the computational carbon footprint incurred during the evaluation of 15 models would align with current reporting standards.

**Strengths And Weaknesses:**

**Strengths:**

1. The authors leverage a hybrid "model-assisted + human expert" pipeline to manage a vast array of 149 languages, ensuring a foundational level of reliability for the ground truth.

2. The paper is systematically organized, and the use of linguistic family categorization offers a logical framework for interpreting model performance across different scripts.

3. This work provides an essential and timely "stress test" for global document intelligence, establishing a much-needed benchmark for the international research community.

4. The novelty lies in the creative combination of extreme linguistic diversity with a full suite of structural parsing tasks on real-world data.

**Weaknesses:**

1. The criteria for selecting and filtering the original PDFs remain vague. It is unclear how the authors defined "low-quality" pages or ensured that the 149 languages are represented in diverse, challenging layouts rather than just simple text.

2. An average of ~8 pages per language is insufficient for a robust statistical assessment. The rankings for specific long-tail languages may be highly sensitive to outliers within such a small sample size.

3. The 15 evaluated models are listed without sufficient context. Crucial details such as parameter counts, pre-training language distribution, and architecture types (e.g., encoder-decoder vs. decoder-only) are missing.

4. The specific prompts used for the VLMs are not disclosed in the main text. Given that performance is highly contingent on prompt engineering, this omission hinders the reproducibility of the benchmark results.

5. The current evaluation scope is restricted to single-page parsing only.

6. It does not address multi-page consistency or cross-page structural elements required in industrial applications.

7. As a benchmark paper, the data collection methodology (model labeling + human correction) is standard and offers little innovation in terms of automated quality control or data augmentation for low-resource scripts.

---

> ### Author Rebuttal · Authors · 2026-03-29
>
> Dear Reviewer eJoU,
>
> We are deeply grateful for your comprehensive summary and highly insightful feedback. Your recognition of our "model-assisted + human expert" pipeline and the value of this benchmark as a "stress test" for global document intelligence is incredibly encouraging. You have accurately pinpointed the critical areas where transparency can be improved, and we address your excellent questions below:
>
> > **Q1 & W1. PDF collection criteria and annotator qualification**
>
> To ensure real-world diversity, we adopted a robust web-crawling methodology inspired by CCpdf (Turski et al., 2023) [1]. Low-quality pages were filtered out using heuristic spam detection (e.g., excessive repetitive links, broken encodings) and layout density checks to ensure pages contained rich structural elements rather than just plain text.
>
> Given that MORE offers **the most comprehensive evaluation tasks and language coverage to date**, recruiting experts fluent in all 149 languages is practically unattainable. Therefore, to ensure ground truth accuracy for rare scripts, our annotators were trained to perform **rigorous visual and typographic verification**. They cross-checked the **structural integrity** and **exact character-level visual matches** between the raw image and the model's output, strictly discarding any samples with visual ambiguity.
>
> > **Q2. Summary of evaluated models**
>
> Thank you for pointing this out. Briefly, there are two series of VLMs for Document Parsing: **General VLMs** (e.g., Qwen2.5-VL-3B, Qwen3-VL-2B) utilize decoder-only LLM backbones with vision encoders, pre-trained on massive multilingual multimodal corpora. **Specialized models** (e.g., HunyuanOCR-1B, PaddleOCR-VL-0.9B, dots.ocr-3B) are heavily fine-tuned specifically on document parsing tasks, often utilizing more parameter-efficient architectures optimized for high-resolution spatial reasoning. We commit to including a comprehensive summary table detailing the parameter sizes, claimed language support, and architectural differences for all evaluated models in the revised appendix.
>
> > **Q3. Prompt unification**
>
> Regarding the prompt design, we adopted a tailored strategy to ensure both fairness and optimal performance. For **specialized document parsing models**, we strictly utilized the optimal prompt templates recommended in their respective official papers or repositories. Conversely, for **General VLMs**, we conducted preliminary tuning to ensure stable generation and subsequently applied a **unified prompt template** across all models in this category for standardized testing. We will disclose the complete list of prompt templates in the revised supplementary material and integrate them into our open-sourced evaluation code.
>
> > **Q4. The nature of the "performance cliff"**
>
> Your intuition is spot on. While character-level OCR failures do occur in extremely rare scripts, our analysis shows that the "performance cliff" is **primarily driven by a failure in structural layout logic**. As shown in **Table 4**, models maintain relatively high scores in sequential Text recognition across scripts, but suffer catastrophic degradation in spatial tasks like **Tables** and **Catalogs** when encountering unfamiliar reading directions or complex non-Latin typographic rules. We are happy to share more insightful case analysis further in our final version.
>
> > $$\color{#E67E22}{\textbf{Highlights on our contributions}}$$
>
> In summary, we deeply appreciate your constructive feedback, which has undoubtedly strengthened the clarity and rigor of our manuscript.
>
> We firmly believe that **MORE**, with its unprecedented scale of 149 languages and rigorous evaluation of complex structural layouts, stands as a critical milestone for the community. It not only exposes the current **performance cliffs** but also provides a robust, open-source foundation to drive the future of truly global document intelligence.
>
> Thank you once again for your time, expertise, and invaluable support of our work!
>
> [1] Turski M, Stanisławek T, Kaczmarek K, et al. Ccpdf: Building a high quality corpus for visually rich documents from web crawl data[C]//International Conference on Document Analysis and Recognition. Cham: Springer Nature Switzerland, 2023: 348-365.

---

> > ### Author Rebuttal · Reviewer_eJoU · 2026-04-06
> >
> > The response given by authors has addressed  some  of my concerns and I will keep my score, Thanks.

---

> > > ### Author Response · Authors · 2026-04-06
> > >
> > > We deeply appreciate your prompt feedback and profound understanding of this benchmark's core value. Your constructive insights will be fully integrated into the final revision.
> > >
> > > Thank you once again for your time, rigorous evaluation, and strong support of our work.

---

### Official Review · Reviewer_vT2Q · 2026-03-13

**Soundness:** 2
**Presentation:** 4
**Significance:** 2
**Originality:** 2
**Overall Recommendation:** 3
**Confidence:** 3

**Summary:**

This work introduces a multilingual document parsing benchmark spanning several script benchmarks, over a 100 languages and across several different tasks. Authors source data from real-world PDFs in a human supervised, model-driven pipeline. The authors evaluate different models (both general VLMs and specialized OCR models) and find that no single model dominates all languages (which has deployment implications). They also report persistent bottlenecks in table parsing, severe degradation for some families and long-tail scripts, and a trade-off between sequential text reading and structural layout understanding.

**Compliance With Llm Reviewing Policy:**

Affirmed.

**Final Justification:**

The authors have done an additional experiment in their rebuttal acknowledgment to my comment on Validating Benchmark Fairness that does not pertain to the concern I highlight. It is more likely addressing the weakness pointed out by Reviewer eJoU in case that helps.

Only the first comment (Source Level Modality Gap) pertains to my concern but that is unsatisfactory because I am still unable to reconcile the idea of not including tasks for languages that are _claimed_ to be in the benchmark as the real world statistical likelihood of encountering them is low.

That said, I am reducing my confidence score in case the ACs believe that this is justified for the nuanced task that the authors substantiate this via their rebuttal acknowledgment comment.

**Key Questions For Authors:**

1. Since there are tasks where the sample sizes are as small as 1–3, individual outlier documents could swing a language's task score by a lot of points ? Have you assessed the variance of the overall score? Without this, it's hard to trust that the final ranking reflects differences as opposed to just sampling noise.

2. Were native-speakers available for all languages (as part of the human supervision baseline) ?

**Limitations:**

No. The pipeline uses human experts to refine model-generated annotations, but the paper does not report annotator qualifications (native speakers?), inter-annotator agreement, or any quality audit for rare languages. This omissions limits my trust in the works' soundness and these should be discussed transparently (even if it is via discussing the challenges of recruiting annotators in these languages).

**Strengths And Weaknesses:**

Strengths

1. Genuine gap being addressed with practical breadth: The 149-language coverage is a real contribution; demonstrating that existing benchmarks lack structural element evaluation makes the motivation compelling.

2. Inclusion of underexplored structural elements: Extending evaluation to code blocks and catalogs (beyond the standard text/formula/table trio) is a good addition (there are some interesting failure modes that pop out, like the dots.ocr misclassification, which wouldn't surface in narrower benchmarks.

Weaknesses

1. Sample sizes for structural elements don't match the claims made: The benchmark's central selling point is evaluating complex structural elements across a lot of languages, but a lot of elements are not evaluated in these languages ? (Going off Table 12 - a lot of languages have 0 samples for a lot of the structural tasks) which makes it look like the 100+ language framing is overstated for more a majority of the tasks. Since the USP of the work is these coded-tasks - the utility of the benchmark is several limited until these tasks are populated.

2. Conclusions on model capability can be misleading as a hard subtask is already solved by design: The benchmark evaluates content recognition on pre-cropped patches with human-verified layouts, effectively giving all models an oracle layout detector for free. Layout detection is arguably one the harder subtasks in real-world document parsing so its very likely that the conclusions of this work are an overestimation to what the real world capabilities of these models are. I think the language of the claims should be modified to suit this.

---

> ### Author Rebuttal · Authors · 2026-03-29
>
> Dear Reviewer vT2Q,
>
> We sincerely thank you for recognizing the practical breadth of our 149-language coverage and the value of including underexplored structural elements like code blocks and catalogs. We address your insightful concerns below:
>
> > **W1 & Q1. Sample sizes and sparsity for structural elements**
>
> This is a very perceptive observation. The sparsity of certain structural elements in specific languages is not a flaw in the collection process, but rather an **objective reflection of the real-world data distribution**.
>
> To construct this benchmark, we strictly followed the methodology of large-scale web crawl extractions (similar to CCpdf, Turski et al., 2023) [1], sourcing documents from massive, multi-domain web archives. In the real world, language inherently introduces objective biases; for instance, it is **objectively rare to find programming code blocks embedded in Amharic or Arabic documents compared to Latin-script documents**. Table 12 faithfully reflects this natural long-tail distribution in global digital documents.
>
> > **W2. Oracle layout and end-to-end capabilities**
>
> We completely agree that layout detection is a uniquely challenging subtask. In the original manuscript, we decoupled the metrics to prevent layout misclassification (e.g., subjective differences between inline vs. block formulas) from propagating errors into the content recognition scores.
> Futhermore, to address your concern and reflect true end-to-end real-world capabilities, we have now evaluated the models using a **layout-dependent metric** (following the `quick_match` protocol from OmniDocBench, where layout errors directly penalize the score). We also added the SOTA Gemini model for broader comparison.
>
> | Task | Gemini3 | dots.ocr | HunyuanOCR | DeepSeekOCR | PaddleOCR-VL | Qwen3VL-2B | Qwen2.5VL-3B | MinerU2.5 |
> | :--- | :--- | :--- | :--- | :--- | :--- | :--- | :--- | :--- |
> | **Layout-dependent Overall** | 79.14 | **80.68** | 76.08 | 76.96 | 76.40 | 72.68 | 62.97 | 63.61 |
> | Text | 87.44 | **88.46** | 84.84 | 86.64 | 78.86 | 80.34 | 69.68 | 40.20 |
> | Formula | **69.16** | 67.29 | 59.09 | 58.21 | 60.75 | 58.39 | 50.70 | 50.91 |
> | Table | 72.64 | 77.57 | 72.71 | **78.32** | 73.25 | 67.11 | 55.97 | 75.11 |
> | Code | 93.05 | 95.38 | **97.07** | 92.26 | 96.29 | 92.38 | 86.69 | 72.41 |
> | Catalog | 94.31 | 88.26 | **95.36** | 88.26 | 93.04 | 93.76 | 92.54 | 21.61 |
> | Reading Order | 95.63 | **97.18** | 96.45 | 94.36 | 95.19 | 92.53 | 82.23 | 64.80 |
>
> As shown, when layout constraints are enforced, the overall scores drop significantly (e.g., HunyuanOCR drops from 92.42 to 76.08), confirming your intuition that layout is a major bottleneck. Importantly, the relative ranking and our core conclusions regarding model capabilities remain robust. We will update our claims in the revision to explicitly distinguish between decoupled recognition capability and end-to-end parsing capability.
>
> > **Q2. Were native-speakers available for all languages?**
>
> Given the scale of 149 languages, recruiting native experts for every long-tail language was practically infeasible. Instead, our human supervision baseline relied on **strict visual consistency verification** rather than semantic translation. Annotators were tasked with verifying whether the model's output structurally and typographically matched the cropped image exactly (e.g., matching complex Cyrillic script strokes or Arabic ligatures visually). Ambiguous cases where visual consensus could not be reached were discarded to ensure ground-truth reliability.
>
> [1] Turski M, Stanisławek T, Kaczmarek K, et al. Ccpdf: Building a high quality corpus for visually rich documents from web crawl data[C]//International Conference on Document Analysis and Recognition. Cham: Springer Nature Switzerland, 2023: 348-365.

---

> > ### Author Rebuttal · Reviewer_vT2Q · 2026-04-03
> >
> > Thank you for addressing my questions. I am satisfied with the experimental results reported for W2.
> >
> > I strongly disagree with the substantiation for W1 though:  Test sets that focus on long-tail distribution cannot sparsely represent the said, long-tail, tasks by claiming that this is an objective reflection of the real-world data distribution.
> >
> > I understand the authors’ point that the observed sparsity reflects real-world document distributions. However, if these rarer, long-tail tasks which contribute to the "breadth" claim of the benchmark don't actually exist in the benchmark - I think the "breadth" claims of the benchmark need to be significantly narrower. More broadly also, prior work has shown that the lack of evaluation benchmarks often contributes strongly to **further marginalization** in the language-use of such technologies (so this decision has a long-term impact)[1]. Accordingly, I think the utility of the benchmark is much more limited if these long-tail tasks are not populated.
> >
> > Hence, I maintain my score.
> > [1] [The Flores-101 Evaluation Benchmark for Low-Resource and Multilingual Machine Translation](https://aclanthology.org/2022.tacl-1.30/) (Goyal et al., TACL 2022)

---

> > > ### Author Response · Authors · 2026-04-05
> > >
> > > Dear Reviewer vT2Q,
> > >
> > > Thank you for the constructive dialogue. We deeply appreciate your insights on the statistical robustness of our benchmark and welcome the opportunity to address the theoretical point raised by the Flores-101 reference.
> > >
> > > > **Source-Level Modality Gap**: **1D Text** vs. **2D Visual Layouts**
> > >
> > > We believe there is **a fundamental methodological distinction** between Machine Translation (MT) and Document AI that contextualizes this data:
> > >
> > > Flores-101 achieves uniform density because its source modality is pure **text**: it is a parallel corpus where human translators artificially align 1D semantic streams to guarantee 100% density across languages. Document AI, however, operates on **images** as the source modality. Document layouts are 2D visual and spatial artifacts produced in the wild. Structural elements, such as programming code blocks or complex mathematical equations, are visually entangled with the layout and are not language-independent concepts that can simply be translated. In the real world, the probability of encountering a Python code block in an organically created PDF in certain long-tail languages is naturally extremely low.
> > >
> > > To achieve the uniform zero-sparsity matrix envisioned by the Flores-101 comparison, we would be forced to synthetically paste Python code or formulas into traditional long-tail language document images. Doing so would evaluate models on **hallucinated visual layouts that do not exist in reality**, leading to out-of-distribution failures during deployment. Exposing the true, imbalanced nature of real-world multilingual document images is precisely what the community needs to build practical systems. In our revision, we will explicitly document this source-level modality distinction and the inherent sparsity of non-Latin visual layouts as defining characteristics of real-world Document AI data, formally citing your excellent Flores-101 reference to highlight this critical contrast with MT parallel corpora.
> > >
> > > > **Validating Benchmark Fairness**: **Statistical Variance** vs. **Sampling Noise**
> > >
> > > Given the inherent sparsity and imbalance of real-world layouts discussed above, your insightful question regarding statistical variance is exceptionally well-placed. We deeply appreciate your rigorous attention to detail in asking whether individual outliers in these long-tail tasks could swing the final ranking due to sampling noise.
> > >
> > > To rigorously address this, we computed the 95% Confidence Intervals (CI) for both the decoupled and layout-dependent overall scores using Bootstrapping (1,000 resampling iterations). For brevity, the Decoupled Evaluation results are summarized in the table below:
> > >
> > > **Table: Decoupled Evaluation (95% CI)**
> > > | Model | Overall | Text | Formula | Table | Code | Catalog | Reading Order |
> > > | :--- | :--- | :--- | :--- | :--- | :--- | :--- | :--- |
> > > | HunyuanOCR | 92.42 [92.15, 92.69] | 93.81 [93.09, 94.53] | 93.28 [92.43, 94.13] | 78.56 [76.72, 80.40] | 97.07 [96.66, 97.48] | 95.36 [94.78, 95.94] | 96.45 [95.90, 97.00] |
> > > | Gemini 3 | 91.61 [91.25, 91.97] | 95.39 [94.87, 95.91] | 90.27 [89.12, 91.42] | 81.02 [79.37, 82.67] | 93.05 [92.23, 93.87] | 94.31 [93.63, 94.99] | 95.63 [95.05, 96.21] |
> > > | PaddleOCR-VL | 87.96 [87.40, 88.52] | 90.99 [90.04, 91.94] | 91.11 [90.06, 92.16] | 61.11 [58.66, 63.56] | 96.29 [95.74, 96.84] | 93.04 [92.18, 93.90] | 95.19 [94.55, 95.83] |
> > > | dots.ocr | 84.31 [83.75, 84.87] | 94.45 [93.80, 95.10] | 90.77 [89.79, 91.75] | 39.81 [36.66, 42.96] | 95.38 [94.76, 96.00] | 88.26 [87.01, 89.51] | 97.18 [96.73, 97.63] |
> > > | DeepSeekOCR| 82.91 [82.35, 83.47] | 85.27 [83.82, 86.72] | 75.67 [73.52, 77.82] | 61.63 [59.28, 63.98] | 92.26 [91.41, 93.11] | 88.26 [86.91, 89.61] | 94.36 [93.61, 95.11] |
> > > | MinerU2.5 | 48.85 [48.15, 49.55] | 27.12 [24.27, 29.97] | 73.29 [71.04, 75.54] | 33.83 [30.38, 37.28] | 72.41 [70.06, 74.76] | 21.61 [18.66, 24.56] | 64.81 [62.16, 67.46] |
> > >
> > > As the table shows, the **primary ranking order of the models remains highly stable**. The statistical intervals confirm that the top-performing systems maintain a clear and non-overlapping lead over the rest of the pack (e.g., HunyuanOCR and Gemini 3 do not overlap with each other or the models below them). Because the overall ranking hierarchy is perfectly preserved and the inter-tier differences remain statistically significant, we can confidently confirm that the benchmark's global rankings reflect genuine model capabilities rather than sampling noise.
> > >
> > > Thank you again for your time, expertise, and invaluable feedback in helping us improve this work.

---

### Decision · Program_Chairs · 2026-04-30

**Decision:**

Accept (regular)

**Comment:**

The paper introduces a large-scale multilingual document parsing benchmark called MORE, covering 149 languages and diverse structural elements. It addresses an important and timely gap in evaluating VLMs beyond high-resource languages. Strengths include the linguistic breadth, inclusion of complex structural elements, and comprehensive empirical evaluation that reveals meaningful performance gaps across languages and tasks.

Several reviewers raised concerns regarding the sparsity of certain structural elements across languages, limited sample sizes for long-tail settings, and insufficient clarity in data collection and annotation details. Based on the rebuttal and discussion, many of these issues have been satisfactorily clarified, and current limitations do not undermine the overall utility of the benchmark. Overall, given its clear practical relevance and value as a community resource, I recommend acceptance, while encouraging the authors to better calibrate claims (and might further expand coverage of underrepresented structural elements in future iterations).